# Quantifying the global climate feedback from energy-based adaptation

Alexander C. Abajian [1] ✉, Tamma Carleton [2,3] ✉, Kyle C. Meng [1,3,4] ✉ & Olivier Deschênes[1,3]

Many behavioral responses to climate change are carbon-intensive, raising concerns that adaptation may cause additional warming. The sign and magnitude of this feedback depend on how increased emissions from cooling balance against reduced emissions from heating across space and time. We present an empirical approach that forecasts the effect of future adaptive energy use on global average temperature over the 21st century. We estimate that energy-based adaptation will lower global mean surface temperature in 2099 by 0.07 to 0.12 °C relative to baseline projections under Representative Concentration Pathways 4.5 and 8.5. This cooling avoids 0.6 to 1.8 trillion U.S. Dollars ($2019) in damages, depending on the baseline emissions scenario. Energy-based adaptation lowers business-as-usual emissions for 85% of countries, reducing the mitigation required to meet their unilateral Nationally Determined Contributions by 20% on average. These findings indicate that while business-as-usual adaptive energy use is unlikely to accelerate warming, it raises important implications for countries' existing mitigation commitments.

Human adaptation will be increasingly critical for moderating harms and exploiting opportunities under climate change[1]. Recent studies highlight that climate adaptation requires significant changes in energy use; increased energy consumption has been shown to reduce excess mortality and protect well-being in homes, workplaces, and schools under extreme temperatures[2–10]. Energy use is carbon-intensive: cooling demand alone comprised 10% of recent global electricity consumption and is expected to rise substantially during the 21st century[11]. This raises the question of whether adaptation to climate change may itself induce additional warming, a phenomenon we call the Climate Adaptation Feedback (CAF). The CAF is the anthropogenic analog to geophysical climate feedback mechanisms (e.g., declines in oceanic $CO_2$ uptake or albedo) that can amplify global climate change[12]. It captures how behavioral responses to climate change may be maladaptive by ultimately increasing future global mean surface temperatures (GMST)[10,13,14].

Prior work has noted and made progress toward assessing the potential feedback between energy used for adaptation and the climate. Recent analyses quantify the responsiveness of historical energy demand to weather and use such estimates to project future global energy consumption under climate change[8,15,16]. However, these studies do not assess the impacts of these changes in energy consumption on global temperatures. One study does estimate a version of the CAF using a regional Integrated Assessment Model (IAM), modeling adaptation as changes in the productivity of different fuels used as inputs in production, and finds a positive feedback in which adaptive energy use raises global mean surface temperature[17]. This analysis assumes regional planners set optimal policy for energy use, adaptation, and mitigation[18] and it captures key general equilibrium channels, such as how adaptation affects prices, expectations, and investment.

This paper develops a data-driven framework for quantifying the CAF driven by adaptive energy consumption. We calculate the CAF over the course of the 21st century by combining high-resolution, subnational projections of energy consumption responses to

[1]Department of Economics, University of California, Santa Barbara, CA, USA. [2]Department of Agricultural and Resource Economics, University of California, Berkeley, CA, USA. [3]National Bureau of Economic Research, Cambridge, MA, USA. [4]Bren School of Environmental Science and Management, University of California, Santa Barbara, CA, USA. ✉e-mail: alexander_abajian@ucsb.edu; tcarleton@berkeley.edu; kmeng@bren.ucsb.edu

anthropogenic climate change with country- and energy-specific $CO_2$ emissions intensities. Our calculation accounts for energy consumption implications of all behaviors and investments that individuals, communities, and firms undertake in response to temperature change across all non-transport sectors including residential, commercial, industrial, and agricultural sectors[8], covering 142 countries or nearly 80% of current global $CO_2$ emissions[19]. It is built from state-of-the-art, globally-comprehensive, empirical estimates of how energy demand responds to changing temperatures[8]. These estimates incorporate heterogeneous effects of a changing climate on energy consumption across fuels and locations[2,3,8,9,15,20–22], as well as how income growth and climate change will alter these responses over time.

In contrast to prior assessments of the CAF[17], we do not impose the structure of an IAM. Instead, we assess how adaptation to climate change will affect future temperatures using empirically-estimated energy demand responses and exogenous future trajectories of baseline warming, economic growth, and population (see "Methods"). Relative to earlier analyzes, our study increases the spatial resolution of energy demand responses used to forecast changes in emissions, includes the response to the full distribution of daily temperature realizations rather than restricting inputs to realizations of local extremes, reports uncertainty from both climatological and statistical sources, and accounts for country-level heterogeneity in emissions intensities and their dynamics for electricity and other fuels. However, our statistically-based method necessarily foregoes modeling general equilibrium channels that are captured by IAMs, making the two approaches complementary.

Prior research and media attention have predominantly focused on the risks posed by behavioral feedbacks that would lead to a positive CAF, such as increased demand for air conditioning raising emissions from the electricity sector[4,11,17,23]. While this specific channel will undoubtedly play an important role, Fig. 1 illustrates that both the sign and magnitude of the CAF stemming from all forms of energy-based adaptation to climate change are unclear. While climate change will lead to warming of daily temperatures in all locations (Fig. 1a, b), the resulting response in energy consumption will be highly heterogeneous. A higher frequency of realized hot days will increase electricity demanded for cooling services in locations that are already warm (Fig. 1d), but warming will simultaneously lower demand for heating in locations that currently experience a large number of cold days (Fig. 1c). This heterogeneity interacts with variation in the $CO_2$ intensities of electricity and other sources of energy, leading to substantial differences in the response of emissions from adaptation to temperature change across countries.

For example, Canada and Sweden will both likely experience declines in heating demand under climate change. However, Canada's electricity sector emits 15 times more carbon dioxide per Gigajoule ($CO_2 \times GJ^{-1}$) than does Sweden's, while its other fuels sector emits 2 times more. These differences lead to differential emissions responses to changing daily temperatures (Fig. 1e). Similarly, energy-based adaptation in India and Brazil will lead to increased demand for electricity due to more energy being required for refrigeration and indoor temperature control, but the dominance of coal in India implies much larger increases in emissions than in Brazil, where hydropower is the primary source of electricity (Fig. 1f). Thus, the change in future global $CO_2$ emissions due to energy-based adaptation, and by extension the sign and magnitude of the CAF, are a priori unknown. A positive CAF implies behavioral adaptations increase global $CO_2$ emissions on net (Fig. 1g), raising projected rates of warming (Fig. 1h). However, if future declines in emissions due to reduced demand for heat from other fuels dominate the additional future emissions due to higher electricity demand, the CAF will be negative.

We develop a framework to sign and quantify the net effects of the forces illustrated in Fig. 1. Specifically, we define the CAF in any given year as the difference in global mean surface temperature between a baseline value (e.g., projected warming under the Representative Concentration Pathway 8.5 (RCP8.5) emissions trajectory) and one accounting for future energy-based adaptation (e.g., projected warming under RCP8.5 plus the net change in emissions from adaptation) (see "Methods"). We note that while energy demand responses to warming in specific sectors and by specific energy sources will be heterogeneous, as illustrated in Fig. 1, the sign and magnitude of the CAF depend only on the net change in resulting emissions across all sectors and sources. Therefore, we focus our analysis on the aggregate emissions changes resulting from all measurable energy-based adaptations.

We implement this framework by leveraging high-resolution projections of future energy changes in response to local temperature realizations from ref. 8. These causal "dose-response" functions represent the change in the use of final energy sources—electricity and all other fuels—in response to variations in daily temperature, pooling energy consumption across residential, commercial, industrial, and agricultural end-uses (excluding transportation). Any adaptive actions taken by individuals, firms, or public agencies across a broad spectrum of sectors, such as the use of air conditioning or space heating, are included in these estimates. The estimated dose-response functions vary across space and time, accounting for the fact that average incomes and baseline climates shape the sensitivity of energy use to temperature changes, for example through changing the adoption and efficiency of energy-intensive technologies. Accounting for such extensive margin adjustments in the energy sector has been shown to have first-order impacts on projected impacts of climate change on energy demand[4,9,15,23]. To account for uncertainty, we report estimates from multiple socioeconomic and emissions scenarios, while accounting for both statistical and climatological uncertainty (see "Methods").

We combine these projections with country-level $CO_2$ emissions intensity factors for each final energy source constructed using data from the International Energy Agency's Emissions Intensities Report (see "Methods"). These granular data are critical for translating adaptive energy use into a global CAF, as the $CO_2$ intensity of energy use varies substantially across fuels and locations, as shown in Fig. 2a, b. When combined with projections of energy consumption from ref. 8, these factors allow us to predict future changes in global $CO_2$ attributable to energy-based adaptation (see "Methods"). As our forecasts for future energy consumption take an underlying emissions (RCP) and socioeconomic (Shared Socioeconomic Pathway; SSP) scenario as given, we fix emissions factors in our projection at historical 2010-2018 levels. While this assumption is restrictive, it avoids the inconsistency that would result from changing emissions factors while maintaining an SSP-RCP that fixes baseline emissions. We discuss and evaluate the potential implications of this assumption for our estimates of the CAF below.

To obtain the CAF, we calculate the annual cumulative change in global $CO_2$ emissions due to energy-based adaptation for horizons from 2020 to 2099 (Fig. 2c). We then translate these cumulative emissions into a change in global temperatures ($\Delta$GMST) using an empirically-derived relationship that leverages simulated warming from an ensemble of Global Climate Models for the two emissions pathways we consider (Fig. 2d; Methods). This method contrasts with other approaches (cf. ref. 17) in that we abstain from integrating the energy demand responses from ref. 8 into an IAM, where adaptation through energy use is an endogenous outcome when agents solve a fully-specified optimization problem. This allows us to be neutral on how future policy may affect adaptive energy use and instead calculate the CAF as a function of future adaptation solely based on historical empirical estimates of the effects of temperature variation on energy consumption.

In this work, we estimate that the CAF is negative at all horizons and decreasing monotonically over time. This feedback breaks the

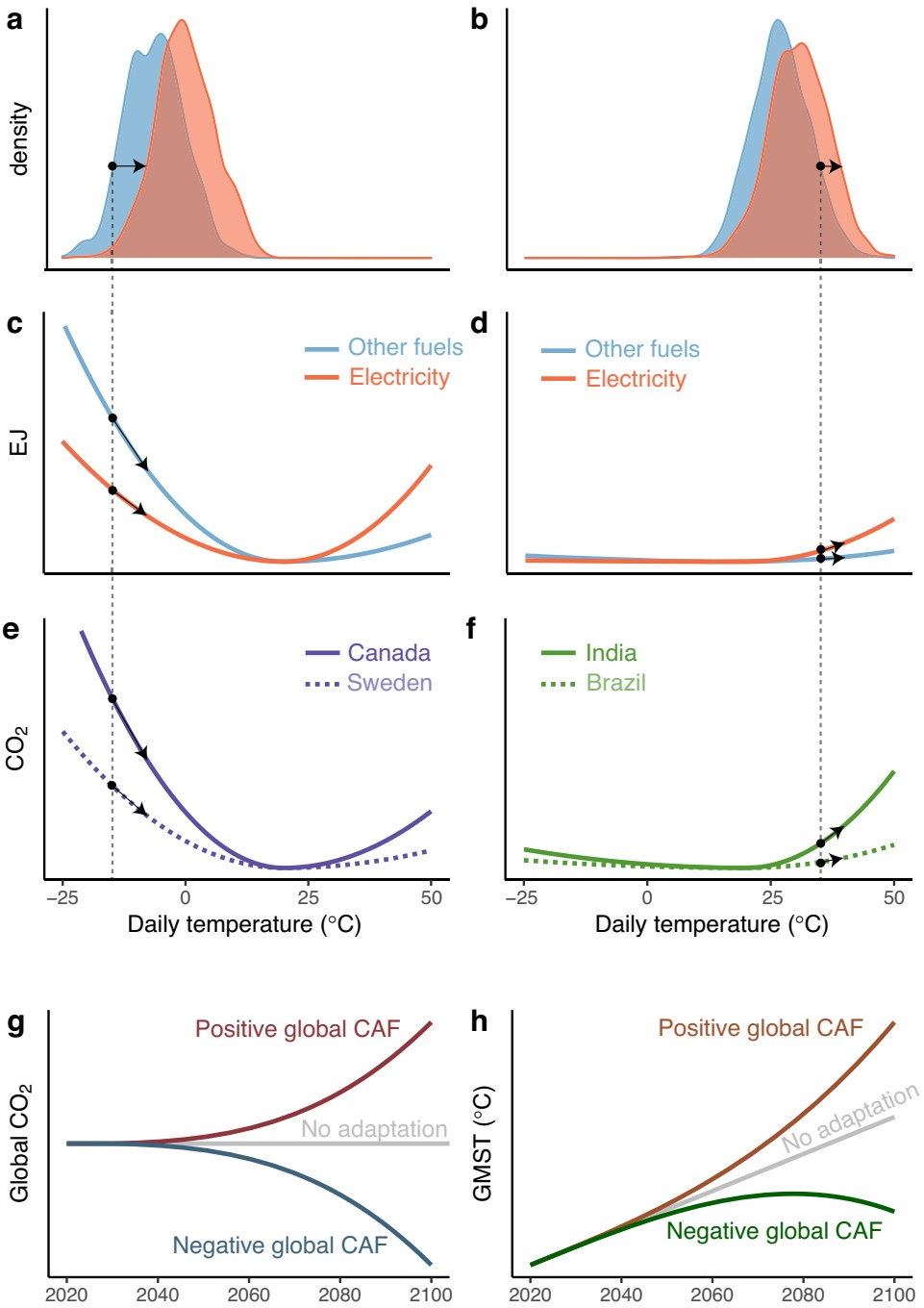

**Fig. 1 | The ambiguous effects of energy-based adaptation on global carbon dioxide ($CO_2$) emissions and global mean surface temperature.** The Climate Adaptation Feedback (CAF) is the net effect of adaptation-induced energy use on global mean surface temperatures (GMST); its sign is theoretically ambiguous. Climate change generates a rightward shift across heterogeneous baseline climate distributions for (**a**) colder and (**b**) warmer locations. This leads to (**c**) declines in energy consumption in cold locations and (**d**) increases in energy consumption in hot locations. Country-specific emissions intensities of electricity and other fuels result in different impacts of changing energy consumption on $CO_2$ emissions in cold locations and (**f**) hot locations. **g** Increases in emissions from elevated cooling demand on hot days balance against decreases in emissions from declining heating demand on cold days, making the net effect on global $CO_2$ emissions ambiguous. **h** When increased emissions from cooling outweigh decreased emissions from heating, a positive CAF increases GMST compared to a baseline rate of warming; when the opposite is true and emissions reductions from decreased heating demand outweigh increased emissions from cooling, the CAF is negative. Only with no energy-based adaptation is there no feedback on GMST.

conventional separation between mitigation (actions that reduce emissions) and adaptation (protective efforts), with important implications for both climate policy and research. For example, global climate models estimate climate responses to exogenous radiative forcing, but omit behavioral responses. A non-zero CAF implies associated global and local temperature projections are inaccurate,

affecting downstream climate damage estimates and key policy metrics like the social cost of carbon[24,25]. Additionally, the CAF has equity implications. The possibility of a positive CAF has raised concerns that adaptation could exacerbate climate change inequities, since the indoor temperature control that is more accessible at higher incomes[4,22] may accelerate climate change damages borne by those for

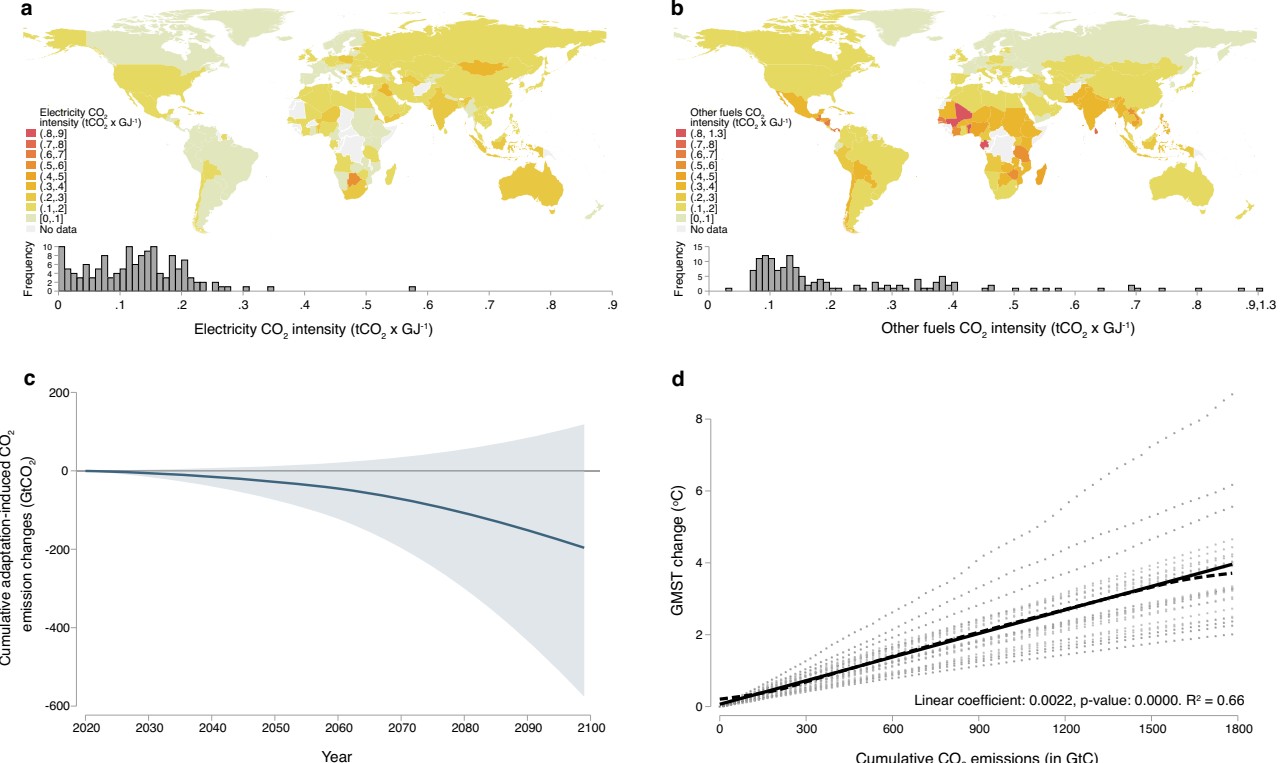

**Fig. 2 | Constructing the Climate Adaptation Feedback (CAF).** The CAF is constructed by combining high-resolution projections of climate change impacts on energy consumption from ref. 8 with the following emissions factors: (**a**) country-level carbon dioxide ($CO_2$) intensities from a gigajoule (GJ) of electricity use (in tons $tCO_2 \times GJ^{-1}$); and (**b**) country-level $CO_2$ intensities for all other fuels combined ($tCO_2 \times GJ^{-1}$). Maps (produced using Stata software with the `spmap` package using GADM shapefiles[73]) show country-level average factor values over the 2010–2018 period. Together, these data allow us to in (**c**) compute the mean value (solid line) and 90 percent confidence intervals (shaded region) of cumulative adaptation-induced $CO_2$ emission changes (in gigatonnes of $CO_2$) for 2020–2099 under our baseline scenario, which pairs Shared Socioeconomic Pathway 2 (SSP2) with Representative Concentration Pathway 8.5 (RCP8.5). Finally, we estimate (**d**) the relationship between projected GMST change (in °C) and cumulative $CO_2$ emissions across RCPs and global climate models over 2020–2099 (see "Methods"). This plot shows a fitted linear model (solid line) with 90% confidence intervals (shaded area) and point estimate and *p*-value of the linear coefficient, as well as a local polynomial fit (dashed line) using an Epanechnikov kernel with a rule-of-thumb bandwidth[71] (*N* = 5200).

whom indoor temperature control is unaffordable[26–29]. Our approach, inclusive of statistical and climatological uncertainty, suggests that a modest positive CAF is unlikely, alleviating such concerns. Finally, a non-zero CAF has ramifications for integrated assessment models. While these models increasingly account for geophysical feedbacks and nonlinearities[30] as well as behavioral margins[17,31–33], it remains difficult for existing IAMs to directly model how local changes in adaptive energy consumption alter global temperatures.

## Results

### Empirical estimation of the global climate adaptation feedback

We find a negative global CAF that declines over time, moderating rates of warming under future anthropogenic climate change. Fig. 3a plots point estimates (solid green line) and 90% confidence intervals (shaded green area) for the annual CAF over the 2020–2099 horizon under our baseline SSP2-RCP8.5 scenario. In 2099, under RCP8.5 the CAF is −0.12 °C; changes in energy consumption driven by adaptation lead to a 0.12 °C lower GMST relative to baseline. Under RCP4.5, our more moderate scenario for emissions and projected warming, the magnitude of the CAF in 2099 falls to −0.07 °C. This projected reduction in warming in 2099 under RCP8.5 alone is equivalent to six years of recent warming at the observed 0.018 °C $\times yr^{-1}$ rate between 1981–2019[34] and is 25 times larger than that implied by a back-of-the-envelope calculation in ref. 8. Under the SSP2-RCP8.5 scenario, our estimated CAF implies that adaptive energy consumption is predicted to lower the change in GMST in 2099 from 4.27 °C to 4.15 °C, relative to the pre-industrial climate. Using the Data-driven Spatial Climate

Impact Model (DSCIM) built by the Climate Impact Lab, we estimate that the decrease in warming due to the CAF lowers the present value of cumulative damages from climate change between 2020 and 2099 by 1.8 trillion ($2019 USD) (see "Methods"). The analog in avoided damages is 0.6 trillion for the RCP4.5 scenario. Accounting for both climatological and statistical uncertainty in 2099 yields a 90% confidence interval for the CAF of −0.35 to 0.073 °C, and the CAF is robust to iteratively leaving out the adaptation of individual countries when forecasting cumulative emissions reductions in 2099 (Supplementary Note 5).

A limitation of our benchmark approach is that it implicitly assumes the effects of adaptive emissions on the GMST pathway do not themselves affect future adaptation. Paths for future changes in energy consumption are fixed in the sense that the demand responses generated by temperature trajectories under each SSP-RCP are determined before we calculate the CAF at a given horizon. We relax this assumption to construct a dynamic version of the CAF, which allows for historical adaptation to affect our projections for adaptive energy use from that point forward (see "Methods"). Allowing for such dynamic linkages leads to a negligible difference, as shown by the dashed green line in Fig. 3a; the two CAFs are indistinguishable given the degree of statistical uncertainty. Our point estimates imply that accounting for dynamic adaptation effects decreases the magnitude of the CAF in 2099 by 0.8%.

We also recalculate the CAF relaxing the assumption of fixed emissions factors. The long dashed lime green line in Fig. 3a shows the CAF under the assumption that historical global trends in emissions

 

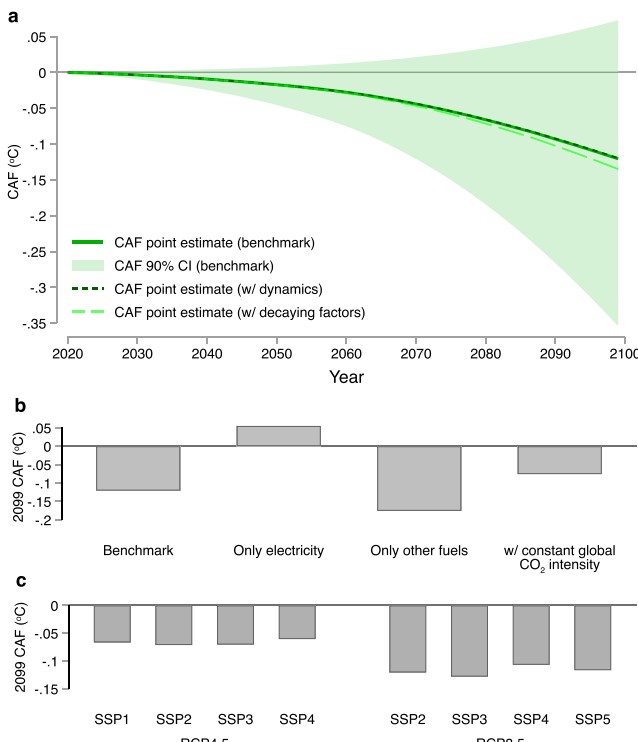

**Fig. 3 | The Climate Adaptation Feedback. a** Solid green line and shaded green area show point estimates and 90% confidence intervals for the mean value of the Climate Adaptation Feedback (CAF) in °C for 2020-2099 under Shared Socio-economic Pathway 2 paired with Representative Concentration Pathway 8.5 (SSP2-RCP8.5) using our benchmark approach. The darker dashed line shows the dynamic CAF, which accounts for how additional climate change from adaptation feeds back into future adaptation (see "Methods"). The lighter dashed line shows the CAF recalculated using emissions factors that follow historical trends between 2000 and 2018, as detailed in Supplementary Note 3. **b** Components of the benchmark CAF in 2099 under our baseline SSP2-RCP8.5 scenario. The first bar is the full CAF (consistent with panel (**a**) for 2099). The second bar shows the CAF component derived from electricity consumption alone. The third bar shows the CAF component derived from only other fuels consumption. The fourth bar shows the CAF component derived using a globally constant $CO_2$ emissions intensity, ignoring heterogeneity both across space and across fuels in emissions intensity of energy-based adaptation. **c** The last set of bar graphs show point estimates for the CAF in 2099 under different combinations of SSPs and RCPs.

intensity for both fuels between 2000 and 2018 continue for all countries through 2099, an assumption taken in prior related work on global climate damages[35,36]. Allowing each sector to continue to decarbonize at historical rates lowers the CAF to −0.13 °C under SSP2-RCP8.5. As emissions from other fuels have remained relatively unchanged on a unit basis when compared to the substantial reduction in the emissions intensity of electricity generation over the past two decades, extrapolation of this trend lowers emissions from the electricity sector and amplifies the mechanisms leading to a negative CAF (Supplementary Note 3).

### Decomposition of the global climate adaptation feedback

Figure 3b illustrates how several factors contribute to the sign and magnitude of the benchmark CAF reported by the first bar. First, adaptation-induced changes in electricity consumption, which are largely driven by increased demand for cooling under rising temperatures[4,8,37], lead to a positive value of 0.06 °C (second bar). This electricity effect has been the focus of prior discussions of energy-based adaptation[11,23]. However, a negative CAF emerges when we add changes in demand for other fuels, whose value of −0.18 °C

(third bar) more than offsets the positive component from electricity, due to substantial projected declines in demand for heating under climate change. These findings highlight the importance of accounting for all forms of energy demand that will change in response to climate change. Moreover, they reinforce results in the previous literature demonstrating that energy demand responses to climate change differ in sign and magnitude across distinct sectors and fuel sources[15].

Heterogeneity in $CO_2$ emissions intensity also plays an important role. Using a constant global $CO_2$ emissions intensity−a simple average across countries and fuels (fourth bar)−results in a CAF that is 37% smaller in magnitude than our benchmark estimate, which allows $CO_2$ emissions intensities to vary across countries and fuels. This result implies that emissions are higher per unit of energy in countries with larger energy-based adaptive responses to climate change. Our estimates of the CAF are largely invariant across socioeconomic scenarios, but depend heavily on the magnitude of baseline greenhouse gas emissions. Fig. 3c shows point estimates for the 2099 CAF under alternative SSP-RCP scenarios, demonstrating that the CAF under RCP8.5 is roughly double that under RCP4.5, due to greater baseline warming leading to larger energy savings from fewer cold days. Although SSP scenarios change total population and levels of income across countries, which can shape the total energy response to daily temperatures[8], we find that within an RCP, global CAF values differ little across SSP scenarios. This is due to the fact that a majority of variation in projected income and population levels across SSPs[38] occurs outside of countries with high historical emissions, which as we show below comprise the primary source of adaptive behavior driving the CAF.

### Country-level heterogeneity

There is substantial heterogeneity in the magnitude of adaptation-induced $CO_2$ emission changes across countries. Fig. 4a shows both a map and histogram of country-level cumulative adaptation-induced $CO_2$ emissions changes by 2099 under SSP2-RCP8.5 (denoted as $\mathcal{E}_{2099}$ in the "Methods" section). While 85% of countries experience $CO_2$ emissions reductions, of those countries, the 5th and 95th percentiles of cumulative adaptation-induced emission changes by 2099 are −0.027 and -6.24 $GtCO_2$, respectively. For the remaining 15% of countries that experience increases in emissions, magnitudes are small, with the 5th and 95th percentile range estimated at 0.0024 to 0.5 $GtCO_2$. These net changes in emissions can be decomposed into country-level changes in emissions from electricity, which are positive in most countries (Supplementary Fig. 1), and in emissions from other fuels, which are negative in virtually all countries (Supplementary Fig. 2). The case of India is particularly striking, as it exhibits large adaptation-induced declines in $CO_2$ emissions by end-of-century, despite facing substantial increases in exposure to extreme heat in future years[39]. This result is driven partially by the demand responses from ref. 8, which estimate that electricity demand will increase by 4.1 Exajoules (EJs) by 2099, while cumulative demand for other energy will fall by 7.0 EJs, due to electricity-temperature demand responses being relatively flat for much of the world. This result is also influenced by heterogeneous emissions intensities: other fuels in India are substantially more emissions-intensive, implying that demand responses translate into an increase of only 0.9 $GTCO_2$ emissions from electricity compared to a decline of 2.7 $GTCO_2$ from other fuels. This decomposition is detailed in Supplementary Note 4.

### Implications for mitigation policy

Heterogeneity in country-level adaptation-induced $CO_2$ emissions can inform the fairness and stringency of countries' Nationally Determined Contributions (NDC) from the Paris Agreement of the United Nations Framework Convention on Climate Change (UNFCCC). One interpretation of reduced $CO_2$ emissions from energy-based adaptation is the accrual of "free" abatement. Unlike typical $CO_2$ abatement, which

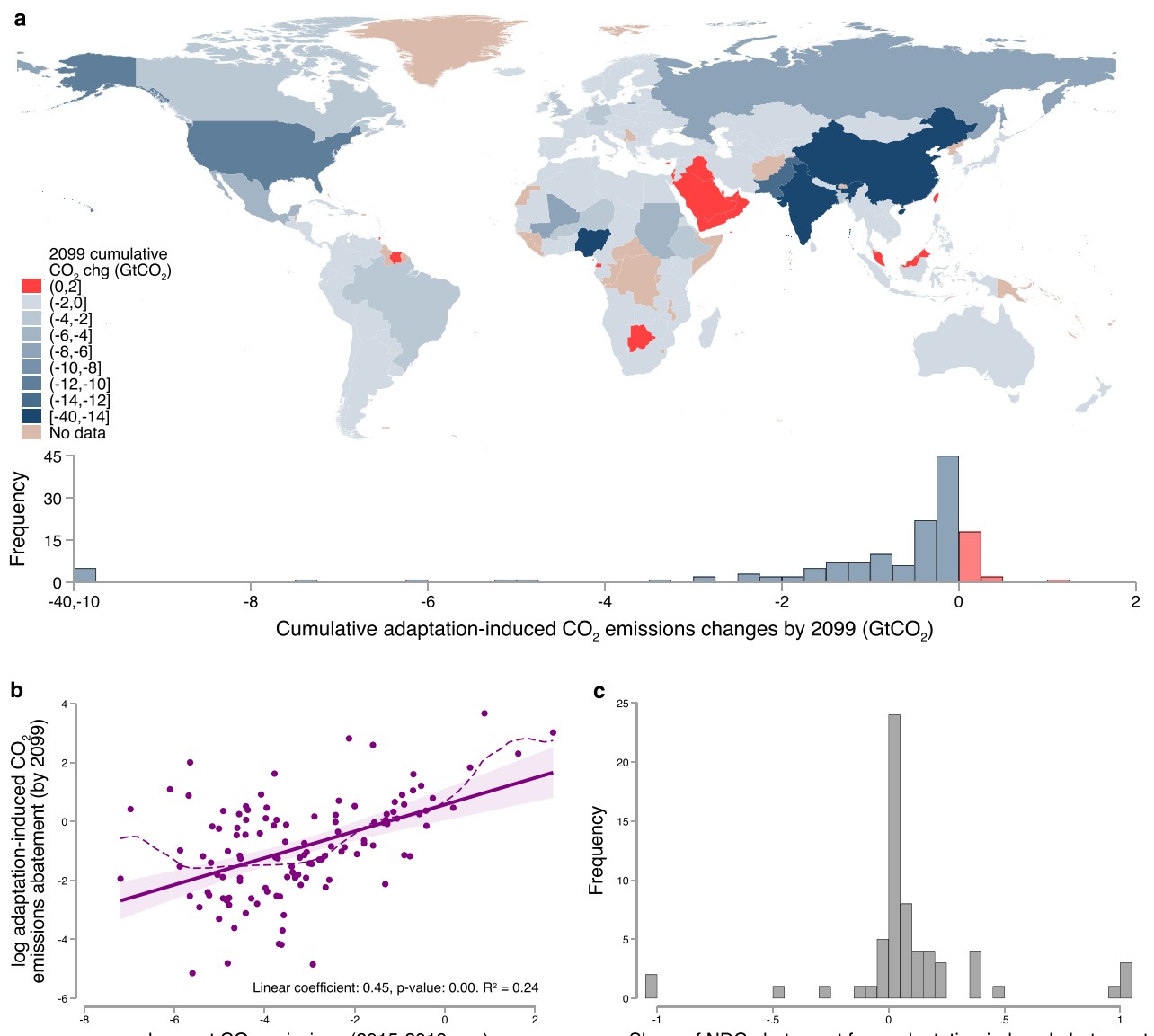

**Fig. 4 | International heterogeneity in adaptation-induced cumulative CO₂ emissions. a** The map (produced using Stata software with the `spmap` package using GADM shapefiles[73]) and histogram display country-level cumulative adaptation-induced CO₂ emissions in 2099 measured in gigatons of carbon dioxide (GtCO₂) calculated using Eq. (3). **b** The plot shows a country-level scatter plot of natural log cumulative adaptation-induced CO₂ emissions reductions by 2099 (y-axis) against natural log of present-day CO₂ emissions (x-axis; emissions averaged between 2015 and 2019). The plot also shows the linear model fit (solid line) with 90% confidence interval (shaded area) and point estimate and p-value of the linear coefficient. It also shows a local polynomial fit (dashed line) using an Epanechnikov kernel with a rule-of-thumb bandwidth[71] ($N = 121$). **c** Histogram shows the distribution of the country-level ratio of cumulative adaptation-induced CO₂ emissions reductions by 2050 to cumulative CO₂ emissions reduction commitments under Nationally Determined Contributions (NDC) taken from ref. 43. A value of 0.5 implies that 50% of NDC commitments are projected to be met by energy-based adaptation alone. **a, b** show results for the scenario of Shared Socioeconomic Pathway 2 paired with Representative Concentration Pathway 8.5 (SSP2-RCP8.5), while panel **c** shows projected abatement for 63 countries under SSP5-RCP8.5 for consistency with ref. 43.

results from climate mitigation policies designed to directly curtail emissions, this abatement emerges solely as a consequence of behavioral adjustments unprompted by mitigation policies. However, the resulting emissions reductions from these adaptations have global benefits identical to those induced by environmental policy and can influence international negotiations and country-level mitigation benchmarks. The magnitude and distribution of this free abatement has two important features relevant for global climate policy.

First, Fig. 4b shows that for the 121 countries projected to experience adaptation-induced CO₂ declines, the magnitude of cumulative abatement by 2099 is strongly correlated with historical emissions. For example, while Eritrea is responsible for less than 0.03

billion tons of CO₂ emissions cumulatively since 1970, we project that it will accrue only 0.14 billion tons of free abatement in the coming century. In contrast, the United States is responsible for more historical emissions than any other nation on earth, but is projected to benefit from over 10 billion tons of adaptation-induced abatement without any mitigation policy. This correlation implies that while the countries responsible for most greenhouse gas emissions to date are often targeted for aggressive mitigation in international climate negotiations[40], they are projected to receive substantially more free CO₂ abatement during the 21st century, relative to today's developing economies. To ensure equity principles are fairly upheld in climate negotiations, these findings suggest that historically large emitters

should increase the stringency of their mitigation targets to appropriately account for the CAF.

Second, we find that the magnitude of adaptation-induced abatement is large relative to existing mitigation targets. To illustrate this, we report each country's cumulative adaptation-induced abatement in 2050 as a fraction of that country's cumulative required abatement in 2050 under its NDC[41-43]. When this ratio takes a value of one, the entirety of a country's obligations under its NDC will be met without any mitigation policy; that is, the projected gap between a country's baseline emissions and its NDC is met entirely by our estimate of adaptation-induced abatement. Fig. 4c shows a histogram of these values, displaying the share of mitigation stipulated under each country's NDC that is realized through adaptation-induced abatement for each of the 63 countries with long-term commitments cataloged by ref. 43. Similar to the the full sample, 82% of the countries in this subsample are projected to experience adaptation-induced abatement. For these 52 countries, adaptation-induced abatement will on average reduce gaps between baseline emissions and their NDCs by 20% in 2050. Several countries are projected to undergo emissions reductions from adaptation that are larger than the mitigation commitments implied by their NDCs, as shown by share values that exceed 1. This highlights that NDC commitments, which already have been shown to be insufficient to meet the Paris Agreement goal of no more than 2 °C increase in global average temperatures relative to pre-industrial[44], are even weaker than previously assessed: once the CAF is accounted for, many countries need to undertake little mitigation policy to achieve their targets.

## Discussion

We develop a framework for quantifying the feedback between energy-based adaptation and anthropogenic climate change, a phenomenon we call the Climate Adaptation Feedback. Our methodology combines high-resolution projections of future energy consumption responses to climate change with country- and energy-specific $CO_2$ intensities to quantify cumulative emissions changes due to adaptation. Under several benchmark pathways for future emissions and socioeconomic development, we consistently find a negative CAF – i.e., that declines in energy use from adapting to fewer days with cold temperatures more than offset increases in energy use from adapting to warmer temperatures. This result is driven in part by the finding that much of the global population is projected to face income constraints that prevent substantial increases in energy demand in response to rising temperatures[8]. Our central estimate implies that adaptive energy use attenuates warming by 0.12 °C in 2099, roughly equivalent to six years of warming at recent rates. This moderation of GMST change between 2020 and 2099 avoids 1.8 trillion in damages (in $2019 USD) in present value terms, an amount equivalent to roughly 2 percent of 2019 global output. When accounting for statistical and climatological uncertainty, our results suggest it is unlikely that the CAF is positive and large in magnitude, limiting concerns that energy-based adaptation will exacerbate future warming.

Our data-driven approach combined with the updated estimates of energy demand responses we use from ref. 8 leads our findings to contradict several existing estimates in the literature, drawn from both reduced-form and computational general equilibrium methods. Prior work projecting energy demand under SSP-RCP scenarios using alternative demand response estimates projects that, in contrast with ref. 8, aggregate energy demand will rise due to adaptation to climate change[15,20]. The other contemporary paper that considers how adaptation affects the trajectory of climate change uses an IAM to do so and, contrary to our study, finds a positive climate feedback from adaptation: ref. 17 estimate that adaptation increases cumulative emissions in 2099 by approximately 347 $GtCO_2$-equivalents, which in our framework corresponds to a CAF of approximately +0.17 °C.

There are numerous differences between prior studies and ours that may contribute to such divergent results (see "Methods"). First, our projections of the energy demand response to climate change stem from ref. 8's approximately 25,000 heterogeneous empirically-derived fuel-specific energy demand responses representing subnational units across the globe. These estimates include demand responses to the full distribution of daily temperature realizations, comprising both moderate and extreme temperatures. In contrast, refs. 20 and 17 use earlier empirical estimates from ref. 15, who estimate two regional fuel-specific demand responses to the annual frequency of average daily low (<12.5 °C) and high (>27.5 °C) temperature extremes. Second, while we directly estimate how climate change affects final energy demand using historically observed behavior, ref. 17 model adaptation as the response of global energy demand to temperature-driven changes in the productivity level of energy inputs in economic production. The energy demand induced by adaptation is then computed as the difference in aggregate energy use between the model where climate change affects productivity levels and an alternative specification where fuel-specific productivity levels are temperature-invariant. Third, we account for country-level heterogeneity in emissions intensities, as opposed to heterogeneity across only 17 global macro-regions[17]. In sum, while ref. 17 compute the CAF associated with energy productivity shocks from climate change under conditions in which regional planners balance the benefits and costs of responding to climate change, our data-driven approach estimates the CAF for a future where agents adapt to climate change in line with historical behavior. While many data inputs also differ, these distinct methodologies render the two approaches highly complementary.

Our analysis has several limitations. First, we define the CAF relative to a fixed SSP for socioeconomic conditions and fixed RCP for baseline global emissions. The advantage of this approach is that SSPs and RCPs are widely used in climate projections and do not already account for emissions arising from energy-based adaptation[45]. The disadvantage is that we cannot readily map the narrative-driven assumptions surrounding global emissions and energy use implied by each SSP-RCP combination into our framework. This leads to an inconsistency between our use of fixed or exponentially-declining emissions factors across all SSP-RCP combinations and the variation in the implied carbon intensity of output induced by the aggregate time series specific to each scenario. However, the baseline coupled SSP-RCP scenarios do not contain sufficient information for us to extract the fuel- and country-specific emissions factors that we would need in order to map the implied variation in decarbonization pathways onto our projections of energy demand. We instead assume that emissions factors stay at measured levels or fall according to current empirically-estimated exponential trends, as these measurements directly reflect the current state of the world and allow us to minimize the normative assumptions we make regarding decarbonization.

While completely remedying this inconsistency would require a fully coupled approach in which behavioral adaptations are built directly into the modeling forming the SSP-RCP scenarios, our key qualitative findings are unlikely to be affected by such a change for two reasons. First, we conduct a sensitivity analysis that captures one dimension of such a coupling; our dynamic version of the CAF shown in Fig. 3 recalculates the energy demand projections from ref. 8 to account for the endogenous response of temperature to prior years' energy demand induced by the CAF itself. Results are virtually unchanged. Second, as long as $CO_2$ intensities associated with non-electricity energy use do not decline dramatically relative to those for electricity, our estimate should serve as an upper bound on the magnitude of the CAF when general equilibrium effects are small. For example, if the electricity sector continues to decarbonize faster than other fuels[46], there will be fewer additional emissions from increased electricity consumption to offset the decreased

emissions from other fuels, implying a more negative CAF than the value we have estimated here. Finally, because adaptation lowers the global net demand for energy services across all fuel sources, our estimate of the CAF would be negative even if all energy services required to meet future adaptive demand were provided by electricity.

Second, relying on a fixed SSP-RCP baseline further omits other general equilibrium channels associated with energy-based climate adaptation. For example, recent integrated assessment models of climate change illustrate the importance of price effects from energy-based adaptation in altering the social cost of carbon[17,20]. While some state-of-the-art, multi-region macroeconomic models do incorporate some heterogeneity in demand for energy across space[17,36,47], these models do not yet capture how adaptation may directly alter how energy enters final consumption worldwide. Incorporating both behavioral responses to climate change through energy use and allowing for these changes to affect prices, expectations, and investment will be essential moving forward in establishing a unified modeling framework whereby socioeconomic conditions and emissions pathways interact dynamically.

Third, our analysis is somewhat limited in its coverage of energy-based adaptations. We omit possible feedbacks arising from transportation-based adaptation due to a paucity of empirical estimates for how transportation-related GHG emissions respond to a warming climate (e.g., ref. 48 provide such estimates, but only for the United States). Our estimates may also fail to account for long-run changes in the response of energy demand driven by shifts in preferences or technological change, leading us to not fully capture some facets of the extensive margin for adaptation[9,16,23]. If the price of electricity falls drastically in the future relative to that of other energy, we may underestimate electricity demand responses which would bias our CAF estimate downward. Conversely, if end-uses for electricity become more efficient in a way previously unexplained by income growth, our CAF will overstate future emissions from electricity. More generally, our estimates of the CAF will not capture future energy demand responses to climate change driven by factors that are not captured by the two-factor model in ref. 8. This omits the potential for government policy or technological breakthroughs to cause structural shifts in how energy use responds to temperature changes. Additionally, our calculation includes all direct energy consumption responses to daily variations in temperature, but omits any indirect effects on energy demand that may arise under climate change. For example, declining agricultural yields due to climate change may induce more fertilizer use, which could alter GHG emissions from the agriculture sector even if it uses the same level of direct energy inputs[49,50]. Such indirect channels of energy-based adaptation have, to our knowledge, not been systematically quantified globally. When these adaptive behaviors are better characterized in the scientific literature, they too may be incorporated into the CAF using the framework developed here.

Even within the non-transportation energy sector, we face two primary data limitations. First, our measure of $CO_2$ emissions intensity corresponds to a country's average emissions intensity, whereas a more appropriate measure would be the $CO_2$ intensities of marginal energy sources that will experience increasing (or decreasing) demand due to variation in local temperatures. Unfortunately, the data for calculating marginal $CO_2$ intensities for every country is not readily available, nor is it clear whether the average $CO_2$ intensities we use systematically under- or overstate true marginal intensities. Second, the absence of non-$CO_2$ GHG emissions intensities prevent us from directly quantifying corresponding changes in non-$CO_2$ emissions due to adaptation. While our empirical estimate of the relationship between GMST changes and global cumulative $CO_2$ emissions implicitly includes non-$CO_2$ GHG emissions (see Fig. 2d and Methods), our analysis does not capture any geographical heterogeneity in the covariance of these emissions and $CO_2$. For example, because phase-out rates of hydrofluorocarbons (HFCs) vary by country under the Kigali Amendments to the Montreal Protocol, some countries may see decreased $CO_2$ emissions from air conditioning coincide with declines in HFC emissions larger than those captured in our estimated global temperature response relationship. Lastly, by definition, the CAF only quantifies the GMST consequences due to GHG emissions caused by energy-based adaptation. In practice, fossil energy consumption for heating and cooling leads to additional local ambient air pollution from power plants and the direct combustion of fossil fuels (e.g., natural gas furnaces). In the case of electricity generation, those local ambient air pollutants (e.g., PM2.5, $SO_2$, and $NO_x$) have been shown to have large effects on human health outcomes[51–53]. Therefore, the declines in energy consumption due to adaptation that we study here may lead to additional local environmental benefits not considered in this analysis.

Our finding that energy-based adaptation may lower global $CO_2$ emissions has implications for countries' existing mitigation commitments. Projections of "business-as-usual" $CO_2$ emissions that fail to account for declining energy use on net due to adaptive behaviors (that occur regardless of policy changes) may lead to an inaccurate measure of policy stringency. As we show by comparing cumulative adaptation-induced $CO_2$ abatement with mitigation commitments under existing NDCs, energy-based adaptation alone may account for a substantial share of NDC abatement for many countries. For a more accurate measure of climate policy stringency, measures of business-as-usual or baseline emissions must incorporate GHG emissions changes due to adaptive behaviors.

More broadly, a non-zero CAF highlights the inherent link between climate mitigation and adaptation that is beginning to be employed in policy and research[23]. Advocates and policymakers have long argued that mitigation and adaptation should be considered separately, in part to isolate the objectives within each domain. With a non-zero CAF, those objectives are inherently linked; mitigation goals must take into account the consequences of adaptive behavior, and climate adaptation must be viewed as an additional channel for mitigation. Our results further emphasize the importance of interdisciplinary research quantifying the future effects of climate change. With both Earth System Models and Integrated Assessment Models increasing in complexity, coupling projections of the climate system with the dynamic responses of human behavior is critical in order to appropriately inform each class of models. Our finding suggests that adjusting existing models to allow for this interaction will play an important role in forming more accurate projections and prescriptions of the human response to anthropogenic climate change going forward.

## Methods
### Constructing the climate adaptation feedback
Our paper develops and implements a methodology to quantify the extent to which changes in future energy use driven by adaptation to anthropogenic climate change (ACC) will alter greenhouse gas (GHG) emissions and in turn affect climate change. We call the difference between global mean surface temperature (GMST) with versus without adaptation-induced energy consumption at time horizon $\tau$ the "Climate Adaptation Feedback", or $CAF_\tau$. $CAF_\tau$ depends on the baseline scenario of emissions and socioeconomic conditions, defined for our purposes as a combination of a Representative Concentration Pathway (RCP) of global anthropogenic GHG emissions[54–58] and a Shared Socioeconomic Pathway (SSP) of projected national populations, incomes, and other socioeconomic characteristics[59–61]. The following calculations fix our baseline SSP2-RCP8.5 scenario to avoid notational clutter, but we repeat the processes below for each SSP-RCP we consider (results for all scenarios are displayed in Fig. 3).

Consider two projections of future warming, one that accounts for adaptive changes in energy use and one that does not. Denote projected GMST in period $t$ as $\overline{T}_t^A$ when adaptive energy use is accounted for, and as $\overline{T}_t^N$ when it is not. Normalize time periods such that $t = 0$ is the year 2020 and let $\Delta$ denote the time difference operator between period $t = 0$ and $t = \tau$. With this notation, we define the CAF at horizon $\tau$ as Eq. (1):

$$CAF_\tau \stackrel{def}{=} \Delta \overline{T}_\tau^A - \Delta \overline{T}_\tau^N. \qquad (1)$$

Equation (1) is the difference in GMST change at horizon $\tau$ due to adaptive changes in energy use around the world. When the CAF is positive, adaptation exacerbates warming globally. When the CAF is negative, adaptation dampens warming.

To construct $CAF_\tau$, we first calculate the change in global $CO_2$ emissions due to adaptation-induced energy use in each period through horizon $\tau$. At the local scale, emissions from adaptive energy use depend on how ACC changes local temperature distributions as well as how different temperature realizations affect energy demand (see Fig. 1). Because local temperature changes, energy use responses, and the $CO_2$ intensity of energy consumption vary substantially across space, we conduct this step at the country level before aggregating globally to compute the global CAF. Specifically, for each year $t$ and country $i$, the $CO_2$ emissions generated by energy-based adaptation are given by Eq. (2):

$$E_{i,t} = \sum_h F_i^h \Delta J_{i,t}^h = \sum_h F_i^h \sum_{p \in i} \underbrace{[J^h(\boldsymbol{T}_{p,t}^N, \boldsymbol{X}_{p,t}) - J^h(\boldsymbol{T}_{p,0}^N, \boldsymbol{X}_{p,t})]}_{\substack{\text{Change in energy use due to} \\ \text{adaptation to temperature change}}} \qquad (2)$$

where $h$ indicates final consumption of either electricity or an aggregate of consumption across all other fuels, including natural gas, oil shale and oil sands, biofuels, and others, as detailed in ref. 8, the source of our energy use projections. In this expression, $p$ indicates one of ~ 25,000 global subnational regions with approximately internally-homogeneous historical temperatures, which are shown as Eq. (2) defined by ref. 8. Each $F_i^h$ is the $CO_2$ emissions factor for a given energy type $h \in \{electricity, other\ fuels\}$ for country $i$, measured in units of in $tCO_2 \times GJ^{-1}$ of fuel consumed. Construction of $F_i^h$ is detailed below. We fix $F_i^h$ to the observed 2010-2018 averages for each country-fuel pair to avoid projecting future changes in $CO_2$ intensities while reflecting current differences in countries' energy mixes.

The underbraced object in Eq. (2) represents the impact of climate change on adaptation-induced total energy use in region $p$. It is defined as the difference in total energy use in GJ between a future climate affected by ACC and a future with stable temperatures representative of the current ($t = 0$) climate. This projected change in energy consumption depends critically on the "dose-response" functions $J^h(\cdot)$, which are constructed by ref. 8 using historical energy consumption data and standard climate econometric tools. These functions relate energy consumption in each fuel category $h$ to daily temperature, capturing the energy consumption that results from all behaviors and investments that individuals and firms undertake in response to local temperature variation across all sectors besides transportation.

As detailed in ref. 8, these energy demand response functions depend primarily on the realization of future daily temperatures within a given impact region, denoted by the vector $\boldsymbol{T}_{p,t}^N$, under a given RCP scenario. The dose-response functions also include higher-order terms of daily grid cell-level temperature realization along with a set of covariates summarized by $\boldsymbol{X}_{p,t}$, which include projections of GDP per capita and population specific to a SSP scenario and long-run averages of cooling and heating degree days under each temperature trajectory. These covariates allow for the response of energy consumption to daily temperature realizations to vary based on how the economic resources and climatology of a given location change in the future.

This ensures the CAF calculation captures the extensive margin of adaptation that takes place over long periods and may be especially important in developing economies, where income growth is likely to lead to substantial increases in cooling and heating technology adoption[4,8,9]. The empirical estimation of such extensive margin effects is conducted in ref. 8 by interacting short-run variation in weather with long-run variation in income and climate in a country-by-year panel regression. This model results in the estimation of heterogeneous dose-response functions in which energy demand sensitivity to daily realizations of temperature differs based on long-run average income and climate. The econometric procedure employed follows previously-developed methods (e.g., ref. 62) to recover nonlinear energy demand responses to temperature at a grid cell-by-day level, which can be applied to predict climate change impacts at the scale of ~ 25,000 regions, even though more aggregated outcome data were employed (see Methods of ref. 8 for details). More generally, these interaction terms allow for the model to identify the heterogeneity in the response of local energy demand to temperature realizations caused by variation in income and climate over space and time. We show the quantitative importance of the extensive margin in shaping the CAF in Supplementary Note 2.

Two sources of uncertainty enter into Eq. (2). The first is climate model uncertainty: for a given emissions scenario, there is uncertainty over future local temperature realizations $\boldsymbol{T}_{p,t}^N$. We account for this uncertainty by utilizing all 33 Global Climate Model (GCM) projections included in the Surrogate Model Mixture Ensemble (SMME) employed by ref. 8 and built from the Coupled Model Intercomparison Project Phase 5 (CMIP5)[63] climate models. The second is statistical uncertainty in the empirical estimates of the energy-temperature dose-response functions $J^h(\cdot)$; this uncertainty is captured by ref. 8 through application of the statistical Delta Method, creating a Gaussian distribution of predicted impacts for each of the 33 climate model projections. To combine both sources of uncertainties, we follow ref. 8 in constructing the mixture distribution of these 33 Gaussian distributions using Newton's method.

There are multiple possible sources of empirically-based energy-temperature demand responses that we could have used to estimate Eq. (2). In selecting a source, we sought to identify estimates that: (i) account for demand responses to changes across the entire temperature distribution, not just extreme temperatures; (ii) are differentiated between electricity and other fuels, allowing for a different effect of temperature on the demand for both types of energy; (iii) incorporate extensive margin responses of energy demand to changes in temperature over time; and (iv) allow for different localities to have different responses to temperature change. While other multi-country energy-temperature demand estimates exist (e.g., ref. 15), to our knowledge ref. 8 represents the only study meeting all four of these criteria.

Summing the results from Eq. (2) over time and across space, we write the cumulative change in global $CO_2$ emissions between years 0 and $\tau$ caused by adaptation-induced energy use as Eq. (3):

$$\mathcal{E}_\tau = \sum_{t=0}^{\tau} \sum_i E_{i,t} \qquad (3)$$

To convert cumulative emissions from adaptation, $\mathcal{E}_\tau$, to changes in future GMST, we estimate a relationship between projected future emissions and warming in the absence of adaptation using the forecasts generated by the 33 GCM projections described above. Below we discuss this relationship in detail, how it relates with the transient climate response to cumulative carbon emissions (TCRE) in the climate science literature, and we show that it is well-approximated by a linear coefficient. We denote this linear relationship between emissions and GMST with the slope coefficient $\beta$. This implies that the GMST change between years 0 and $\tau$ due to adaptation is: $\Delta T_\tau^A = \Delta T_\tau^N + \beta \mathcal{E}_\tau$. Rewriting

this expression in terms of the climate adaptation feedback definition in Eq. (1) gives us Eq. (4):

$$CAF_\tau = \beta \mathcal{E}_\tau. \qquad (4)$$

For each SSP-RCP combination, we obtain a point estimate for $CAF_\tau$ from Eq. (4), as well as a 90% confidence interval that account for both climate uncertainty across GCMs and statistical uncertainty in the energy response functions, as discussed above.

## Valuation of the climate adaptation feedback

To convert our estimates of the CAF into dollar value of avoided damages, we use the Climate Impact Lab's Data-driven Spatial Climate Impact Model (DSCIM)[29]. This model includes climate change damages to mortality, coastal storms and sea level rise, labor, energy, and agriculture. Mortality risk is monetized using the U.S. EPA VSL with a value of life years lost adjustment and an income elasticity of one, following ref. 29. DSCIM assigns a monetary value to the damages from global warming in every year along a baseline socioeconomic and climatic trajectory. Avoided damages due to the negative CAF are calculated as the difference between predicted damages in the baseline scenario and in the scenario inclusive of the CAF, for each year between 2020 and 2099. When discounting damages, the DSCIM model generates a stochastic discount factor (SDF) for all future periods based on the Ramsey rule calculated along the exogenous consumption pathway for a given socioeconomic scenario, and after incorporating consumption losses from baseline warming. Specifically, we use a coefficient of relative risk aversion equal to $\eta = 2$, and we discount future values using Ramsey discounting with $\eta = 2$ and a pure rate of time preference of $\rho = 0.0001$. We convert the avoided consumption losses due to the CAF into a present value in 2019 equivalents using this SDF.

## Comparison with the integrated assessment model approach

Unlike prior literature, we do not use an Integrated Assessment Model (IAM) to estimate the effects of adaptive energy demand on global temperatures. This abstracts from fully specifying a demand system wherein agents balance the benefits of using energy for adaptation against the costs of foregoing income for other forms of consumption. Our data-driven approach avoids making structural assumptions governing the channels through which adaptation to climate change occurs, which an IAM framework usually imposes. This abstention from using an IAM of course poses a tradeoff in that our empirically-based CAF omits general equilibrium channels that may affect energy demand, as highlighted in the Discussion section.

The benchmark study to date examining the potential for feedback between energy used for adaptation and global temperatures[17] uses the WITCH IAM[18] framework. WITCH is a multi-region IAM based on an augmented Ramsey model of optimal savings designed to assess the welfare and temperature effects of climate policy in a dynamic setting[18]. Agents in the model consist of regional planners who select sequences of consumption, savings, and energy use in their local economy so as to maximize residents' welfare[18]. The model allows for both a non-cooperative formulation, where up to 17 regional planners maximize the welfare of their populations in a global extensive game, as well as a cooperative formulation under which a single social planner maximizes welfare across all regions globally. Reference 17 use the non-cooperative formulation of the model as their laboratory for examining the effects of adaptation on energy demand and temperatures. While the non-cooperative formulation is the closest version of the model to a competitive equilibrium, unlike in a decentralized economy, regional planners in the non-cooperative formulation still internalize some climate externalities. These planners choose mitigation and/or abatement efforts to maximize regional welfare; the Nash

equilibrium in the non-cooperative formulation of WITCH is a second-best optimal solution, as opposed to a model of outcomes under laissez-faire[18].

To use the WITCH model to estimate the effects of adaptation on energy use, ref. 17 modify the model to allow climate change to affect the productivity of different fuels used as intermediate inputs for producing the modeled final consumption good. Adaptation inside of this alternate specification is taken to be how energy use changes in response to the productivity shocks driven by future climate change[17]. The authors then calculate aggregate demand in the alternative (baseline) model specifications that include (exclude) the adaptive energy demand induced by the productivity shocks for three scenarios: no climate policy, a 2.5 °C limit on warming, and a < 2 °C limit on warming. For each scenario, ref. 17 solve both specifications of the model in the non-cooperative mode and raise carbon taxes until the resulting Nash equilibrium yields a level of aggregate emissions that coincides with the policy goal in the scenario. Reference 17 quantify the additional energy demand (and in turn, emissions) induced by adaptation as the difference between aggregate energy demand in the alternative specification with adaptation and a baseline version of the WITCH model where adaptation is absent and energy-specific productivity levels are temperature-invariant.

In contrast, our data-driven approach is agnostic regarding the underlying structure of the economy, relying instead on empirical estimates of the historical response of energy demand to temperature fluctuations. Our methodology trades the structure of an IAM for increased spatial granularity and a greater ability to account for both statistical uncertainty in future demand responses as well as model uncertainty in how future temperature dynamics will respond to emissions. To do so, we use existing forecasts from ref. 8 that predict how future climate change will affect energy demand based on realized historical adaptation to intra-annual temperature changes. This avoids measuring adaptation as the change in energy used by regional social planners or assuming that the shifts in demand are driven by productivity shocks and/or changes in relative prices. At the same time, the lack of structure precludes us from accounting for future government policy or voluntary actions that may alter how energy demand responds to climate change beyond historical patterns. In that sense, we sacrifice the ability to estimate what the CAF would be in a world where global policy governing emissions or carbon prices change (cf. ref. 17).

## Data

We obtain point estimates and 90% confidence intervals for projections of adaptation-induced energy use (i.e., the underbraced terms in Eq. (2)) at the country-year-fuel level for two emissions scenarios and four socioeconomic scenarios directly from ref. 8.

We obtain projection-specific annual series of GMST between 2020 and 2099 under the RCP4.5 and RCP8.5 pathways directly from ref. 8. The SMME employed by ref. 8 generates 33 projections of annual GMST under each RCP scenario. These global averages correspond directly with the impact-region specific daily temperature realizations that drive future $\Delta f_{i,t}^h$ s under each model run in ref. 8.

We obtain five-year country-level GDP per capita and population projections for the 2020–2099 period under each SSP scenario from the International Institute for Applied Systems Analysis (IIASA) model[59–61] and from the Organization for Economic Co-operation and Development (OECD) Env-Growth model. For projections under each SSP scenario, we take the average between these two model outputs.

To convert adaptation-induced final energy consumption of energy source $h \in \{electricity, other\ fuels\}$ to $CO_2$ emissions, we need energy source-specific $CO_2$ emissions intensities that account for heterogeneity in the mix of primary fuels (e.g, coal, natural gas, and renewables) in each country. For example, electricity in Poland is

mostly generated using coal, while in Costa Rica it comes almost exclusively from renewables, each with very different resulting $CO_2$ emissions intensities.

For each country $i$ and year $t$, let $r \in \mathcal{H}_{i,t}^h$ index the primary fuels used to generate final consumption of energy source $h$. The final energy source $CO_2$ emissions intensity, $F_i^h$, is the weighted average of primary fuel $CO_2$ emissions intensities, $f_{i,r,t}^h$, across primary fuels $r$ used to produce final energy source $h$, where each weight, $\omega_{i,r,t}^h$, is the total amount of energy fuel $r$ contributes to final consumption of energy type $h$ in year $t$. To account for year-to-year fluctuations in primary energy use, we take this average over 2010–2018 values. Country-level final energy source $CO_2$ emissions intensities are calculated as Eq. (5):

$$F_i^h = \frac{\sum_{t=2010}^{2018} \sum_{r \in \mathcal{H}_{i,t}^h} f_{i,r,t}^h \, \omega_{i,r,t}^h}{\sum_{t=2010}^{2018} \sum_{r \in \mathcal{H}_{i,t}^h} \omega_{i,r,t}^h}. \tag{5}$$

We obtain primary fuel $CO_2$ emissions intensities $f_{i,r,t}^h$ from the International Energy Agency (IEA) Emissions Intensities Report for each form of final use[64]. We assign weights, $\omega_{i,r,t}^h$, based on consumption data from the IEA World Energy Balances (WEB)[65]. The WEB catalogs country-level primary fuel consumption at the sector level which we aggregate to form our final energy use sectors. Electricity is one such sector (i.e., code ELOUTPUT). For other fuels, we follow ref. 8 for consistency and pool together the industrial, residential, commercial and public services, agricultural, fishing, and other sectors not elsewhere specified (i.e., codes TOTIND, RESIDENT, COMMPUB, AGRICUL, FISHING, and ONONSPEC respectively).

To construct the globally constant emissions-weighted average $CO_2$ intensity across final energy sources and countries used in Fig. 3b, we compute Eq. (6):

$$\overline{F} = \frac{\sum_{i=1}^n E_i^{2019} \sum_h F_i^h}{\sum_{i=1}^n E_i^{2019}} \tag{6}$$

where the weights $E_i^{2019}$ are set equal to 2019 GHG emissions measured in $CO_2$-equivalents from ref. 66.

We obtain country-level baseline $CO_2$ emissions pathways and Nationally Determined Contributions (NDCs) from ref. 67. Reference 67 provide two sets of NDCs for most countries: a more stringent "conditional" NDC path (in the sense that the pathway is conditioned on action by other countries) and a "unconditional" NDC path, both available only under SSP1 and SSP5. We use the more stringent conditional NDCs along with baseline $CO_2$ emissions projections under the SSP5 scenario, both for the year 2050, to construct the ratio of cumulative adaptation-induced $CO_2$ emissions reduction over cumulative $CO_2$ emissions reduction under NDCs by 2050 in Fig. 4c.

## Estimating the GMST-cumulative $CO_2$ relationship
A key challenge to quantifying the CAF is that available emissions intensity data only apply to $CO_2$ emissions, while energy-based adaptation is likely to feed back into climate change via other greenhouse gases as well. Specifically, the IEA data detailed above does not contain emissions intensities for non-$CO_2$ greenhouse gas emissions from final consumption outside of the electricity and heat and power sectors. To address this data limitation, we construct an empirical relationship between GMST and cumulative $CO_2$ emissions that includes any changes in non-$CO_2$ emissions which covary with $CO_2$ emissions. Such a relationship is similar to, but not the same as, the transient climate response to cumulative emissions of $CO_2$ (TCRE), which is the direct (causal) effect of cumulative $CO_2$ emissions on GMST change and has been documented in the climate science literature and shown to be well-approximated by a linear relationship[12,68–70]. However, our empirical relationship additionally includes the effects of non-$CO_2$

greenhouse gases on GMST, to the extent that these gases correlate with $CO_2$ emissions in historical data.

To illustrate this approach, suppose the change in GMST over time horizon $\tau$, $\Delta \overline{T}_\tau$, responds to cumulative $CO_2$ emissions, $\mathcal{E}_\tau^{CO2}$ and cumulative emissions of another GHG, $\mathcal{E}_\tau^{other}$, as given by Eq. (7):

$$\Delta \overline{T}_\tau = \rho \mathcal{E}_\tau^{CO2} + \alpha \mathcal{E}_\tau^{other} \tag{7}$$

In this expression, $\rho$ is the TCRE—the direct effect of changing $CO_2$ emissions on GMST holding cumulative emissions of all other GHGs constant. However, due to IEA data limitations, we cannot estimate projected changes in $\mathcal{E}_\tau^{other}$ due to adaptive energy use. Instead, we can estimate the same regression omitting the effects of other GHG emissions in Eq. (8):

$$\Delta \overline{T}_\tau = \beta \mathcal{E}_\tau^{CO2} + error_\tau \tag{8}$$

Since future emissions of $CO_2$ and other GHGs are likely to be positively correlated, $\beta$ can be expressed as Eq. (9):

$$\beta = \rho + \alpha \frac{cov\left(\mathcal{E}_\tau^{CO2}, \mathcal{E}_\tau^{other}\right)}{var(\mathcal{E}_\tau^{CO2})} > \rho \tag{9}$$

The coefficient $\beta$ is therefore our object of interest; it is the projected GMST change from an observed increase in cumulative carbon emissions. This coefficient combines the direct effect of a unit increase in cumulative $CO_2$ emissions and the indirect effect that accounts for the covariance between $CO_2$ and the other GHG emissions that are inputs into the SMME used to forecast future temperature pathways. When that covariance is positively correlated, one would expect $\beta$ to exceed the TCRE, or $\rho$.

In practice, to estimate $\beta$, we use variation in GMST and cumulative $CO_2$ emissions (in the absence of adaptation) across RCP4.5 and RCP8.5 and the 33 GCM predictions drawn from the SMME based on CMIP5. Letting $s$ index the 66 RCP-GCM combinations, we estimate Eq. (10):

$$\Delta \overline{T}_{\tau,s}^N = \beta \mathcal{E}_{\tau,s}^N + error_{\tau,s} \tag{10}$$

using the temperature and emissions time series generated by the ensemble of models in CMIP5 ($N = 5, 200$). Our estimate of $\beta$ from Eq. (10) is $2.2 \times 10^{-3}\,°C \times GTC^{-1}$ ($p(|T| > |t|) < 0.01$). To examine whether our linearity assumption is valid, Fig. 2d shows a scatter plot between $\Delta T_{\tau,s}^N$ and $\mathcal{E}_{\tau,s}^N$ along with a flexible relationship estimated using a local polynomial function with an Epanechnikov kernel and a rule-of-thumb bandwidth[71,72] that reveals any data-driven nonlinearities. We do not detect any nonlinearities. As a point of comparison, our estimate for $\beta$ is 1.4 times the median estimate for the TCRE (or $\rho$) detected in the literature, although well within confidence intervals for the TCRE[70]. This is consistent with cumulative emissions of $CO_2$ and other GHGs being positively correlated.

## A dynamic climate adaptation feedback
As discussed in the main text, our baseline estimate for the CAF takes projected changes in emissions from adaptation as given by the calculations in ref. 8. That is, we assume that the additional climate change due to the CAF does not itself lead to additional adaptive energy demand. In doing so, the estimated CAF outlined above implicitly assumes adaptive changes in emissions have no concurrent effects on the GMST pathway that determines future adaptation. This is a strong assumption if emissions changes due to adaptation have large immediate effects on the GMST path each year after they enter the atmosphere. In this section, we develop a dynamic version of the CAF and show that, in practice,

the result is nearly identical to the approximated value we center in our analysis.

Specifically, we account for the dynamics of energy-based adaptation by updating the original projections of $E_{i,t}$ from Eq. (2) at each time horizon to account for how historical adaptive emissions through time $t-1$ will have affected the GMST pathway that year. This is implemented by iteratively updating the temperature pathway at each horizon relative to a given RCP baseline to account for temperature change due to the CAF, and then using this distance from the baseline to adjust projected adaptive energy use from that point forward. Starting in 2021 (the second projection year from the[8] data we use), we adjust the baseline projected temperature pathway each year to account for the cumulative effects of emissions from adaptation. We then update each country-year-fuel emissions tuple in that year to account for the adjusted GMST pathway. We repeat this procedure out to 2099 and recalculate the cumulative emissions changes to form an estimate of the CAF that accounts for concurrent dynamics between adaptation use and GMST change.

Specifically, we begin with the set of 66 projected time series of emissions changes computed under our baseline SSP2-RCP8.5 scenario from Eq. (2). These forecasts are country-level changes in emissions for each climate model $m$ given by Eq. (11):

$$E_{i,t,m}^h = F_i^h \sum_{p \in i} \left[ J^h(\boldsymbol{T}_{p,m,t}^N, \boldsymbol{X}_{p,t}) - J^h(\boldsymbol{T}_{p,m,0}^N, \boldsymbol{X}_{p,t}) \right] \quad (11)$$

in year $t$ for fuel $h$ in country $i$ under climate model $m$. We use these country-level sets of projected horizon-$t$ emissions changes to estimate, for each country-fuel-year combination, the reduced form response function in Eq. (12):

$$E_{i,t,m}^h = \alpha_{i,t}^h \Delta GMST_{t,m} + \gamma_{i,t}^h \Delta GMST_{t,m}^2 + \varepsilon_{i,t,m}^h \quad (12)$$

Equation (12) captures the additional emissions due to adaptation for each $i$, $t$, and $h$, as a quadratic function of changes in GMST. We estimate Eq. (12) separately for each $(i, t, h)$ tuple under the SSP2-RCP8.5 combination to form a time series of estimated $\hat{\alpha}_{i,t}^h$ and $\hat{\gamma}_{i,t}^h$ coefficients for each country-fuel combination. From these estimates we construct time-country-fuel impulse response functions (IRFs): each IRF (for an $i$, $t$, $h$ pair) gives the estimated additional change in emissions from energy-based adaptation induced by marginal changes in projected GMST derived from fuel $h$ and country $i$ at the time horizon $t$. This object is the derivative of Eq. (12) with respect to $\Delta GMST$, shown in Eq. (13):

$$\hat{\Theta}_{i,t}^h \overset{def}{=} \frac{\widehat{\partial E_{i,t}^h}}{\partial \Delta GMST_t} = \hat{\alpha}_{i,t}^h + 2\hat{\gamma}_{i,t}^h \Delta GMST_t \quad (13)$$

These impulse responses give, by fuel-horizon-country, the local effects on concurrent emissions from adaptive energy use due to additional warming. We use this to project how prior temperature changes from adaptation will affect contemporary adaptive energy use relative to the baseline pathway. Starting in 2021, we update the values of emissions from energy-based adaptation using a first-order Taylor expansion around their baseline levels. For each country-fuel-year, we define an updated series of emissions using Eq. (14):

$$\widetilde{E_{i,t}^h} = \underbrace{\bar{E}_{i,t}^h}_{\text{baseline emissions}} + \left( \underbrace{\hat{\Theta}_{i,t}^h}_{\text{response functions}} \times \underbrace{\hat{\beta} \widetilde{\Delta \mathcal{E}_{t-1}}}_{\text{horizon } \tau-1 \text{ CAF}} \right) \quad (14)$$
$$\underbrace{\phantom{xxxxxxxxxxxxxxxxxxxxxx}}_{\text{dynamic effect}}$$

where $\widetilde{\Delta \mathcal{E}}$ is the cumulative emissions change due to energy-based adaptation between 2021 and year $t-1$ accounting for the dynamic effects of adaptation as defined in Eq. (15):

$$\widetilde{\Delta \mathcal{E}_{t-1}} = \sum_{s=2021}^{t-1} \sum_i \sum_h \widetilde{E_{i,s}^h}, \quad (15)$$

and $\hat{\beta}$ is our mapping between emissions and temperature as described in Eq. (10) in Methods. Starting in 2021, we calculate the cumulative change in emissions due to adaptation while accounting for how adaptive emissions in turn affect future use for adaptation through the GMST channel. The first term in Eq. (14) captures baseline emissions from adaptation as calculated in Eq. (2) and used in the main text. The second term accounts for how the effects of past emissions from adaptation on the pathway of GMST in turn affect emissions in that year.

We iterate over the process in Eq. (14) over all country-fuel pairs at each horizon through 2099, updating the series for cumulative emissions and temperature each period. The iterative procedure ensures the baseline temperature path (under RCP8.5) is adjusted to reflect the cumulative level of adaptive emissions changes each year. We then recalculate the CAF with concurrent dynamics using the updated energy changes due to adaptation through year 2099. We track the dynamic CAF each year using Eq. (16):

$$CAF_\tau^{dyn} \overset{def}{=} \hat{\beta} \sum_i \sum_h \sum_t^\tau \widetilde{E_{i,t}^h}. \quad (16)$$

The dashed line in Fig. 3a displays the time series of our baseline $CAF_\tau$ without dynamic emissions-temperature linkages and the dynamic version $CAF_\tau^{dyn}$ incorporating concurrent emissions-GMST linkages over the 2021-2099 period under the baseline SSP2-RCP8.5 scenario.

The concurrent emissions-GMST linkages channel decreases cumulative emissions and temperature changes (in magnitude) attributable to adaptation. This occurs because of the negative nature of the CAF: lower emissions from declines in heating use outweigh additional emissions from increased cooling at each horizon. This in turn lowers cumulative emissions relative to baseline each year and with it our forecast for $\Delta GMST$. These lower temperature levels in turn result in smaller future changes in emissions from adaptation through the IRFs, as declines in temperature lead to smaller declines in consumption of other fuels and smaller increases in emissions from electricity. This is indicated by an increasing gap between $CAF_\tau$ and $CAF_\tau^{dyn}$ over time; the larger the cumulative negative effect from the CAF, the lower the updated temperature series is relative to baseline. This further decreases emissions changes from adaptation (in magnitude) each year, further lowering the temperature. However, empirically, we find that this gap between $CAF_\tau$ and $CAF_\tau^{dyn}$ is small in magnitude. By 2099, $CAF_{2099}$ is −0.1206 while $CAF_{2099}^{dyn}$ is −0.1196. Given the minor consequence played by such dynamic emissions-GMST linkages, in our main text we emphasize $CAF_\tau$ over $CAF_\tau^{dyn}$.

## Reporting summary

Further information on research design is available in the Nature Portfolio Reporting Summary linked to this article.

## Data availability

Emissions intensities and annual consumption at the country-by-fuel level are taken from the International Energy Agency's Emissions Intensities Report and World Energy Balance datasets, respectively. The raw IEA data are a subscription-only product that cannot be publicly shared. All other raw and processed data used in the main manuscript and supplementary information required for a bottom-up replication of all analysis are publicly available in the Zenodo repository associated with the paper at https://doi.org/10.5281/zenodo.10476310.

## Code availability

A permanent reference to the specific version of the computer code used to produce the analysis presented in this manuscript is archived at https://doi.org/10.5281/zenodo.14894953. The active repository for the paper is publicly available on GitHub at https://github.com/xabajian/ACDM_Climate_Adaptation_Feedback.

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

## Acknowledgements
We thank Max Auffhammer, Chris Costello, Ranjit Deshmukh, Kelsey Jack, Michael Oppenheimer, Andrew Plantinga, Ashwin Rode, Peter Sherman, and participants at the Bren School of Environmental Science & Management (UCSB) and Environmental Markets Lab (UCSB) seminars for helpful comments. This research was supported by funding from the Environmental Defense Fund.

## Author contributions
A.C.A., T.C., K.C.M., and O.D. conceived of the project. A.C.A., T.C., and K.C.M. conducted the analysis. A.C.A., T.C., K.C.M., and O.D. wrote the paper.

## Competing interests
The authors declare no competing interests.
