## [Transparent Peer Review file · Nature Communications]

Quantifying the global climate feedback from energy-based adaptation

Corresponding Author: Dr Tamma Carleton

Version 0:

Reviewer comments:

Reviewer #1

(Remarks to the Author)

This work provides a framework for quantifying the implications of energy used for adaptation on global GHG emissions and mitigation goals.

This is a timely and relevant work, that focuses on a channel of anthropogenic emissions often understudied. The paper is also concise and well written.

The methodology is well described, and robust as for the method directly proposed in this work for the computation of the CAF. The only important remark I have concerns the possible limitations of one key input used to compute the CAF: the energy demand elasticities, taken from Rode et al., 2021. My concern derives from the following points:

- 1) the sectoral aggregation of energy demand performed in Rode et al., 2021 may mask several different responses of opposite sign that add up to zero in the estimated elasticities, eg consumption of fossil fuels by buildings for heating and by the industrial sector for cooling (as documented eg in De Cian and Sue Wing, 2019; Schleypen et al., 2019).
- 2) the elasticities estimated by Rode et al., 2021 do not provide an indication of the long-run adaptive adjustments of actors (the so called "extensive margin"), as they are estimated exploiting contemporaneous energy-demand and weather co-variation. The works that have estimated the extensive margin in energy demand, although with different methods, come to a similar conclusion: i.e. long-term elasticities differ (and are typically higher) higher than short-term ones (De Cian and Sue Wing, 2019; Auffhammer 2022).

My concern derives from the fact that works that adopt sectorally explicit coefficients for the calibration of the energy demand for adaptation (and its tradeoff with mitigation), reach opposite conclusions with respect to this work (see an Ruijven et al., 2019; Colelli et al. 2022).

My suggestion to the authors would therefore be to clearly state in the discussion section the limitations that follow from the adoption of Rode et al., 2021, particularly as for point 1) and 2). Similarly, a more extensive discussion of the authors' explanations to why this manuscript finds opposite conclusions to the few other comparable studies (Colelli et al. 2022; Van Ruijven et al., 2019) is needed.

A sensitivity analysis of the results based on elasticities from different sources would be ideal in order to really convince the readership that the strong conclusions reached by the manuscript do not depend on one specific empirical work used for calibration.

Minor comments:

I found it hard to investigate heterogeneity across regions with the scale of Figure 4-a, which collapses a lot of variation in

the lowest colour interval making India look very similar to the US. Furthermore, I would suggest to produce two maps: one based on the cooling channel and one based on the heating channel (i.e. providing a decomposition of the CAF).

I would suggest to discuss regional heterogeneity more thoroughly in the results section. One example would be clarifying why India is a country with a large negative CAF despite (as the author mention): a) its currently carbon-intensive power mix; b) that energy demand changes should come mostly from additional electricity for cooling than from lower fossil demand from avoided heating.

References not cited in the manuscript:

Auffhammer, M. (2022). Climate Adaptive Response Estimation: Short and long run impacts of climate change on residential electricity and natural gas consumption. *Journal of Environmental Economics and Management*, 114, 102669.

Schleypen, J.R., Dasgupta, S., Borsky, S., Jury, M., Ščasný, M., Bezhanishvili, L. (2019). D2.4 Impacts on Industry, Energy, Services, and Trade. Deliverable of the H2020 COACCH project.

Reviewer #2

(Remarks to the Author)
Dear Editor

This paper analyzes to what extent changes in energy demand due to climate change impacts may alter the magnitude of global warming. While this question is interesting and relevant, it is not very clear to me what this article really adds to the existing literature.

The main conclusion of this study, namely the fact that climate change may globally decrease energy demand and GHG emissions (because it will reduce demand for heating more than it will increase demand for air conditioning) has already been analyzed in different papers, with approaches close to the one done here. Much work remains to be done on this subject, though, as these existing studies have many limitations, but none of their research gaps is addressed here.

To justify the novelty of their work, the authors say that their study is "dramatically expanding the scope and increasing the accuracy of prior efforts to capture individual components of the CAF" without further explanations. I have a hard time being convinced by this strong statement. I encourage the authors to give more details, and especially analyze what their paper adds for instance to the many papers published through the Energya project (e.g. <https://www.nature.com/articles/s41467-022-32471-1> <https://www.nature.com/articles/s41467-019-10399-3> or <https://www.sciencedirect.com/science/article/pii/S0378778822004868?via=ihub>).

Moreover, the authors derive very detailed figures from their analysis (e.g. on the reduced economic costs of climate change), even though their assumptions are extremely simplistic. Their hypothesis that countries will keep the same emission factors as today is especially strange, in the context of prospective scenarios. It is also slightly incoherent with the SSP framework, as SSP scenarios include hypotheses on technological changes, notably in energy production. Changing the energy mix of a country would deeply impact the results presented here.

Moreover, along a similar idea, the authors use fixed energy consumption responses to daily variations in temperature. This is again a simplistic approximation, as this response will a priori change drastically over time, due to the increase in air conditioning equipment rate across the world (see e.g. <https://www.iea.org/reports/the-future-of-cooling>), to changes in heating and cooling technologies, and, more generally, to changes in buildings insulation and to the adoption of varying climate change adaptation strategies (see e.g. <https://iopscience.iop.org/article/10.1088/1748-9326/abc044>).

Reviewer #3

(Remarks to the Author)
Review of "The Climate Adaptation Feedback"

Summary

This paper studies the so-called "Climate Adaptation Feedback" (CAF), namely whether adaptation to anthropogenic climate change results in a positive or negative contribution to anthropogenic greenhouse gas emissions. In particular, the paper leverages recently published research to study the CAF from global energy consumption. It estimates quite a substantial

negative feedback. That is, adaptation to climate change reduces emissions from energy consumption on net, due to emissions from reduced heating demand outweighing emissions from increased cooling demand.

I like this paper a lot. It makes a crisp point with a nice conceptual framing, substantiated by some incredibly detailed modelling. Therefore, I would like to see it published in Nature Communications. However, first I would like to table some queries about modeling choices.

Comments

- I am not entirely happy about the assumption of fixed emissions factors at their 2010-18 averages. The authors defend this choice as avoiding “the inconsistency that would result from changing emissions factors while maintaining an SSP-RCP that fixes baseline emissions”. I understand this point, but it seems surmountable, for example by generating scenario-specific forecasts for emissions factor changes. Alternatively, a simpler approach that seems to me to still be safe enough would be to use longer-term historical trends in emissions factors as the basis for projecting presumably declining trends in the future that are fixed across scenarios. That is to say, even if you worry about the inconsistency, I don't think you need to assume fixed emissions factors. Whatever happens to climate policy from now on, it seems reasonable to suppose that renewables are here to stay, will continue to diffuse and assume a rising share of at least the electricity generation mix, and will come down in cost.
- I am also not entirely happy with the choice of RCP8.5 as the baseline scenario. I think few people regard RCP8.5 as a plausible baseline anymore. It tends to lead to bigger climate impact projections, which might explain its continuing popularity in the literature, but developments in policy and technology make the most likely scenario of future emissions lower. RCP8.5 is outside even the 90% confidence interval of the RFF-SPs, I think. RCP4.5 would be the SSP that best fits a current policies scenario. If you want a higher emissions baseline, then I would not go beyond RCP7.0.
- My last comment on modeling choices relates to the comparison of the CAF with countries' NDCs. I am not sure this is currently set up as a fair comparison, because the NDCs prescribe emissions reductions in 2030, yet the CAF is quantified up to 2050. This makes the CAF look larger in relation to the NDCs than it probably is. However, I may have misunderstood how the underlying literature on the NDCs, which this paper relies on, deals with post-2030 emissions reductions. Yet, if this literature essentially freezes climate action in 2030, I would say my point stands.
- The authors rightly steer the reader towards interpreting the results in terms of a 'battle' between declining emissions from space heating and increasing emissions from AC, yet technically the modeling includes all uses of energy in the economy outside transportation. The authors may want to square that circle by discussing more directly how heating/cooling drives the numbers. For example, it might be useful to report how much of non-transportation energy consumption goes to space heating and AC.
- While cumulating discounted damages from 2020 to 2099 is a perfectly natural thing to do from an economic point of view, it is a little difficult to put the resulting \$1.8 trillion in context. Most people would think of comparing it with current GDP, but that is a flow and this is more like a change in a stock. Could you put this number in context?
- The fact that the CAF varies little across SSP scenarios is at first unintuitive, given how much uptake of AC is determined by income (see also Davis et al., 2021). It would be helpful to have a slightly deeper explanation of this result.
- The quantification of the heterogeneity in adaptation-induced emissions changes across countries would be more convincing if it controlled for country size (normalizing by population or GDP) – at present this result seems quite strongly influenced by country size but size seems of secondary interest.
- I would like the authors to consider whether the language in the final paragraph of the discussion exaggerates the extent to which current thinking and policy separates mitigation and adaptation. I do agree with this point to a certain extent, but I think equally any climate policy expert worth their salt would agree that (i) a good climate policy does both adaptation and mitigation and (ii) there are interactions between the two.

Minor comments

- Line 37: “incorporates” should be “incorporate”.
- Line 226: “avoids 1.8 trillion [OF...] in present value terms”
- Figure 1: perhaps you want to add an explanation of panels (e) and (f), for completeness?
- Line 347: I think ACC means anthropogenic climate change, but I didn't remember seeing it spelled out previously.
- Estimating the GMST-cumulative CO₂ relationship: in general, I liked this approach a lot, but I did wonder about the estimate of beta that is 1.4 times the median estimate of the TCRE. In particular, I wanted to get a handle on how the indirectly estimated contribution of non-CO₂ GHGs compares with other estimates.

Version 1:

Reviewer comments:

Reviewer #1

(Remarks to the Author)

I thank the authors for their detailed work and replies, which I consider satisfactory for the purpose of accepting the paper for publication.

I only have a few minor edits I would suggest, to enhance the clarity of some key points of the work:

- While the authors correctly specify that the spatial resolution of energy demand responses is much larger than in previous works (~25,000 subnational localities), I would find it useful if the authors included a brief explanation on the source of such heterogeneity, namely the possibility to identify locality-specific shocks thanks to the interaction coefficients between temperature exposure, income and average climate estimated in a single econometric equation by Rode et al.

- The in depth case study on India is very interesting and useful. A key point that drives this work's result, stressed indeed by the authors in the SI - is:

"This result is driven by a key finding of the ref. (1) paper, which is that for lower-income populations, electricity demand is essentially unresponsive to high temperatures, while other fuels (including biomass burning, among other sources) are used by such populations to respond to moderate to cold conditions. Even though India is projected to experience substantial economic growth under the SSP2 socioeconomic scenario, its relatively low level of baseline income inhibits a stronger electricity demand response (1)."

In fact, the authors note that the fossil fuels reductions they project are "about half of aggregate consumption today".

First, I think the above-mentioned comment provided by the authors could be moved to the discussion section, as it can help readers to understand the underlying dynamics driving the results.

Second, the specific result on India's fuel reductions suggests that a great amount of energy is already being used today to heat Indian homes - or for other heating-related energy uses. If possible, it would be very useful to add in the SI section a rough calculation/statistic of present-day fossil fuel consumption for heating services in the country (eg exploiting IEA or other institutional data), to put the projected reductions into perspective.

(Remarks on code availability)

Reviewer #2

(Remarks to the Author)

Dear Editor

While I acknowledge the efforts made by the authors, I remain unconvinced by their response and by their results. In this paper, they simulate with an econometric approach how changes in temperature due to climate change may impact energy demand. They then use country-specific carbon intensities to translate these energy demand variations into carbon emission variations.

My first remark is about the novelty of this approach, and what it adds to the literature. The authors explain that their work is novel because, compared to existing papers:

1. it assesses energy demand variations caused by changes in temperature with a much finer spatial resolution (25,000 subnational localities in 146 countries) and a much finer temperature resolution (energy demand response to the full distribution of daily temperature realizations, instead of commonly used heating-degree days and cooling-degree days)
2. it goes beyond estimating energy demand variations, and computes variations in GHG emissions

Regarding point 1, assessing an energy demand response with an extremely fine resolution does not necessarily lead to a better assessment, because of the risk of over-fitting. This is especially concerning because the authors find conclusions opposite to the rest of the literature, and justify this difference by the fact that they precisely compute more response functions. The authors should try to find out why using their data leads to such results. They should try to do their analysis with temperature thresholds, see if the results are different, and explain why assessing an energy-demand response to the full distribution of daily temperature realizations could lead to better predictions than using the commonly used temperature thresholds. They should also assess the spatial heterogeneity of the response functions, and see for instance if some specific locations are driving the results. Simply saying that, because they have more data points and do many more

regressions, they are more correct than other papers, is not correct.

Regarding point 2, the authors use an extremely crude, and, to my sense, misleading, approach to translate energy demand variations into GHG emissions variations. Compared to the first version of the manuscript, the authors do not suppose any more that countries emission factors will remain constant over the XXIst century. However, they now suppose that countries emission factors will evolve following the current average global trend in emission intensity. Such a scenario does not make any sense to me: the global trend in emission intensities hides a wide variety of changes at each country level. Moreover, even if the authors were using country-specific trends, it would be strange to rely on such trends. Most countries have indeed already begun to implement deep changes in their electricity production technologies, for instance, so current trends are not particularly representative of what is expected over the next decades. The authors state that there are "no existing country-specific projections of future emissions factors up to 2100". This assumption is partly false, as this is specifically what Integrated Assessment models are providing (see for instance <https://unfccc.int/topics/mitigation/workstreams/response-measures/modelling-tools-to-assess-the-impact-of-the-implementation-of-response-measures/integrated-assessment-models-iams-and-energy-environment-economy-e3-models> or <https://www.carbonbrief.org/qa-how-integrated-assessment-models-are-used-to-study-climate-change/>). This is for instance what is done in ref. 22 of the paper, in which "fuel and technology choices are optimized intertemporally, under a set of constraints, including carbon and other energy prices".

More globally, the discrepancy between the extremely detailed regressions of point 1 and the overly simplistic factors used for point 2 is also of concern. Even if point 1 uses very detailed data, the overall level of confidence of any scientific result is always determined by the weakest point of the analysis. The fact that the energy demand response function relies on very fine data cannot be used as a pretext to hide the fact that simplistic hypotheses on emission factors are one of the main drivers of the results.

Another general comment is the fact that the authors often tend to exaggerate their contribution. The term "Climate Adaptation Feedback", especially, is rather misleading. As stated by the authors, they do not capture future shifts in energy demand responses used to adapt to climate change driven by factors outside the two-factor model of Ref. 8 (changes in income and temperature) and they do not take into account any potential changes in building insulation levels, in heating or cooling technologies, or, more generally, any specific climate change adaptation strategies (urban greening etc.). For the sake of clarity I would recommend using a more specific term, such as "heating and cooling demand if countries keep present technologies and emission factors".

(Remarks on code availability)

Reviewer #3

(Remarks to the Author)

I think the authors have taken adequate steps to address my concerns and therefore from my perspective the paper can proceed to publication.

(Remarks on code availability)

Version 2:

Reviewer comments:

Reviewer #1

(Remarks to the Author)

I thank the authors for their detailed replies and recommend the paper for publication.

(Remarks on code availability)

Reviewer #2

(Remarks to the Author)

The authors did a nice job carrying out extra analyses to support the robustness of their methodological approach, and properly answered my comments regarding this topic.

However, I think that they did not really respond to my comment stating that they tend to exaggerate their contribution to the literature. Some claims made in the paper are indeed still slightly misleading, especially regarding the novelty of the topic and the comparison with existing results. The authors state that their main contribution is to "formalize the notion of the CAF and provide a tractable framework for its calculation". The CAF, as the authors indicate in the text, has already been presented in several papers, and it is difficult to blankly state, as the authors do, that they "formalize" it in any innovative way. It is also difficult to see why the existing assessment of the CAF, that the authors cite, was not computed through a "tractable framework".

I have the impression that the main contribution of this paper is to assess the CAF with a new approach, and to show that it leads to different results than the 2022 paper by Colelli et al. As the authors explained in their rebuttal letter, existing papers use IAMs to assess the CAF or future energy demand responses to climate change impacts. Such an approach comes at the expense of granularity. The approach used here relies on historical data, and enables to take into account much finer changes in temperature variations and a much larger spatial heterogeneity. However, it can't take into account any voluntary government policy or technological innovations in the scenarios derived, as IAMs do. There is a long-standing debate on the respective qualities and drawbacks of both approaches, and it is not really possible to tell if one is really more informative than the other. The main policy-relevant message from the paper, to my sense, is therefore not that the CAF is negative, but that, depending on the approach used to assess it, there is an uncertainty regarding its sign and its magnitude.

I would recommend to clearly state this at the beginning of the text and in the abstract. In section 2, some figure should be given about the results obtained by Colelli et al. (2022) when presenting the concept of the CAF, and, in the methods section, the key differences between the approach used here and in this paper should be explained. As a large part of the difference is coming from the choice of future energy changes projections, the authors should also present the main differences between the projection by Rode et al. (2021) and the other existing projections, both in terms of methodology and in terms of global results.

Minor comments:

- line 296/297 "This prior study is complementary to our analysis in that it computes the CAF under future conditions in which a global social planner optimally balances benefits and costs of mitigating climate change"  if I am not mistaken, the WITCH model used by Colelli et al. does not balance benefits and costs of mitigating climate change (i.e. it does not compute an optimal climate change mitigation level), but balances the benefits and costs of various mitigation policies under the constraint that it should reach a prescribed CO2 emission trajectory corresponding to some SSP/RCP scenarios

- line 310/311 "The use of a fixed SSP-RCP baseline prohibits us from examining the CAF in tandem with decarbonization scenarios more sophisticated than extrapolating historical trends."  I do not understand this sentence. In different SSP-RCP scenarios, decarbonization is supposed to occur at different speeds. Why would using exogenous decarbonization scenarios make less sense than using a fixed historical trend ?

(Remarks on code availability)

Version 3:

Reviewer comments:

Reviewer #2

(Remarks to the Author)

Major Comments

=====

The authors did a nice job modifying the text, which now much clearer, especially regarding its contribution to the literature.

These modifications have only been done in the methodology and introduction sections, however, and the conclusion sections of the paper have not been updated (especially the section "Implications for mitigation policy"). They are still written, therefore, as if the results of this paper were definitive, and not a complementary view in an existing debate.

That is why I would suggest removing all the blank statements, such as "Our findings, inclusive of statistical and climatological uncertainty, indicate that even a modest positive CAF is highly unlikely, alleviating such concerns." and replace them by actual discussions of what the new results add to the debates. For instance, it could be written "Our findings indicate that a positive CAF may not be as likely as thought before".

Moreover, in the "Implications for mitigation policy" section, the authors should discuss to what extent their hypotheses may be more policy-relevant, or not, than what was done in the existing literature.

Other Comment

=====

"First, we conduct a sensitivity analysis that captures one dimension of such a coupling; our dynamic version of the CAF shown in Figure 3 recalculates the energy demand projections from ref. (8) to account for the endogenous response of temperature to prior years' energy demand induced by the CAF itself. Results are virtually unchanged"

 This computation is interesting, and useful in the paper, however, I do not see why doing this actually captures one dimension of such a coupling ? I would suggest removing this sentence to keep only the second argument "as long as CO2 intensities associated with non-electricity energy use do not decline dramatically relative to those for electricity, our estimate should serve as an upper bound on the magnitude of the CAF when general equilibrium effects are small"

Also, if I am not mistaken, your argument also holds only if there is no switch from non-electricity energy use to electricity energy use (e.g. people switching from coal or gas heating systems to electric heating systems, or people switching from

gasoline cars to electric cars). As the electrification of energy uses is a common policy goal in many countries, and is a trend which is widely observable, this precision is important, and questions the relevance of the approach.

Minor comments

=====

In the abstract: "These findings indicate that while business-as-usual adaptive energy use is unlikely to accelerate warming, a reevaluation of existing mitigation commitments is merited"

 I am not convinced by the last statement that you introduce in your abstract, and would suggest removing the sentence "a reevaluation of existing mitigation commitments is merited".

"This allows us to be neutral on how future policy may affect adaptive energy use and instead calculate the CAF as a function of future adaptation solely based on empirical estimates of

(Remarks on code availability)

REVIEWER 1

We thank Reviewer 1 for their insightful comments and suggestions. We believe that by addressing each of them, the paper has been much improved. This document provides a guide to the nature and location of the changes prompted by your comments. Specifically, we reproduce each of your comments italicized in gray and provide our replies in black. Excerpts from the paper are indented with the line numbers noted. Figures that are part of this document are prefixed with the letter “R” in order to distinguish them from figures in the paper.

This work provides a framework for quantifying the implications of energy used for adaptation on global GHG emissions and mitigation goals. This is a timely and relevant work, that focuses on a channel of anthropogenic [sic] emissions often understudied. The paper is also concise and well written. The methodology is well described, and robust as for the method directly proposed in this work for the computation of the CAF. The only important remark I have concerns the possible limitations of one key input used to compute the CAF: the energy demand elasticities, taken from Rode et al., 2021. My concern derives from the following points:

1) the sectoral aggregation of energy demand performed in Rode et al., 2021 may mask several different responses of opposite sign that add up to zero in the estimated elasticities, eg consumption of fossil fuels by buildings for heating and by the industrial sector for cooling (as documented eg in De Cian and Sue Wing, 2019; Schleypen et al., 2019).

The reviewer is correct that energy demand across different sectors and/or fuels is likely to exhibit different responses to climate change, as has been elucidated in prior work (De Cian and Sue Wing 2019; Colelli et al. 2022). However, the goal of our analysis is to define and quantify the “climate adaptation feedback” – that is, how *total* energy demand responses to climate change will alter global greenhouse gas (GHG) emissions, ultimately changing global average temperatures. Because GHGs are rapidly mixed in the global atmosphere and lead to the same global warming regardless of where and from what source they were emitted, the value of the CAF depends *only* on the aggregate energy demand changes under climate change, and not on the individual sectoral components. It is critical that the aggregate energy demand response to changes in the climate be econometrically unbiased, but a decomposition into sectoral and fuel-specific effects has no bearing on the value of the CAF. Therefore, we define and focus our analysis on the concept of the CAF, which depends only on the aggregate response of energy demand to temperature change.

However, we do agree with the reviewer that it is interesting to understand how different fuel sources may differentially contribute to the aggregate CAF value. We therefore ensure that these differential effects are conceptually clear in Figure 1, where we diagrammatically define the CAF and illustrate how it is composed of heterogeneous demand responses from different fuels used in both heating and cooling. We revisit the critical distinction between fuel sources in our empirical analysis in Figure 3, where we break down the total CAF to show that there is a warming feedback caused by increased electricity consumption (which is largely used for cooling) but also a cooling feedback caused by decreased other fuels consumption (which are largely used for heating).

To ensure that our focus on the CAF, as opposed to sectoral responses, is clear, and to highlight the fact that the CAF combines responses of opposite sign originating from heterogeneous underlying demand responses, we have amended the text in two places. First, in the section defining the CAF and introducing our methodology, we now highlight that the warming or cooling potential of energy-based adaptation depends only on the aggregate changes in demand. The new text reads (lines 96-106; edited text in red):

“We develop a framework to sign and quantify the net effects of the forces illustrated in Figure 1. Specifically, we define the CAF in any given year as the difference in global mean surface temperature (GMST) between a baseline value (e.g., projected warming under the Representative Concentration Pathway 8.5 (RCP8.5) emissions trajectory) and one accounting for future energy-based adaptation (e.g., projected warming under RCP8.5 plus the net change in emissions from adaptation) (see Supplementary Materials Section 5.1). We note that while energy demand responses to warming in specific sectors and by specific energy sources will be heterogeneous, as illustrated in Figure 1, the sign and magnitude of the CAF depend only on the net change in resulting emissions across all sectors and sources. Therefore, we focus our analysis on the aggregate emissions changes resulting from all measurable energy-based adaptations.”

Second, in the Results section, we point to well-established literature (cited by the reviewer) that, consistent with our findings, shows the CAF is composed of fuel-specific responses that point in two different directions. The new text reads (lines 176-186; edited text in red):

“Figure 3b displays how several factors contribute to the sign and magnitude of the benchmark CAF (first bar). First, adaptation-induced changes in electricity consumption, which are largely driven by increased demand for cooling under rising temperatures (4; 36; 8), lead to a positive value of 0.06°C (second bar). This electricity effect has been the focus of prior discussions of energy-based adaptation (11; 21). However, a negative CAF emerges when we add changes in demand for other fuels, whose value of -0.18°C (third bar) more than offsets the positive component from electricity, due to substantial projected declines in demand for heating under climate change. These findings highlight the importance of accounting for all forms of energy demand that will change in response to climate change. Moreover, they reinforce results in the previous literature demonstrating that energy demand responses to climate change differ in sign and magnitude across distinct sectors and fuel sources (16).”

2) the elasticities estimated by Rode et al., 2021 do not provide an indication of the long-run adaptive [sic] adjustments of actors (the so called [sic] "extensive margin"), as they are estimated exploiting contemporaneous energy-demand and weather co-variation. The works that have estimated the extensive margin in energy demand, although with different methods, come to a similar conclusion: i.e. long-term elasticities differ (and are typically higher) higher than short-term ones (De Cian and Sue Wing, 2019; Auffhammer 2022).

We agree wholeheartedly with the reviewer that accounting for extensive margin changes in energy demand responses to weather variation is of first order importance when generating climate change impact projections. We recognize that our manuscript was unclear on whether and how such adaptive responses

are accounted for in our analysis; in this revision, we have remedied this lack of clarity. Importantly, the elasticities we use from Rode et al. (2021a) *do* include extensive margin adjustments. Specifically, these elasticities account for both long-term adjustments due to a gradually evolving climate (e.g., the increased adoption of air conditioning in today’s temperate climates as they warm in the future) *and* due to growing incomes (e.g., the increased adoption of air conditioning in today’s developing economies as they experience economic growth in the future). To execute this extensive margin adjustment, Rode et al. (2021) estimate regression models that interact long-run average climate and long-run average GDP per capita with local short-run variation in the weather (see equation (3) of the Methods section in Rode et al. for details), an approach analogous to the two-step methods used in Auffhammer (2022) and Colelli et al. (2023). These econometric estimates are then used in combination with climate and socioeconomic projections to estimate how energy demand responses to daily weather realizations will change due to extensive margin adjustments.

We apologize that this critical feature of our analysis was unclear in the first version of the manuscript. We have remedied this with updated text and with a new analysis that demonstrates the quantitative importance of accounting for the extensive margin. We detail these changes here.

First, in Section 2 of the paper, where we define the CAF and briefly describe its methodology, we now clarify that extensive margin adjustments to energy demand *are* accounted for in our estimates. We specifically link to prior literature cited by the reviewer on the importance of this feature. The new text now reads (lines 114-119; edited in red):

“The estimated dose-response functions vary across space and time, accounting for the fact that average incomes and baseline climates shape the sensitivity of energy use to temperature changes, for example through changing the adoption and efficiency of energy-intensive technologies. **Accounting for such extensive margin adjustments in the energy sector has been shown to have first-order impacts on projected impacts of climate change on energy demand (4; 9; 16; 21).**”

We additionally clarify how we incorporate these extensive margin adjustments in the Methods section, highlighting that the interaction terms estimated in the Rode et al. (2021) demand response functions represent extensive margin responses. Updated text now reads (lines 447-459; edited text in red):

“As detailed in ref. (8), these energy demand response functions depend primarily on the realization of future daily temperatures within a given impact region, denoted by the vector $T_{p,t}^N$, under a given RCP scenario. The dose-response functions also include higher-order terms of daily grid cell-level temperature realization along with a set of covariates summarized by $X_{p,t}$, which include projections of GDP per capita and population specific to a SSP scenario and long-run averages of cooling and heating degree days under each temperature trajectory. These covariates allow for the response of energy consumption to daily temperature realizations to vary based on how the economic resources and climatology of a given location change in the future. **This ensures the CAF calculation captures the extensive margin of adaptation that takes place over long periods and may be especially important in developing economies, where income growth is likely to lead to substantial increases in cooling and heating technology adoption (4; 8; 9).** We show the quantitative

importance of the extensive margin in shaping the CAF in Supplementary Information Section 2.”

Second, we have added to the paper an analysis that quantifies the critical role of the extensive margin in generating an accurate CAF. Specifically, we use an alternative set of energy demand projections under climate change from Rode et al. (2021a) that *do not* account for how local responses to temperature realizations change over time. That is, these estimates assume that energy demand response functions (or “elasticities”, as the reviewer refers to them) are held fixed into the future. Figure R.1 displays the result, which shows the importance of accounting for the extensive margin when calculating the CAF (in this example, for an SSP2-RCP4.5 scenario). We find that the CAF would be only 20 percent as large without these dynamic extensive margin adjustments. This figure has been added to a new section in the Supplementary Information (as Figure S3) in the revised manuscript. This new Supplementary section reads (SI Section 2, lines 28-49; new text in red):

“Accounting for the Extensive Margin of Adaptation

This section demonstrates the importance of the extensive margin of adaptive energy use for constructing an accurate estimate of the Climate Adaptation Feedback (CAF). In the baseline CAF calculation shown in the main text, estimates of energy demand changes under climate change are constructed using dose-response functions from ref. (1) that account for both long-term adjustments due to gradually evolving climate (e.g., the increased adoption of air conditioning in today’s temperate climates as they warm in the future) and due to growing incomes (e.g., the increased adoption of air conditioning in today’s developing economies as they experience economic growth in the future). To account for such extensive margin adjustments, ref. (1) estimate regression models that interact long-run average climate and long-run average GDP per capita with local short-run variation in the weather (we direct the reader to the Methods section in ref. (1) as well as their Supplementary Information for details), an approach analogous to the two-step method used in ref. (2). These econometric estimates are then used in combination with climate and socioeconomic projections to estimate how energy demand responses to daily weather realizations will change due to extensive margin adjustments.

Here, we quantify the importance of accounting for such extensive margin effects by computing the CAF using a set of demand responses from ref. (1) that *do not* include extensive margin adjustments. Specifically, Figure S3 displays the baseline CAF for the SSP2-RCP4.5 scenario in blue, as well as a counterfactual CAF in red for the same scenario but without extensive margin adjustments. We find that by 2100, the CAF would be only 20 percent as large without these dynamic extensive margin adjustments.

Figure R.1 (Figure S3 in Supplementary Information): This figure compares the baseline CAF from the main text (blue line) to an alternative in which all energy demand responses to future temperature realizations are fixed at 2001-2010 levels (red line). This alternative shows what the CAF would be were extensive margin adjustments in energy demand responses to temperature ignored. Both estimates use the SSP2-RCP4.5 socioeconomic and emissions scenario.”

This treatment of the extensive margin, as estimated by Rode et al. (2021), is standard in the literature (e.g., De Cian and Sue Wing, 2019; Auffhammer, 2022; Colelli et al., 2023). However, this approach is not without its limitations. In particular, future technological innovation and/or rates of adoption may evolve in ways that are not well captured by the two-factor (long-run climate and income) model estimated in historical data by Rode et al. (2021) (or, equivalently, by Auffhammer, 2022 or Colelli et al., 2023). For example, if the relative price of cooling technologies falls in the future, our CAF estimate could be too low, understating positive emissions responses from future cooling-based adaptation. However, it is also possible that energy efficiency in cooling and/or heating technologies may improve with economic growth at rates faster than those observed historically, pushing the CAF in the opposite direction. It is incredibly difficult to forecast these out-of-sample responses beyond what can be motivated by historical data. We have updated the text in our Discussion to openly acknowledge and discuss these challenges, and we detail them in Supplementary Information Section 2. Specifically, we speak to the differential short- and long-run elasticities of energy use that are demonstrated in the literature (Viguié 2021; Davis et al. 2021; Auffhammer 2022). Updated text in the Discussion Section now reads (lines 322-334; edited text in red):

“Third, our analysis is somewhat limited in its coverage of energy-based adaptations. We omit possible feedbacks arising from transportation-based adaptation due to a paucity of empirical estimates for how transportation-related GHG emissions respond to a warming climate (e.g., ref. (48) provide such estimates, but only for the United States). Our estimates may also fail to account for long-run changes in the response of energy demand driven by shifts in preferences or technological change, leading us to not fully capture some facets of the extensive margin for adaptation (9; 20; 21). If the price of electricity falls drastically in the future relative to that of other energy, we may underestimate electricity demand responses which would bias our CAF estimate downward. Conversely, if end-uses for electricity become more efficient in a way previously unexplained by income growth, our CAF will overstate future emission from electricity. More generally, our estimates may fail to capture future shifts in energy demand responses used to adapt to climate change driven by factors outside the two-factor model in ref. (8).”

These limitations are also discussed at the end of the new Supplementary Information Section 2 (SI Section 2, lines 54-62; new text in red):

“The treatment of the extensive margin for adaptation in ref. (1) is standard in the literature (e.g., refs. (2; 3)). However, this approach is not without its limitations. In particular, future technological innovation and/or rates of adoption may evolve in ways that are not driven by the two factors (long-run climate and income) used in the estimations of refs. (1), (2), and (3), among others. For example, if the relative price of cooling technologies falls in the future, our CAF estimate may be too low and understate the increase in future cooling demand and emissions driven by warming climates. However, it is also possible that energy efficiency in cooling and/or heating technologies may improve with economic growth at rates faster than those observed historically, pushing the CAF in the opposite direction.”

My concern derives from the fact that works that adopt sectorally explicit coefficients for the calibration of the energy demand for adaptation (and its tradeoff with mitigation), reach opposite conclusions with respect to this work (see an Ruijven et al., 2019 [sic]; Colelli et al. 2022). My suggestion to the authors would therefore be to clearly state in the discussion section the limitations that follow from the adoption of Rode et al., 2021, particularly as for point 1) and 2). Similarly, a more extensive discussion of the authors' explanations to why this manuscript finds opposite conclusions to the few other comparable studies (Colelli et al. 2022; Van Ruijven et al., 2019) is needed.

We thank the reviewer for highlighting that we did not sufficiently discuss how our findings fit into the broader literature. We have amended this in two ways. First, we added to the Introduction additional context for our framework. Specifically, the second half of the second Introduction paragraph now reads (lines 36-52; edited text in red):

“It is built from state-of-the-art, globally-comprehensive, empirical estimates of how energy demand responds to changing temperatures (8). These estimates incorporate heterogeneous effects of a changing climate on energy consumption across fuels and locations (2; 3; 8; 9; 16; 17; 18; 19), as well as how income growth and climate change will alter these responses over time.

To our knowledge, this work represents the first definition and calculation of the CAF. In so doing, it builds on prior efforts to empirically estimate how energy demand responds to warming temperatures (16; 20; 21), as these inputs are critical to estimating the adaptation feedback. It also expands the scope and improves the accuracy of the only existing estimate of the global warming effects of energy-based adaptation (22) by: dramatically increasing the spatial resolution of energy demand responses (from 2 macro-regions to ~25,000 subnational localities); leveraging energy demand responses estimated on a comprehensive sample of 146 countries (instead of <50 countries); including the energy demand response to the full distribution of daily temperature realizations, instead of only assessing the effects of temperature events in the tails of the distribution; and accounting for country-level heterogeneity in emissions intensities of electricity and other fuels (instead of using common emission factors across regions).”

We note that these citations point to prior research adjacent to this topic, namely that of De Cian and Sue Wing (2019), Van Ruijen et al. (2019), Vigiú (2021), Colelli et al. (2022) and Colelli et al. (2023), modulo the order of citations in the excerpt above.

In the Discussion section, we have also added the following material explicitly stating that some prior research has found opposite conclusions to ours, and helping the reader understand the key differences between our study and this prior work. Updated text now reads (lines 284-298; edited text in red):

“One existing study used computational general equilibrium methods to endogenize the dynamic relationship between climate change and energy demand (22). The authors find, in contrast to our results, evidence of a positive climate feedback from adaptation. However, this study relies on incomplete and coarse empirical estimates of energy demand responses, which are likely responsible for divergent findings. For example, our CAF is built from 24,378 heterogeneous empirically-derived fuel-specific energy demand responses representing subnational units across the globe, while there are just two aggregate fuel-specific demand responses in ref. (22) representing two large global regions. Similarly, our estimates include demand responses to the full distribution of daily temperature realizations, instead of only the effects of temperature events in the tails of the distribution in ref. (22). Moreover, we account for country-level heterogeneity in emissions intensities, as opposed to heterogeneity across only 17 global macro-regions. Notwithstanding, our sensitivity analysis outlined in Methods Section 5.4 computes a dynamic version of the CAF, finding a nearly identical result to the baseline estimate shown in Figure 3.”

A sensitivity analysis of the results based on elasticities from different sources would be ideal in order to really convince the readership that the strong conclusions reached by the manuscript do not depend on one specific empirical work used for calibration.

We thank the reviewer for this comment. It prompted us to realize the paper was not clear enough as to why we selected the estimates from Rode et al. (2021) to derive our estimate of the CAF. We have remedied this in the updated draft, as detailed below.

In order to construct a comprehensive and accurate CAF, estimates of the temperature-energy demand function (what the reviewer calls elasticities) must satisfy four key criteria: (1) The estimates must take into account possible changes across the entire temperature distribution, not just specific temperature thresholds (like $<0\text{ }^{\circ}\text{C}$ or $>30\text{ }^{\circ}\text{C}$), because climate change will alter the entire distribution of daily temperatures; (2) The estimates must be differentiated between electricity and other fuels, allowing for a different effect of temperature on the demand for both types of energy. This differentiation is important since electricity typically has a lower carbon intensity than other fuels, and responds most strongly to cooling demand (whereas other fuels responds more strongly to heating demand); (3) The estimates must incorporate extensive margin responses of energy demand to changes in temperature over time (the importance of this is discussed above); and (4) The estimates must allow for different localities to have different responses to temperature change. This is necessary because climate change will have different impacts across space, and because energy demand in each location depends on local energy infrastructure and policy (e.g., subsidies).

To our knowledge, the only empirical estimates of the temperature-energy demand function that satisfy these 4 criteria are generated by Rode et al. (2021). This paper represents the most recent and comprehensive global analysis of energy demand responses to climate change published to date. Importantly, prior studies that do not meet all criteria are very likely to generate different results for the CAF because they omit key aspects of the energy-temperature adaptive response. For example, De Cian and Sue Wing (2019) represents a possible set of empirical estimates that could be used in place of Rode et al. (2021), as the authors estimate multi-country energy-temperature demand responses. However, this paper: excludes the effect of temperature realizations in the open interval (12.5, 27.5) degrees C; estimates just two sets of fuel-specific demand responses (one for temperate and one for tropical regions, as opposed to the 24,378 heterogeneous sets of fuel-specific responses estimated in Rode et al.); relies on data from 6-49 countries, depending on the fuel and sector (vs. 149 countries in Rode et al.); and includes natural gas responses to extreme cold only for the commercial sector (vs. all non-transport sectors in Rode et al.). This lack of comprehensiveness will bias estimates of the total emissions, and therefore warming, resulting from energy-based adaptation.

To demonstrate the importance of just one of these 4 criteria, Figure R.2 shows how the distribution of country-level average daily temperatures within the $[12.5^{\circ}\text{C}, 27.5^{\circ}\text{C}]$ interval is a crucial aspect of future climate change. The figure compares the global frequency of country-level daily average temperatures across 160 countries in 2006 with forecasts for average daily countries in 2100 under RCP8.5 using the ACCESS1-3 model (Copernicus, 2021). In 2006, average daily temperature readings in 160 countries were between the 12.5°C and 27.5°C degree interval on 58.6 percent of country-days. In 2100, this interval is forecast to contain only 37.8 percent of country days – such large changes in exposure to these common temperatures has important implications for total energy use. The conditional mean of days realized in this interval falls under a changing climate from 22.3°C to 20.7°C ; this difference in conditional means is both statistically significant [$P(T > t) < 0.0001$] and large magnitude. This difference extends to higher moments; a Kolmogorov-Smirnov test rejects the null hypothesis of equal distributions in 2006 and 2100 of average daily temperatures on this interval at the 1 percent level. The estimates in De Cian and Sue Wing (2019) do not account for the shift in the distribution of temperature realizations that account for almost 40 percent of country-days across all countries, and over 50 percent of realizations in places that comprise a large share of energy demand such as the United States or China.

Figure R.2: Frequency of daily mean surface temperatures for 160 countries. Blue bars show historical realizations in 2006 while red bars show forecasts under RCP8.5 from the ACCESS1-3 model in the year 2100 (Copernicus 2021). The 160 countries are drawn from the shapefiles for each ISO-3 code taken from the GADM database (<https://gadm.org>).

In the paper, we now explicitly list these four criteria and justify our choice of Rode et al. (2021) for inputs into the CAF calculation. Specifically, in the Methods section 5.1 we write (lines 471-480; new text in red):

“There are multiple possible sources of empirically-based energy-temperature demand responses that we could have used to estimate Equation 2. In selecting a source, we sought to identify estimates that: (1) account for demand responses to changes across the entire temperature distribution, not just extreme temperatures; (2) are differentiated between electricity and other fuels, allowing for a different effect of temperature on the demand for multiple types of energy; (3) incorporate extensive margin responses of energy demand to changes in temperature over time; and (4) allow for different localities to have different responses to temperature change. While other multi-country energy-temperature demand estimates exist (e.g., ref. (16)), to our knowledge ref. (8) represents the only study meeting all four of these criteria.”

Because no other study meets all 4 criteria above, we do not conduct sensitivity analyses in which we substitute alternative energy-temperature demand responses for Rode et al. (2021) estimates. However, we do perform several sensitivity analyses in the paper to probe the consistency of results that rely on Rode et al. (2021) demand responses (in addition to showing confidence intervals for the CAF that account for statistical uncertainty in the Rode et al. (2021) estimates themselves). These include: a) a version of the

CAF that endogenizes the dynamics of energy-based adaptation by updating the original projections of global emissions at each time horizon to account for the effects of historical adaptive emissions through time (Figure 3); b) a version of the CAF that projects changes in fuel-specific emissions intensities (Figure 3); c) versions of the CAF for distinct socioeconomic and emissions scenarios (Figure 3); and d) a version of the CAF that shuts down the extensive margin of the energy-temperature demand response (Figure S3).

Minor comments:

I found it hard to investigate heterogeneity across regions with the scale of Figure 4-a, which collapses a lot of variation in the lowest colour interval making India look very similar to the US. Furthermore, I would suggest to produce two maps: one based on the cooling channel and one based on the heating channel (i.e. providing a decomposition of the CAF).

We thank the reviewer for these suggestions, both of which we have implemented. First, we updated the color scale in Figure 4a to highlight the large variation across countries in adaptation-induced CO₂ emissions. As the reviewer pointed out, the previous color scale masked much of this variation. The new figure is:

Figure R.3 (Figure 4a in main text). International heterogeneity in adaptation-induced cumulative CO₂ emissions. The map and histogram display country-level cumulative adaptation-induced CO₂ emissions in 2099 measured in GtCO₂ (ξ_{2099} in Equation (3)).

We also follow the reviewer’s second recommendation, making two maps that separate emissions driven by electricity (largely used for cooling) from those driven by other fuels (largely used for heating). Figures R.4 and R.5 below show the results. These figures are now in the Supplementary Information and

described in the main text, as detailed below. In line with intuition, most countries are estimated to experience fairly large increases in emissions from increasing electricity demand under climate change, as shown in Figure R.4. However, emissions from increasing electricity demand are offset by the near-universal decline in demand for other fuels driven by warmer temperatures globally, as shown in Figure R.5. Most countries experience more-than-offsetting emissions declines from other fuels, as shown in the main text and Figure R.3.

Figure R.4 (Figure S1 in Supplementary Information): The map displays the country-level cumulative adaptation-induced CO₂ emissions in 2099 due to the electricity component of the CAF. This is measured in GtCO₂, as in main text Figure 4a, and corresponds to the electricity component of Equation (3) in the main text.

Figure R.5 (Figure S2 in Supplementary Information): The map displays the country-level cumulative adaptation-induced CO₂ emissions in 2099 due to the other fuels component of the CAF. This is measured in GtCO₂, as in main text Figure 4a, and corresponds to the electricity component of Equation (3) in the main text.

These new figures are discussed and cited in the main text on lines 206-213 (edited text in red):

“While 85% of countries experience CO₂ emissions reductions, of those countries, the 5th and 95th percentiles of cumulative adaptation-induced emission changes by 2099 are -0.027 and -6.24 GtCO₂, respectively. For the remaining 15% of countries that experience increases in emissions, magnitudes are small, with the 5th and 95th percentile range estimated at 0.0024 to 0.5 GtCO₂. **These net changes in emissions can be decomposed into country-level changes in emissions from electricity, which are positive in most countries (Supplementary Figure S1), and in emissions from other fuels, which are negative in virtually all countries (Supplementary Figure S2).”**

I would suggest to discuss regional heterogeneity more thoroughly in the results section. One example would be clarifying why India is a country with a large negative CAF despite (as the author mention): a) [sic] its currently carbon-intensive power mix; b) that energy demand changes should come mostly from additional electricity for cooling than from lower fossil demand from avoided heating.

We appreciate the suggestion - the reviewer is correct that there is a great deal of regional heterogeneity in the emissions effects of energy-based adaptation (as shown in Figure 4a of the main text), and that we can do more to discuss and explain these differences across space, particularly with respect to India. In response, we have added text in the Results section that discusses regional heterogeneity, with a particular focus on India (noting that we have limited space in the main text and view it as important to maintain a discussion of this heterogeneity in the context of Nationally Determined Contributions, as shown in Figure 4c). We have also added a new Supplementary Information section detailing our investigation of the results in India.

In the results section of the main text, the updated text now reads (lines 187-192; edited text in red):

“Heterogeneity in CO₂ emissions intensity also plays an important role. Using a constant global CO₂ emissions intensity -- a simple average across countries and fuels (fourth bar) -- results in a CAF that's 37% smaller in magnitude than our benchmark estimate, which allows CO₂ emissions intensities vary across countries and fuels. **This result implies that emissions are higher per unit of energy in countries with larger energy-based adaptive responses to climate change.”**

And later at lines 203-223 (edits again in red):

“...There is substantial heterogeneity in the magnitude of adaptation-induced CO₂ emission changes across countries. Figure 4a shows both a map and histogram of country-level cumulative adaptation-induced CO₂ emissions changes by 2099 under SSP2-RCP8.5 (this is denoted as ξ_{2099} in Equation (3)). While 85% of countries experience CO₂ emissions reductions, of those countries, the 5th and 95th percentiles of cumulative adaptation-induced emission changes by 2099 are -0.027 and -6.24 GtCO₂, respectively. For the remaining 15% of countries that experience increases in emissions, magnitudes are small, with the 5th and 95th percentile range estimated at 0.0024 to 0.5 GtCO₂. **These net changes in emissions can be decomposed into country-level changes in emissions from electricity, which are positive in most countries (Supplementary Figure S1), and in emissions from other fuels, which are negative in virtually all countries (Supplementary**

Figure S2). The case of India is particularly striking, as it exhibits large adaptation-induced declines in CO₂ emissions by end-of-century, despite facing substantial increases in exposure to extreme heat in future years (38). This result is driven partially by the demand responses from ref. (8), which estimate that electricity demand will increase by 4.1 Exajoules (EJs) by 2099, while cumulative demand for other energy will fall by 7.0 EJs, due to electricity-temperature demand responses being very low for lower-income populations. This result is also influenced by heterogeneous emissions intensities: other fuels in India are substantially more emissions-intensive, implying that demand responses translate into an increase of only 0.9 GTCO₂ emissions from electricity compared to a decline of 2.7 GTCO₂ from other fuels. This decomposition is detailed in Supplementary Information Section 4.”

For the specific case of India, we have added an entirely new Supplementary Information section, as we think this case study is both important quantitatively for the global CAF (given India’s size), and that it helps elucidate the mechanisms behind the CAF. This new material in the Supplementary Information reads (SI Section 4, lines 86-123; new text in red):

“A Case Study of India

Figure 4a in the main text shows that India exhibits large adaptation-induced declines in CO₂ emissions by end-of-century, despite facing substantial increases in exposure to extreme heat in future years (6). In this section, we decompose this result, showing that it is driven both by heterogeneous demand responses across electricity (largely used for cooling) and other fuels (largely used for heating), as well as heterogeneous emissions intensities of these two fuel types. Specifically, we show our forecasts for how adaptation to climate change will affect energy demand (Figure S5) and in turn, emissions (Figure S6) for India under our baseline SSP2-RCP8.5 scenario.

Figure S5 shows that by 2099, adaptation to climate change will induce an additional 5 Exajoules (EJs) of electricity demand relative to a counterfactual demand absent adaptation. This effect is almost 50% larger in magnitude than total annual electricity consumption in India today (c.f. Figure 2b in ref. (1)), representing a large increase in demand for cooling. However, Figure S5 also shows that the increased demand for electricity is offset by lower demand for other fuels in a future under climate change. The dashed line in Figure S5 shows that other fuels consumption is projected to fall by over 6 EJs by 2099, or about half of aggregate consumption today. This result is driven by a key finding of the ref. (1) paper, which is that for lower-income populations, electricity demand is essentially unresponsive to high temperatures, while other fuels (including biomass burning, among other sources) *are* used by such populations to respond to moderate to cold conditions. Even though India is projected to experience substantial economic growth under the SSP2 socioeconomic scenario, its relatively low level of baseline income inhibits a stronger electricity demand response (1).

This larger (in magnitude of EJs) demand effect on other fuels is compounded by the fact that on a unit basis, other fuels consumption in India is more carbon intensive than is

electricity. Figure S6 shows the associated effects on emissions from the changes in demand plotted in Figure S6. India's other fuels consumption emits 1.38 kgCO₂ per kWh of energy produced, almost double the factor of 0.8 kgCO₂ per kWh for electricity. These two facts combine to produce the large (negative) change in cumulative emissions for India in 2099, shown in main text Figure 4a.

Figure R.6 (Figure S5 in Supplementary Information): This figure shows the mean value of projected annual changes in energy demand from electricity (solid line) and other fuels (dashed line) due to adaptation to climate change in India between 2020 and 2099. Values are averaged over projections under an SSP2-RCP8.5 scenario.

Figure R.7 (Figure S6 in Supplementary Information): This figure converts the energy use changes in Figure S5 into carbon dioxide emissions using emissions factors for India. CO₂ emissions due to energy-based adaptation are shown for electricity (solid line) and other fuels (dashed line).”

References:

Auffhammer, Maximilian. "Climate Adaptive Response Estimation: Short and long run impacts of climate change on residential electricity and natural gas consumption." *Journal of Environmental Economics and Management* 114 (2022): 102669.

Colelli, Francesco Pietro, Johannes Emmerling, Giacomo Marangoni, Malcolm N. Mistry, and Enrica De Cian. "Increased energy use for adaptation significantly impacts mitigation pathways." *Nature Communications* 13, no. 1 (2022): 4964.

Colelli, Francesco Pietro, Ian Sue Wing, and Enrica De Cian. "Air-conditioning adoption and electricity demand highlight climate change mitigation–adaptation tradeoffs." *Scientific Reports* 13.1 (2023): 4413.

Copernicus Climate Change Service, Climate Data Store, (2021): In situ total column ozone and ozone soundings from 1924 to present from the World Ozone and Ultraviolet Radiation Data Centre. Copernicus Climate Change Service (C3S) Climate Data Store (CDS). DOI: 10.24381/cds.99842490 (Accessed on 28-02-2024).

Davis, L., Gertler, P., Jarvis, S. and Wolfram, C., 2021. Air conditioning and global inequality. *Global Environmental Change*, 69, p.102299

De Cian, Enrica, and Ian Sue Wing. "Global energy consumption in a warming climate." *Environmental and Resource Economics* 72 (2019): 365-410.

Rode, Ashwin, Tamma Carleton, Michael Delgado, Michael Greenstone, Trevor Houser, Solomon Hsiang, Andrew Hultgren et al. "Estimating a social cost of carbon for global energy consumption." *Nature* 598, no. 7880 (2021): 308-314.

Van Ruijven, Bas J., Enrica De Cian, and Ian Sue Wing. "Amplification of future energy demand growth due to climate change." *Nature Communications* 10, no. 1 (2019): 2762.

Viguié, V., S. Juhel, T. Ben-Ari, M. Colombert, J. D. Ford, L. G. Giraudet, and D. Reckien. "When adaptation increases energy demand: A systematic map of the literature." *Environmental Research Letters* 16, no. 3 (2021): 033004

REVIEWER 2

We thank Reviewer 2 for their insightful comments and suggestions, which we think have substantially improved the paper. This document provides a guide to the nature and location of the changes prompted by your comments. Specifically, we reproduce each of your comments italicized in gray and provide our replies in black. Excerpts from the paper are indented with the line numbers noted. Figures that are part of this document are prefixed with the letter “R” in order to distinguish them from figures in the paper.

This paper analyzes to what extent changes in energy demand due to climate change impacts may alter the magnitude of global warming. While this question is interesting and relevant, it is not very clear to me what this article really adds to the existing literature.

The main conclusion of this study, namely the fact that climate change may globally decrease energy demand and GHG emissions (because it will reduce demand for heating more than it will increase demand for air conditioning) has already been analyzed in different papers, with approaches close to the one done here. Much work remains to be done on this subject, though, as these existing studies have many limitations, but none of their research gaps is addressed here.

We thank the reviewer for making clear that we did not sufficiently clarify the key contributions of the analysis, relative to the broader literature. We have amended the text to better situate our paper within the literature and to clarify the novel aspects of our analysis.

Most importantly, the goal of our paper is to introduce and define the Climate Adaptation Feedback (CAF), to explain its importance, and to provide the first global estimate of the CAF using the most recent, comprehensive, and accurate projections of energy demand responses to climate change. To our knowledge, just one prior paper (Colelli et al., 2022) estimates a similar object, computing the warming induced by adaptive energy use under climate change. We detail below how our paper expands the scope and improves the accuracy of that prior result. In addition, our analysis uncovers substantial regional heterogeneity in the CAF and uses such heterogeneity to assess the implications of adaptive energy use for mitigation policy – specifically, the Nationally Determined Contributions under the Paris Agreement.

Our textual changes are in two places. First, we added to the Introduction additional context for our framework. Specifically, the second half of the second Introduction paragraph now reads (lines 36-52; edited text in red):

“It is built from state-of-the-art, globally-comprehensive, empirical estimates of how energy demand responds to changing temperatures (8). These estimates incorporate heterogeneous effects of a changing climate on energy consumption across fuels and locations (2; 3; 8; 9; 16; 17; 18; 19), as well as how income growth and climate change will alter these responses over time.

To our knowledge, this work represents the first definition and calculation of the CAF. In so doing, it builds on prior efforts to empirically estimate how energy demand responds to warming temperatures (16; 20; 21), as these inputs are critical to estimating the adaptation feedback. It also expands the scope and improves the accuracy of the only existing estimate of the global warming effects of energy-based adaptation (22) by: dramatically increasing

the spatial resolution of energy demand responses (from 2 macro-regions to ~25,000 subnational localities); leveraging energy demand responses estimated on a comprehensive sample of 146 countries (instead of <50 countries); including the energy demand response to the full distribution of daily temperature realizations, instead of only assessing the effects of temperature events in the tails of the distribution; and accounting for country-level heterogeneity in emissions intensities of electricity and other fuels (instead of using common emission factors across regions).”

As the reviewer points out, our analysis still has important limitations. To ensure these are fully discussed, and to help the reader understand the key differences between our study and the only prior analogous analysis, we have added text to the Discussion section. Updated text in the Discussion Section now reads (lines 279-298; edited text in red):

“Our analysis has several limitations. First, we define the CAF relative to a fixed SSP for socioeconomic conditions and fixed RCP for baseline global emissions. The advantage of this approach is that SSPs and RCPs are widely used in climate projections and do not already account for emissions arising from energy-based adaptation (44). As discussed above and in Methods Section 5.4, this abstracts from a full characterization of the dynamic interplay between changing adaptive energy demand and climate change. **One existing study used computational general equilibrium methods to endogenize the dynamic relationship between climate change and energy demand (22). The authors find, in contrast to our results, evidence of a positive climate feedback from adaptation. However, this study relies on incomplete and coarse empirical estimates of energy demand responses, which are likely responsible for divergent findings. For example, our CAF is built from 24,378 heterogeneous empirically-derived fuel-specific energy demand responses representing subnational units across the globe, while there are just two aggregate fuel-specific demand responses in ref. (22) representing two large global regions. Similarly, our estimates include demand responses to the full distribution of daily temperature realizations, instead of only the effects of temperature events in the tails of the distribution in ref. (22). Moreover, we account for country-level heterogeneity in emissions intensities, as opposed to heterogeneity across only 17 global macro-regions. Notwithstanding, our sensitivity analysis outlined in Methods Section 5.4 computes a dynamic version of the CAF, finding a nearly identical result to the baseline estimate shown in Figure 3.**”

Along with later in the updated manuscript (lines 322-334; edited text in red):

“Third, our analysis is somewhat limited in its coverage of energy-based adaptations. We omit possible feedbacks arising from transportation-based adaptation due to a paucity of empirical estimates for how transportation-related GHG emissions respond to a warming climate (e.g., ref. (48) provide such estimates, but only for the United States). **Our estimates may also fail to account for long-run changes in the response of energy demand driven by shifts in preferences or technological change, leading us to not fully capture some facets of the extensive margin for adaptation (9; 21; 22). If the price of electricity falls drastically in the future relative to that of other energy, we may underestimate electricity demand responses which would bias our CAF estimate downward. Conversely, if end-uses for**

electricity become more efficient in a way previously unexplained by income growth, our CAF will overstate future emission from electricity. More generally, our estimates may fail to capture future shifts in energy demand responses used to adapt to climate change driven by factors outside the two-factor model in ref. (8).”

To justify the novelty of their work, the authors say that their study is "dramatically expanding the scope and increasing the accuracy of prior efforts to capture individual components of the CAF" without further explanations. I have a hard time being convinced by this strong statement. I encourage the authors to give more details, and especially analyze what their paper adds for instance to the many papers published through the Energya project (e.g. <https://www.nature.com/articles/s41467-022-32471-1> <https://www.nature.com/articles/s41467-019-10399-3> or <https://www.sciencedirect.com/science/article/pii/S0378778822004868?via=ihub>).

We agree with the reviewer that this statement exaggerated our contribution – we have removed it from the revised manuscript, along with any other analogous claims. As detailed in our response to the prior comment, we have clarified our contribution relative to the existing literature throughout the text. Here we list the multiple places in the updated manuscript in which we cite, discuss, and compare prior literature to our analysis (noting that some of these were also mentioned above in response to the related prior comment).

In the Introduction, we write (lines 41-52; edited text in red):

“To our knowledge, this work represents the first definition and calculation of the CAF. In so doing, it builds on prior efforts to empirically estimate how energy demand responds to warming temperatures (16; 20; 21), as these inputs are critical to estimating the adaptation feedback. It also expands the scope and improves the accuracy of the only existing estimate of the global warming effects of energy-based adaptation (22) by: dramatically increasing the spatial resolution of energy demand responses (from 2 macro-regions to ~25,000 subnational localities); leveraging energy demand responses estimated on a comprehensive sample of 146 countries (instead of <50 countries); including the energy demand response to the full distribution of daily temperature realizations, instead of only assessing the effects of temperature events in the tails of the distribution; and accounting for country-level heterogeneity in emissions intensities of electricity and other fuels (instead of using common emission factors across regions).”

In our short methodological summary in the main text, we write (lines 107-119; edited text in red):

“We implement this framework by leveraging newly-available high-resolution projections of future energy changes in response to local temperature realizations from ref. (8). These causal “dose-response” functions represent the change in the use of final energy sources -- electricity and all other fuels -- in response to variations in daily temperature, pooling energy consumption across residential, commercial, industrial, and agricultural end-uses (excluding transportation). Any adaptive actions taken by individuals, firms, or public agencies across a broad spectrum of sectors, such as the use of air conditioning or space heating, are included in these estimates. The estimated dose-response functions vary across space and time, accounting for the fact that average incomes and baseline climates shape the sensitivity of

energy use to temperature changes, for example through changing the adoption and efficiency of energy-intensive technologies. Accounting for such extensive margin adjustments in the energy sector has been shown to have first-order impacts on projected impacts of climate change on energy demand (4; 9; 16; 21).”

In the Results section, we put our estimate of a negative CAF in context of a broader literature (lines 180-186; edited text in red):

“However, a negative CAF emerges when we add changes in demand for other fuels, whose value of -0.18°C (third bar) more than offsets the positive component from electricity, due to substantial projected declines in demand for heating under climate change. These findings highlight the importance of accounting for all forms of energy demand that will change in response to climate change. Moreover, they reinforce results in the previous literature demonstrating that energy demand responses to climate change differ in sign and magnitude across distinct sectors and fuel sources (16).”

As mentioned above, in the Discussion section we help the reader understand the key differences between our study and the only prior analogues (lines 279-298; edited text in red):

“Our analysis has several limitations. First, we define the CAF relative to a fixed SSP for socioeconomic conditions and fixed RCP for baseline global emissions. The advantage of this approach is that SSPs and RCPs are widely used in climate projections and do not already account for emissions arising from energy-based adaptation (44). As discussed above and in Methods Section 5.4, this abstracts from a full characterization of the dynamic interplay between changing adaptive energy demand and climate change. One existing study used computational general equilibrium methods to endogenize the dynamic relationship between climate change and energy demand (22). The authors find, in contrast to our results, evidence of a positive climate feedback from adaptation. However, this study relies on incomplete and coarse empirical estimates of energy demand responses, which are likely responsible for divergent findings. For example, our CAF is built from 24,378 heterogeneous empirically-derived fuel-specific energy demand responses representing subnational units across the globe, while there are just two aggregate fuel-specific demand responses in ref. (22) representing two large global regions. Similarly, our estimates include demand responses to the full distribution of daily temperature realizations, instead of only the effects of temperature events in the tails of the distribution in ref. (22). Moreover, we account for country-level heterogeneity in emissions intensities, as opposed to heterogeneity across only 17 global macro-regions. Notwithstanding, our sensitivity analysis outlined in Methods Section 5.4 computes a dynamic version of the CAF, finding a nearly identical result to the baseline estimate shown in Figure 3.”

Along with later in the updated manuscript (lines 322-334; edited text in red):

“Third, our analysis is somewhat limited in its coverage of energy-based adaptations. We omit possible feedbacks arising from transportation-based adaptation due to a paucity of empirical estimates for how transportation-related GHG emissions respond to a warming climate (e.g., ref. (48) provide such estimates, but only for the United States). Our estimates

may also fail to account for long-run changes in the response of energy demand driven by shifts in preferences or technological change, leading us to not fully capture some facets of the extensive margin for adaptation (9; 21; 22). If the price of electricity falls drastically in the future relative to that of other energy, we may underestimate electricity demand responses which would bias our CAF estimate downward. Conversely, if end-uses for electricity become more efficient in a way previously unexplained by income growth, our CAF will overstate future emission from electricity. More generally, our estimates may fail to capture future shifts in energy demand responses used to adapt to climate change driven by factors outside the two-factor model in ref. (8).”

We note that these added and more thoroughly discussed citations include many papers from the Energya project cited by the reviewer. Specifically: Van Ruijen et al., 2019 (ref. 17), Colelli et al., 2022 (ref. 21), and Colelli et al., 2023 (ref. 22).

Moreover, the authors derive very detailed figures from their analysis (e.g. on the reduced economic costs of climate change), even though their assumptions are extremely simplistic. Their hypothesis that countries will keep the same emission factors as today is especially strange, in the context of prospective scenarios. It is also slightly incoherent with the SSP framework, as SSP scenarios include hypotheses on technological changes, notably in energy production. Changing the energy mix of a country would deeply impact the results presented here.

We agree with the reviewer that the assumption of constant emission factors at the country level is simplistic. There are many technological, economic, and policy-based factors that lead us to expect that emissions factors may decline over time. To the best of our knowledge, there are no existing country-specific projections of future emissions factors up to 2100, and this explains why we used country-specific, *time-invariant* emission factors in our analysis. Nevertheless, in order to assuage the reviewer’s concern (and a concern shared among our research team), in our updated manuscript we now produce alternative CAF estimates in which we allow for each country’s emission factors to evolve over time based on estimated trends in recent years.

Specifically, we performed a sensitivity analysis that shows how the CAF changes when we allow for country-specific emissions factors to vary over time. To do so, we calculate the observed global trend in average emission factors between 2000 and 2018, and then re-run our analysis assuming that the emission factors in each country evolve up to 2100 based on the observed historical global trend. We find that this has an incredibly small impact on the estimated CAF: for our baseline SSP2-RCP8.5 scenario, the CAF declines from -0.12°C to -0.13°C.

A new Supplementary Information section details this analysis (SI Section 3, lines 63-85; new text in red):

“Solving for Global Emissions Factors

The dashed light green line in Figure 3a of the main text displays an alternative estimate of our CAF calculation for SSP2-RCP8.5 in which future fuel-specific emissions factors are adjusted to decay at historical rates. As emissions from both fuels have declined on a unit basis throughout the 21st century, we estimate the CAF in a world where historical

trends continue out through 2099. To calculate an appropriate degree of decay for each fuel, we estimate the annual rate of change in global average emissions factors (in units of metric tons of CO₂ per kilowatt-hour of final energy use) between 2000 and 2018 using data on emissions and final energy consumption from the IEA (4; 5).

Figure S4 displays annual global average emissions factors. To construct rates of change over time, for each fuel h , we estimate the following exponential decay coefficient:

$$\Delta \ln F_t^h = \gamma_h + \varepsilon_{h,t} \quad (1)$$

where ΔF_t^h is the (log) change in the global emissions factor for fuel h from the previous year and $\varepsilon_{h,t}$ are random error terms. The estimated global decay factors $\hat{\gamma}_h$ are -.0008 and -.009 for other fuels and electricity, respectively. We use these estimates to generate a future pathway for each country-by-fuel specific factor as follows:

$$F_{i,t}^h = \exp \left\{ \hat{\gamma}_h (t - 2020) \right\} \bar{F}_i^h \quad (2)$$

where \bar{F}_i^h denotes the original 2010-18 emissions factors calculated based on country-level data between 2010-18 and used for the baseline CAF.

Figure R.8 (Figure S4 in Supplementary Information): This figure shows annual global emissions factors for electricity and other fuels between 2000 and 2018 calculated using data from the International Energy Agency.”

The result from this analysis is presented in Figure 3a in the main text (see below, as Figure R.9). As we estimate electricity’s rate of decarbonization to be an order of magnitude larger than that of other fuels, we find an even smaller increase in emissions from adaptation using electricity. This decline in emissions from new electricity demand due to decarbonization more than offsets the smaller emissions reductions from less demand for other fuels, lowering the CAF from -0.12°C to -0.13°C . This finding reinforces an important aspect of our findings – as long as electricity decarbonizes at least as quickly as other fuels, our estimate of the CAF is an upper bound. We emphasize this fact in the manuscript for the exact reason that *a priori* there is a larger concern over the upper bound on the feedback effect underlying the CAF. Figure 3a and its caption now read (line 387; new text in red):

Figure R.9 (Figure 3a in the main text). The Climate Adaptation Feedback. (a) Solid green line and shaded green area show point estimates and 90% confidence intervals for the Climate Adaptation Feedback (in $^{\circ}\text{C}$), for 2020-2099 under SSP2-RCP8.5 using our benchmark approach. The darker dashed line shows the dynamic CAF, which accounts for how additional climate change from adaptation feeds back into future adaptation, as detailed in Methods Section 5.4. The lighter dashed line shows the CAF recalculated using emissions factors that follow historical trends between 2000 and 2018, as detailed in Supplementary Information Section 3.

In the main text we discuss this new analysis by adding the following at the end of the second paragraph in the results section (lines 167-175; new text in red):

“...We also recalculate the CAF relaxing assumptions over fixed emissions factors. The last series in Figure 3a, the long dashed lime green line, shows the CAF under the assumption that historical global trends in emissions intensity for both fuels between 2000 and 2018

occurs in all countries through 2099. Allowing each sector to continue to decarbonize at historical rates lowers the CAF to -0.13°C under SSP2-RCP8.5. As emissions from other fuels have remained relatively unchanged on a unit basis when compared to the substantial reduction in the emissions intensity of electricity generation over the past two decades, extrapolation of this trend lowers emissions from the electricity sector and amplifies the mechanisms leading to a negative CAF.”

Moreover, along a similar idea, the authors use fixed energy consumption responses to daily variations in temperature. This is again a simplistic approximation, as this response will a priori change drastically over time, due to the increase in air conditioning equipment rate across the world (see e.g. <https://www.iea.org/reports/the-future-of-cooling>), to changes in heating and cooling technologies, and, more generally, to changes in buildings insulation and to the adoption of varying climate change adaptation strategies (see e.g. <https://iopscience.iop.org/article/10.1088/1748-9326/abc044>).

We agree wholeheartedly with the reviewer that accounting for temporal changes in energy demand responses to weather variation is of first order importance when generating climate change impact projections. We recognize that our manuscript was unclear on whether and how such adaptive responses are accounted for in our analysis; in this revision, we have remedied this lack of clarity. Importantly, the demand responses to temperature that we use from Rode et al. (2021a) *do* include estimates of technological adoption and other long-run adjustments (often called the “extensive margin” of the energy demand-temperature relationship). Specifically, these elasticities account for both long-term adjustments due to a gradually evolving climate (e.g., the increased adoption of air conditioning in today’s temperate climates as they warm in the future) *and* due to growing incomes (e.g., the increased adoption of air conditioning in today’s developing economies as they experience economic growth in the future). To execute this extensive margin adjustment, Rode et al. (2021) estimate regression models that interact long-run average climate and long-run average GDP per capita with local short-run variation in the weather (see section of the Methods section in Rode et al. for details), an approach analogous to the two-step method used in Auffhammer (2022). These econometric estimates are then used in combination with climate and socioeconomic projections to estimate how energy demand responses to daily weather realizations will change due to extensive margin adjustments.

We apologize that this critical feature of our analysis was unclear in the first version of the manuscript. We have remedied this with updated text and with a new analysis that demonstrates the quantitative importance of accounting for the extensive margin. We detail these changes here.

First, in Section 2 of the paper, where we define the CAF and briefly describe its methodology, we now clarify that extensive margin adjustments to energy demand *are* accounted for in our estimates. We specifically link to prior literature cited by the reviewer on the importance of this feature. The new text now reads (lines 114-119; edited in red)

“The estimated dose-response functions vary across space and time, accounting for the fact that average incomes and baseline climates shape the sensitivity of energy use to temperature changes, for example through changing the adoption and efficiency of energy-intensive technologies. **Accounting for such extensive margin adjustments in the energy**

sector has been shown to have first-order impacts on projected impacts of climate change on energy demand (4; 9; 16; 21).”

We additionally clarify how we incorporate these extensive margin adjustments in the Methods section, highlighting that the interaction terms estimated in the Rode et al. (2021) demand response functions represent extensive margin responses. Updated text now reads (lines 447-459; edited text in red):

“As detailed in ref. (8), these energy demand response functions depend primarily on the realization of future daily temperatures within a given impact region, denoted by the vector $T_{p,t}^N$, under a given RCP scenario. The dose-response functions also include higher-order terms of daily grid cell-level temperature realization along with a set of covariates summarized by $X_{p,t}$, which include projections of GDP per capita and population specific to a SSP scenario and long-run averages of cooling and heating degree days under each temperature trajectory. These covariates allow for the response of energy consumption to daily temperature realizations to vary based on how the economic resources and climatology of a given location change in the future. **This ensures the CAF calculation captures the extensive margin of adaptation that takes place over long periods and may be especially important in developing economies, where income growth is likely to lead to substantial increases in cooling and heating technology adoption (4; 8; 9). We show the quantitative importance of the extensive margin in shaping the CAF in Supplementary Information Section 2.**”

Second, we have added to the paper an analysis that quantifies the critical role of these long-run adjustments for generating an accurate CAF. Specifically, we use an alternative set of energy demand projections under climate change from Rode et al. (2021a) that *do not* account for how local responses to temperature realizations change over time. That is, these estimates assume that energy demand response functions are held fixed into the future. Figure R.10 displays the result, which shows the importance of accounting for the extensive margin when calculating the CAF (in this example, for an SSP2-RCP4.5 scenario). We find that the CAF would be only 20 percent as large without these dynamic extensive margin adjustments. This figure has been added to a new section in the Supplementary Information (as Figure S3) in the revised manuscript. This new Supplementary Information section reads (SI Section 2, lines 28-49; new text in red):

“Accounting for the Extensive Margin of Adaptation

This section demonstrates the importance of the extensive margin of adaptive energy use for constructing an accurate estimate of the Climate Adaptation Feedback (CAF). In the baseline CAF calculation shown in the main text, estimates of energy demand changes under climate change are constructed using dose-response functions from ref. (1) that account for both long-term adjustments due to gradually evolving climate (e.g., the increased adoption of air conditioning in today’s temperate climates as they warm in the future) and due to growing incomes (e.g., the increased adoption of air conditioning in today’s developing economies as they experience economic growth in the future). To account for such extensive margin adjustments, ref. (1) estimate regression models that interact long-run average climate and long-run average GDP per capita with local short-run variation in the weather

(we direct the reader to the Methods section in ref. (1) as well as their Supplementary Information for details), an approach analogous to the two-step method used in ref. (2). These econometric estimates are then used in combination with climate and socioeconomic projections to estimate how energy demand responses to daily weather realizations will change due to extensive margin adjustments.

Here, we quantify the importance of accounting for such extensive margin effects by computing the CAF using a set of demand responses from ref. (1) that *do not* include extensive margin adjustments. Specifically, Figure S3 displays the baseline CAF for the SSP2-RCP4.5 scenario in blue, as well as a counterfactual CAF in red for the same scenario but without extensive margin adjustments. We find that by 2100, the CAF would be only 20 percent as large without these dynamic extensive margin adjustments.

Figure R.10 (Figure S3 in Supplementary Information): This figure compares the baseline CAF from the main text (blue line) to an alternative in which all energy demand responses to future temperature realizations are fixed at 2001-2010 levels (red line). This alternative shows what the CAF would be were extensive margin adjustments in energy demand responses to temperature ignored. Both estimates use the SSP2-RCP4.5 socioeconomic and emissions scenario.”

This treatment of the extensive margin, as estimated by Rode et al. (2021), is standard in the literature (e.g., De Cian and Sue Wing, 2019; Auffhammer, 2022; Colelli et al., 2023). However, this approach is not

without its limitations. In particular, future technological innovation and/or rates of adoption may evolve in ways that are not well captured by the two-factor (long-run climate and income) model estimated in historical data by Rode et al. (2021) (or, equivalently, by Auffhammer, 2022). For example, if the relative price of cooling technologies falls in the future, our CAF estimate could be too low, understating positive emissions responses from future cooling-based adaptation. However, it is also possible that energy efficiency in cooling and/or heating technologies may improve with economic growth at rates faster than those observed historically, pushing the CAF in the opposite direction. It is incredibly difficult to forecast these out-of-sample responses beyond what can be motivated by historical data. We have updated the text in our Discussion to openly acknowledge and discuss these challenges, and we detail them in Supplementary Information Section 2. Specifically, we speak to the differential short- and long-run elasticities of energy use that are demonstrated in the literature (Viguié 2021; Davis et al. 2021; Auffhammer 2022). Updated text in the Discussion Section now reads (lines 322-334; edited text in red):

“Third, our analysis is somewhat limited in its coverage of energy-based adaptations. We omit possible feedbacks arising from transportation-based adaptation due to a paucity of empirical estimates for how transportation-related GHG emissions respond to a warming climate (e.g., ref. (48) provide such estimates, but only for the United States). Our estimates may also fail to account for long-run changes in the response of energy demand driven by shifts in preferences or technological change, leading us to not fully capture some facets of the extensive margin for adaptation (9; 20; 21). If the price of electricity falls drastically in the future relative to that of other energy, we may underestimate electricity demand responses which would bias our CAF estimate downward. Conversely, if end-uses for electricity become more efficient in a way previously unexplained by income growth, our CAF will overstate future emission from electricity. More generally, our estimates may fail to capture future shifts in energy demand responses used to adapt to climate change driven by factors outside the two-factor model in ref. (8).”

These limitations are also discussed at the end of the new Supplementary Information Section 2 (SI Section 2, lines 54-62; new text in red):

“The treatment of the extensive margin for adaptation in ref. (1) is standard in the literature (e.g., refs. (2; 3)). However, this approach is not without its limitations. In particular, future technological innovation and/or rates of adoption may evolve in ways that are not driven by the two factors (long-run climate and income) used in the estimations of refs. (1), (2), and (3), among others. For example, if the relative price of cooling technologies falls in the future, our CAF estimate may be too low and understate the increase in future cooling demand and emissions driven by warming climates. However, it is also possible that energy efficiency in cooling and/or heating technologies may improve with economic growth at rates faster than those observed historically, pushing the CAF in the opposite direction.”

References:

Auffhammer, Maximilian. "Climate Adaptive Response Estimation: Short and long run impacts of climate change on residential electricity and natural gas consumption." *Journal of Environmental Economics and Management* 114 (2022): 102669.

Colelli, Francesco Pietro, Johannes Emmerling, Giacomo Marangoni, Malcolm N. Mistry, and Enrica De Cian. "Increased energy use for adaptation significantly impacts mitigation pathways." *Nature Communications* 13, no. 1 (2022): 4964.

Colelli, Francesco Pietro, Ian Sue Wing, and Enrica De Cian. "Air-conditioning adoption and electricity demand highlight climate change mitigation–adaptation tradeoffs." *Scientific Reports* 13.1 (2023): 4413.

Davis, L., Gertler, P., Jarvis, S. and Wolfram, C., 2021. Air conditioning and global inequality. *Global Environmental Change*, 69, p.102299

De Cian, Enrica, and Ian Sue Wing. "Global energy consumption in a warming climate." *Environmental and Resource Economics* 72 (2019): 365-410.

Hsiang, Solomon, Robert Kopp, Amir Jina, James Rising, Michael Delgado, Shashank Mohan, D. J. Rasmussen et al. "Estimating economic damage from climate change in the United States." *Science* 356, no. 6345 (2017): 1362-1369.

Rode, Ashwin, Tamma Carleton, Michael Delgado, Michael Greenstone, Trevor Houser, Solomon Hsiang, Andrew Hultgren et al. "Estimating a social cost of carbon for global energy consumption." *Nature* 598, no. 7880 (2021): 308-314.

Van Ruijven, Bas J., Enrica De Cian, and Ian Sue Wing. "Amplification of future energy demand growth due to climate change." *Nature Communications* 10, no. 1 (2019): 2762.

Viguié, V., S. Juhel, T. Ben-Ari, M. Colombert, J. D. Ford, L. G. Giraudet, and D. Reckien. "When adaptation increases energy demand: A systematic map of the literature." *Environmental Research Letters* 16, no. 3 (2021): 033004

REVIEWER 3

We thank Reviewer 3 for their insightful comments and suggestions, which we think have substantially improved the paper. This document provides a guide to the nature and location of the changes prompted by your comments. Specifically, we reproduce each of your comments italicized in gray and provide our replies in black. Excerpts from the paper are indented with the line numbers noted. Figures that are part of this document are prefixed with the letter “R” in order to distinguish them from figures in the paper.

Summary

This paper studies the so-called “Climate Adaptation Feedback” (CAF), namely whether adaptation to anthropogenic climate change results in a positive or negative contribution to anthropogenic greenhouse gas emissions. In particular, the paper leverages recently published research to study the CAF from global energy consumption. It estimates quite a substantial negative feedback. That is, adaptation to climate change reduces emissions from energy consumption on net, due to emissions from reduced heating demand outweighing emissions from increased cooling demand.

I like this paper a lot. It makes a crisp point with a nice conceptual framing, substantiated by some incredibly detailed modelling [sic]. Therefore, I would like to see it published in Nature Communications. However, first I would like to table some queries about modeling choices.

Comments

• I am not entirely happy about the assumption of fixed emissions factors at their 2010-18 averages. The authors defend this choice as avoiding “the inconsistency that would result from changing emissions factors while maintaining an SSP-RCP that fixes baseline emissions”. I understand this point, but it seems surmountable, for example by generating scenario-specific forecasts for emissions factor changes. Alternatively, a simpler approach that seems to me to still be safe enough would be to use longer-term historical trends in emissions factors as the basis for projecting presumably declining trends in the future that are fixed across scenarios. That is to say, even if you worry about the inconsistency, I don’t think you need to assume fixed emissions factors. Whatever happens to climate policy from now on, it seems reasonable to suppose that renewables are here to stay, will continue to diffuse and assume a rising share of at least the electricity generation mix, and will come down in cost.

We agree with the reviewer that the assumption of constant emission factors at the country level is simplistic. There are many technological, economic, and policy-based factors that lead us to expect that emissions factors may decline over time. To the best of our knowledge, there are no existing country-specific projections of future emissions factors up to 2100, and this explains why we used country-specific, *time-invariant* emission factors in our analysis. Nevertheless, in order to assuage the reviewer’s concern (and a concern shared among our research team), in our updated manuscript we now produce alternative CAF estimates in which we allow for each country’s emission factors to evolve over time based on estimated trends in recent years.

Specifically, we performed a sensitivity analysis that shows how the CAF changes when we allow for country-specific emissions factors to vary over time. To do so, we calculate the observed global trend in average emission factors between 2000 and 2018, and then re-run our analysis assuming that the emission

factors in each country evolve up to 2100 based on the observed historical global trend. We find that this has an incredibly small impact on the estimated CAF: for our baseline SSP2-RCP85 scenario, the CAF declines from -0.12°C to -0.13°C.

A new Supplementary Information section details this analysis (SI Section 3, lines 63-85; new text in red):

“Solving for Global Emissions Factors

The dashed light green line in Figure 3a of the main text displays an alternative estimate of our CAF calculation for SSP2-RCP8.5 in which future fuel-specific emissions factors are adjusted to decay at historical rates. As emissions from both fuels have declined on a unit basis throughout the 21st century, we estimate the CAF in a world where historical trends continue out through 2099. To calculate an appropriate degree of decay for each fuel, we estimate the annual rate of change in global average emissions factors (in units of metric tons of CO₂ per kilowatt-hour of final energy use) between 2000 and 2018 using data on emissions and final energy consumption from the IEA (4; 5).

Figure S4 displays annual global average emissions factors. To construct rates of change over time, for each fuel h , we estimate the following exponential decay coefficient:

$$\Delta \ln F_t^h = \gamma_h + \varepsilon_{h,t} \quad (1)$$

where ΔF_t^h is the (log) change in the global emissions factor for fuel h from the previous year and $\varepsilon_{h,t}$ are random error terms. The estimated global decay factors $\hat{\gamma}_h$ are -.0008 and -.009 for other fuels and electricity, respectively. We use these estimates to generate a future pathway for each country-by-fuel specific factor as follows:

$$F_{i,t}^h = \exp \left\{ \hat{\gamma}_h (t - 2020) \right\} \bar{F}_i^h \quad (2)$$

where \bar{F}_i^h denotes the original 2010-18 emissions factors calculated based on country-level data between 2010-18 and used for the baseline CAF.

Figure R.11 (Figure S4 in the Supplementary Information): This figure shows annual global emissions factors for electricity and other fuels between 2000 and 2018 calculated using data from the International Energy Agency.”

The result from this analysis is presented in Figure 3a in the main text (see below, as Figure R.12). As we estimate electricity’s rate of decarbonization to be an order of magnitude larger than that of other fuels, we find an even smaller increase in emissions from adaptation using electricity. This decline in emissions from new electricity demand due to decarbonization more than offsets the smaller emissions reductions from less demand for other fuels, lowering the CAF from -0.12°C to -0.13°C. This finding reinforces an important aspect of our findings – as long as electricity decarbonizes at least as quickly as other fuels, our estimate of the CAF is an upper bound. We emphasize this fact in the manuscript for the exact reason that *a priori* there is a larger concern over the upper bound on the feedback effect underlying the CAF. Figure 3a and its caption now read (line 387; new text in red):

Figure R.12 (Figure 3a in the main text). The Climate Adaptation Feedback. (a) Solid green line and shaded green area show point estimates and 90% confidence intervals for the Climate Adaptation Feedback (in °C), for 2020-2099 under SSP2-RCP8.5 using our benchmark approach. The darker dashed line shows the dynamic CAF, which accounts for how additional climate change from adaptation feeds back into future adaptation, as detailed in Methods Section 5.4. The lighter dashed line shows the CAF recalculated using emissions factors that follow historical trends between 2000 and 2018, as detailed in Supplementary Information Section 3.

In the main text we discuss this new analysis by adding the following at the end of the second paragraph in the results section (lines 167-175; new text in red):

“...We also recalculate the CAF relaxing assumptions over fixed emissions factors. The last series in Figure 3a, the long dashed line green line, shows the CAF under the assumption that historical global trends in emissions intensity for both fuels between 2000 and 2018 occurs in all countries through 2099. Allowing each sector to continue to decarbonize at historical rates lowers the CAF to -0.13°C under SSP2-RCP8.5. As emissions from other fuels have remained relatively unchanged on a unit basis when compared to the substantial reduction in the emissions intensity of electricity generation over the past two decades, extrapolation of this trend lowers emissions from the electricity sector and amplifies the mechanisms leading to a negative CAF.”

• I am also not entirely happy with the choice of RCP8.5 as the baseline scenario. I think few people regard RCP8.5 as a plausible baseline anymore. It tends to lead to bigger climate impact projections, which might explain its continuing popularity in the literature, but developments in policy and technology make the most likely scenario of future emissions lower. RCP8.5 is outside even the 90% confidence interval of the RFF-SPs, I think. RCP4.5 would be the SSP that best fits a current policies scenario. If you want a higher emissions baseline, then I would not go beyond RCP7.0.

We thank the reviewer for highlighting that the choice of RCP8.5 as a central scenario is perhaps not a neutral one. In response, we have changed the abstract and results sections of the paper to report and equally emphasize the estimate of the CAF under *both* RCP8.5 and RCP4.5 baselines. The abstract now reads (lines 7-11; new text in red):

“...We find energy-based adaptation will decrease cumulative CO₂ emissions, lowering global mean surface temperature in 2099 by 0.12°C (0.07°C) relative to baseline projections under RCP8.5 (RCP4.5) and avoiding 1.8 (0.6) trillion USD (\$2019) in global damages over the next 80 years.”

Analogously, in the Results section we write (lines 145-149; new text in red):

“In 2099, under RCP8.5 (RCP4.5) the CAF is -0.12°C (-0.07°C); changes in energy consumption driven by adaptation lead to a 0.12°C (0.07°C) lower GMST relative to baseline. This projected reduction in warming in 2099 alone is equivalent to six (four) years of recent warming at the observed 0.018°C/yr rate between 1981-2019 (36) and is 25 times larger than that implied by a back-of-the-envelope calculation in ref. (8).”

along with (lines 152-156; new text in red):

“Using the Data-driven Spatial Climate Impact Model (DSCIM) built by the Climate Impact Lab, we estimate that the decrease in warming due to the CAF lowers the present value of cumulative damages from climate change between 2020 and 2099 by 1.8 (0.6) trillion (2019 USD) (see Methods Section 5.1).”

We note that in Figure 3, the CAF under RCP4.5 is shown for all four socio economic scenarios (the Shared Socioeconomic Pathways).

• My last comment on modeling choices relates to the comparison of the CAF with countries' NDCs. I am not sure this is currently set up as a fair comparison, because the NDCs prescribe emissions reductions in 2030, yet the CAF is quantified up to 2050. This makes the CAF look larger in relation to the NDCs than it probably is. However, I may have misunderstood how the underlying literature on the NDCs, which this paper relies on, deals with post-2030 emissions reductions. Yet, if this literature essentially freezes climate action in 2030, I would say my point stands.

We apologize for the confusion regarding the interpretation of the NDCs. We obtain estimates of committed emissions reductions under the NDCs from Meinhaussen et al. (2022). To the best of our knowledge, Meinhaussen et al. (2022) provides cumulative emissions goals for countries that have stated commitments *through* 2050. Specifically, Figure 2 of Meinhaussen et al. (2022) shows long-term targets for 76 countries through 2050. We show this figure as Figure R.13 below. These long-term targets are the ones we use in the analysis in Figure 4c.

Figure R.13: Replication of Meinhaussen et al. (2022) Figure 2. The GHG emissions per capita shown in the far right panel for 2050 are those used to assess the implications of the CAF for NDCs in the present manuscript.

To make this clear, we have amended the main text as follows (lines 245-250; edited text in red):

“Second, we find that the magnitude of adaptation-induced abatement is large relative to existing mitigation targets. To illustrate this, we report each country's cumulative adaptation-induced abatement in 2050 as a fraction of that country's cumulative required

abatement **in 2050** under its Nationally Determined Contribution (NDC) from the Paris Agreement of the United Nations Framework Convention on Climate Change (UNFCCC) (40; 41; 42).”

In Methods Section 5.2, we clarify this as follows (lines 558-567; edited text in red):

“Baseline country-level emissions and Nationally Determined Contributions

We obtain country-level baseline CO₂ emissions pathways and Nationally Determined Contributions (NDCs) from ref. (66). The repository is <https://zenodo.org/record/6383612#.Y4wnzC-B27c> and the version we use is dated February 14, 2022. Ref. (66) provide two sets of NDCs for most countries: a more stringent “conditional” NDC path (in the sense that the pathway is conditioned on action by other countries) and a “unconditional” NDC path, both available only under SSP1 and SSP5. We use the more stringent conditional NDCs along with baseline CO₂ emissions projections under the SSP5 scenario, **both for the year 2050**, to construct the ratio of cumulative adaptation-induced CO₂ emissions reduction over cumulative CO₂ emissions reduction under NDCs by 2050 in Figure 4c.”

• *The authors rightly steer the reader towards interpreting the results in terms of a ‘battle’ between declining emissions from space heating and increasing emissions from AC, yet technically the modeling includes all uses of energy in the economy outside transportation. The authors may want to square that circle by discussing more directly how heating/cooling drives the numbers. For example, it might be useful to report how much of non-transportation energy consumption goes to space heating and AC.*

We agree that these summary statistics might help frame our results more clearly. However, as the reviewer implies, our analysis of the demand responses of electricity and other fuels to temperature captures a broad scope of energy demand, including but not limited to indoor heating and cooling services. Because these demand responses encapsulate many other types of energy use, and cannot be separately decomposed into space heating and air conditioning (because of empirical constraints on our input data), we are concerned that presenting them may be misleading. For example, air conditioning accounted for about 10 percent of global electricity consumption (between 2 and 3 percent of final energy demand) in 2018, and space heating counted for half of energy used by the buildings sector (IEA 2018), translating to roughly 15 percent of total final energy demand across all sectors (IEA 2023). In contrast with this, the estimates in Rode e al. (2021) cover well over two thirds of final energy demand globally. We have changed the CAF section to make the difference between total electricity and/or other fuels demand, on the one hand, and cooling and/or heating on the other, more clear. The first paragraph in our short methodology section in the main text now reads (lines 75-77; new text in red):

“A higher frequency of realized hot days will increase **electricity demanded by cooling services** in locations that are already warm (Figure 1d), ...”

In the second paragraph describing the CAF, we now write (lines 85-95; new text in red):

“Similarly, energy-based adaptation in India and Brazil will lead to increased demand for **electricity due to more energy being required for refrigeration and indoor temperature**

control... However, if future declines in emissions due to reduced demand for heat from other fuels dominate the additional future emissions due to higher electricity demand, the CAF will be negative.”

We note that the manuscript does motivate our analysis by stating that (lines 21-23):

“Energy use is carbon-intensive: cooling demand alone comprised 10% of recent global electricity consumption and is expected to rise substantially during the 21st century (11).”

• *While cumulating discounted damages from 2020 to 2099 is a perfectly natural thing to do from an economic point of view, it is a little difficult to put the resulting \$1.8 trillion in context. Most people would think of comparing it with current GDP, but that is a flow and this is more like a change in a stock. Could you put this number in context?*

We thank the reviewer for this point and have added context for this number in the revised manuscript. The present value of these CAF damages is based on the cumulative discounted loss of consumption (due to the effects of climate change) in each year between 2020 and 2099. With a negative CAF, these values can be considered gains (avoided damages) each year that arise due to a lower level of anthropogenic climate change relative to the baseline warming that would occur absent the feedback captured by the CAF. After discounting the forecasted future gains into present dollar, we report the cumulative value of these gains as a share of global GDP in 2019 (a “flow”, as the reviewer refers to it). This amount captures the quantity of output we would need to give humanity in 2020 under a future where the CAF is absent to achieve the higher level of welfare that we predict will occur when including the gains from the CAF. In our central scenario, the avoided damages due to the CAF along the new temperature path lead to a welfare change over the 2020-2099 horizon equivalent to that which would be achieved by increasing global consumption in 2019 by about 2 percent. We integrate this comparison in the Discussion section to put the number in context. Specifically, we write (lines 274-276; edited text in red):

“This moderation of GMST change between 2020 and 2099 avoids 1.8 (0.6) trillion of damages (in 2019 USD) in present value terms, **an amount equivalent to roughly 2 percent of 2019 global output.**”

• *The fact that the CAF varies little across SSP scenarios is at first unintuitive, given how much uptake of AC is determined by income (see also Davis et al., 2021). It would be helpful to have a slightly deeper explanation of this result.*

We thank the reviewer for raising this question, which we have now addressed in the updated manuscript with an expanded discussion of the SSP-specific results.

The reviewer’s intuition is correct, in that income and population growth are important determinants of energy demand responses to a changing climate (Davis and Gertler, 2015; De Cian and Sue Wing 2019; Rode et al., 2021; Auffhammer, 2022), and that they can differ strongly across SSPs. However, most of the variation in population and income growth across SSPs occurs in low-income countries (Jones and O’Neill 2019), which exert far less influence over the CAF than do the large emitters like the Americas and Europe (as can be seen in Figure 4b in the main text). For instance, projections for global population in 2100 range between 6.9 and 12.6 billion across the five SSPs, with a standard deviation of 2.0 billion persons. For

higher income regions – Europe, the Americas, and China – the standard deviation of population in 2100 across SSPs are 72, 160, and 178 million respectively. In contrast with this, the standard deviation of the African population as a whole is 822 million, while that of Asia *ex*-China is 1.1 billion. Population variation in China, the Americas, and Europe explain only 16 percent of global population variation in 2100 (all calculations above based on Jones and O’Neill 2019).

In turn, these low income countries (despite comprising a large share of future population and income variation) account for a very limited share of future shifts in energy demand due to adaptation. On an emissions basis, China, North America, and Europe alone comprise over 50 percent of global GHG emissions (Ritchie, Rosado, and Roser, 2023). The remaining (relatively-low income) regions begin with very flat demand responses to climate change, as shown in Figure 1a. of Rode et al. (2021). Thus, while their income and population growth over the 2020-99 period change substantially across SSPs, they spend a majority of this period at income-levels where demand responses are low.

To make this clear to readers, we have expanded the discussion of the SSP-specific results as follows (lines 197-202; edited text in red):

“Although SSP scenarios change total population and levels of income across countries, which can shape the total energy response to daily temperatures (8), we find that within an RCP, global CAF values differ little across SSP scenarios. **This is likely due to the fact that income and population levels vary most across SSPs within low-income countries (37), and yet, as we show below, the CAF is driven primarily by adaptive responses in higher-income, high-emissions countries.**”

• *The quantification of the heterogeneity in adaptation-induced emissions changes across countries would be more convincing if it controlled for country size (normalizing by population or GDP) – at present this result seems quite strongly influenced by country size but size seems of secondary interest.*

We agree with the reviewer that the relationship between adaptation-induced abatement and cumulative CO2 emissions is heavily influenced by size, as larger populations tend to emit more. However, we view this correlation as important in the context of historical emissions and future emissions reduction targets. Large historical emitters have a differentiated responsibility under the UNFCCC regardless of population size, and as such we wanted to show that as free abatement is correlated with emissions at the national level, it should also be considered in the context of committed abatement. This insight shown in Figure 4b provides a link between country-level estimates of adaptation-induced abatement mapped in Figure 4a and the investigation of Nationally Determined Contributions explored in Figure 4c.

• *I would like the authors to consider whether the language in the final paragraph of the discussion exaggerates the extent to which current thinking and policy separates mitigation and adaptation. I do agree with this point to a certain extent, but I think equally any climate policy expert worth their salt would agree that (i) a good climate policy does both adaptation and mitigation and (ii) there are interactions between the two.*

We agree with the referee that this language was too strong. In the updated manuscript. we’ve edited the final paragraph of the discussion to read (lines 374-375; edited text in red):

“More broadly, a negative CAF **highlights the inherent link between** climate mitigation and adaptation **that is beginning to be employed in policy and research** (21).”

This edited text acknowledges there is at least a degree of common knowledge over this issue in policy circles, as well as in the scientific literature (e.g., reference 21 is Colelli et al., 2023).

Minor comments

- *Line 37: “incorporates” should be “incorporate”.*

We corrected this typo, thank you.

- *Line 226: “avoids 1.8 trillion [OF...] in present value terms”*

We corrected this typo, thank you.

- *Figure 1: perhaps you want to add an explanation of panels (e) and (f), for completeness?*

Thank you for this suggestion. We added explanations of each panel of Figure 1 in the caption and in the main text.

- *Line 347: I think ACC means anthropogenic climate change, but I didn't remember seeing it spelled out previously.*

We verify that ACC is spelled out before line 347.

- *Estimating the GMST-cumulative CO₂ relationship: in general, I liked this approach a lot, but I did wonder about the estimate of beta that is 1.4 times the median estimate of the TCRE. In particular, I wanted to get a handle on how the indirectly estimated contribution of non-CO₂ GHGs compares with other estimates.*

The intuition here is as follows. Carbon dioxide accounts for only 80% of cumulative increases in radiative forcing under RCP8.5. To attempt to account for the fact that reduced energy demand should also affect GHG emissions outside of CO₂, each modeled unit of CO₂ reduction should implicitly affect the course of global temperatures, due to affecting the remaining 20% of emissions of other GHGs. To calculate our analogue of the TCRE, we assume reductions in other greenhouse gasses are a linear function of reduced energy consumption that translates to lower CO₂ emissions.

A back-of-the-envelope calculation to compare our estimate with the TCRE in the scientific literature is as follows. Under RCP8.5, additional forcing due to anthropogenic emissions is 8.388 W/m² in 2100 in the MESSAGE climate modeling framework (Riahi et al., 2007). Of this, 6.592 W/m² are attributed to carbon dioxide emissions. The ratio of carbon dioxide's forcing to total forcing is 0.786. The median TCRE in the literature is 1.6°C (Matthews et al., 2018) per trillion tons of carbon (TtC) with a range of 0.8-2.4°C. If we take our estimated response to carbon (and associated emissions) of 2.23°C and attribute only a fraction of it (0.786) to carbon, this implies of the 2.23°C from all emissions, about 1.75°C is attributable to carbon emissions alone, a figure roughly in line with the median point estimate above.

References:

Auffhammer, Maximilian. "Climate Adaptive Response Estimation: Short and long run impacts of climate change on residential electricity and natural gas consumption." *Journal of Environmental Economics and Management* 114 (2022): 102669.

Colelli, Francesco Pietro, Johannes Emmerling, Giacomo Marangoni, Malcolm N. Mistry, and Enrica De Cian. "Increased energy use for adaptation significantly impacts mitigation pathways." *Nature Communications* 13, no. 1 (2022): 4964.

Colelli, Francesco Pietro, Ian Sue Wing, and Enrica De Cian. "Air-conditioning adoption and electricity demand highlight climate change mitigation–adaptation tradeoffs." *Scientific Reports* 13.1 (2023): 4413.

Davis, Lucas, Paul Gertler, Stephen Jarvis, and Catherine Wolfram. "Air conditioning and global inequality." *Global Environmental Change* 69 (2021): 102299.

De Cian, Enrica, and Ian Sue Wing. "Global energy consumption in a warming climate." *Environmental and resource economics* 72 (2019): 365-410.

IEA (2018), The Future of Cooling, IEA, Paris <https://www.iea.org/reports/the-future-of-cooling>, License: CC BY 4.0

IEA (2023), Buildings, IEA, Paris <https://www.iea.org/energy-system/buildings#>, License: CC BY 4.0

Jones, Bryan, and Brian C. O'Neill. "Spatially explicit global population scenarios consistent with the Shared Socioeconomic Pathways." *Environmental Research Letters* 11, no. 8 (2016): 084003.

Matthews, H. Damon, Kirsten Zickfeld, Reto Knutti, and Myles R. Allen. "Focus on cumulative emissions, global carbon budgets and the implications for climate mitigation targets." *Environmental Research Letters* 13, no. 1 (2018): 010201.

Meinshausen, Malte, Jared Lewis, Christophe McGlade, Johannes Gütschow, Zebedee Nicholls, Rebecca Burdon, Laura Cozzi, and Bernd Hackmann. "Realization of Paris Agreement pledges may limit warming just below 2 C." *Nature*. 604, no. 7905 (2022): 304-309.

Riahi, K., and Nakicenovic, N. (eds): 2007, Greenhouse Gases - Integrated Assessment, Technological Forecasting and Social Change, Special Issue, 74(7), September 2007, 234 pp. (ISSN 0040-1625)

Ritchie, Hannah, Pablo Rosado and Max Roser (2023) - "CO₂ and Greenhouse Gas Emissions" Published online at OurWorldInData.org. Retrieved from: <https://ourworldindata.org/co2-and-greenhouse-gas-emissions> [Online Resource]

Rode, Ashwin, Tamma Carleton, Michael Delgado, Michael Greenstone, Trevor Houser, Solomon Hsiang, Andrew Hultgren et al. "Estimating a social cost of carbon for global energy consumption." *Nature* 598, no. 7880 (2021): 308-314.

REVIEWER 1

We thank Reviewer 1 for their additional suggestions. In response, we reproduce each of your comments italicized in gray and provide our replies in black. Excerpts from the paper are indented with the line numbers noted. Figures and tables that are part of this document are prefixed with the letter “R” in order to distinguish them from figures in the paper.

I thank the authors for their detailed work and replies, which I consider satisfactory for the purpose of accepting the paper for publication.

I only have a few minor edits I would suggest, to enhance the clarity of some key points of the work:

- While the authors correctly specify that the spatial resolution of energy demand responses is much larger than in previous works (~25,000 subnational localities), I would find it useful if the authors included a brief explanation on the source of such heterogeneity, namely the possibility to identify locality-specific shocks thanks to the interaction coefficients between temperature exposure, income and average climate estimated in a single econometric equation by Rode et al.

We thank the reviewer for highlighting that the reader may benefit from a more detailed explanation of the identification strategy in Rode et al., (2021). We have added the following to the Methods section of the paper (lines 461-484 in Section 5.1, changes in red):

“As detailed in ref. (8), these energy demand response functions depend primarily on the realization of future daily temperatures within a given impact region, denoted by the vector $T_{p,t}^N$, under a given RCP scenario. The dose-response functions also include higher-order terms of daily grid cell-level temperature realization along with a set of covariates summarized by $X_{p,t}^N$, which include projections of GDP per capita and population specific to a SSP scenario and long-run averages of cooling and heating degree days under each temperature trajectory. These covariates allow for the response of energy consumption to daily temperature realizations to vary based on how the economic resources and climatology of a given location change in the future. This ensures the CAF calculation captures the extensive margin of adaptation that takes place over long periods and may be especially important in developing economies, where income growth is likely to lead to substantial increases in cooling and heating technology adoption (4; 8; 9). **The empirical estimation of such extensive margin effects is conducted in ref. (8) by interacting variables capturing short-run variation in weather with variables representing long-run variation in income and climate in a country-by-year panel regression. This model results in the estimation of heterogeneous dose-response functions in which energy demand sensitivity to daily realizations of temperature differs based on long-run average income and climate. The econometric procedure employed follows previously-developed methods (e.g., ref. (63)) to recover**

nonlinear energy demand responses to temperature at a *grid cell-by-day* level, which can be applied to predict climate change impacts at the scale of ~25,000 regions, even though more aggregated outcome data were employed (see Methods of ref. (8) for details). We show the quantitative importance of the extensive margin in shaping the CAF in Supplementary Information Section 2.”

- *The in depth case study on India is very interesting and useful. A key point that drives this works' result, stressed indeed by the authors in the SI - is:*

"This result is driven by a key finding of the ref. (1) paper, which is that for lower-income populations, electricity demand is essentially unresponsive to high temperatures, while other fuels (including biomass burning, among other sources) are used by such populations to respond to moderate to cold conditions. Even though India is projected to experience substantial economic growth under the SSP2 socioeconomic scenario, its relatively low level of baseline income inhibits a stronger electricity demand response (1)."

In fact, the authors note that the fossil fuels reductions they project are "about half of aggregate consumption today".

First, I think the above-mentioned comment provided by the authors could be moved to the discussion section, as it can help readers to understand the underlying dynamics driving the results.

We thank the reviewer for this suggestion. While we face space constraints in the main text, we have added a short version of this comment to the Discussion to highlight the importance of lower-income populations in driving the overall result of a negative CAF.

The first paragraph of our Discussion now reads (lines 271-286; edited text in red):

We develop a framework for quantifying the feedback between energy-based adaptation and anthropogenic climate change, a phenomenon we label the Climate Adaptation Feedback. Our methodology combines high-resolution projections of future energy consumption responses to climate change with country- and energy-specific CO₂ intensities to quantify cumulative emissions changes due to adaptation. Under several benchmark pathways for future emissions and socioeconomic development, we consistently find a negative CAF -- *i.e., that under climate change, declines in energy use from adapting to fewer cold temperatures overpower increases in energy use from adapting to warmer temperatures. This result is driven in part by the finding that much of the global population is projected to face income constraints that prevent substantial increases in energy demand in response to rising temperatures (8).* Our central estimate implies that adaptive energy use attenuates warming by 0.12°C in 2099, roughly equivalent to six years of warming at recent rates. This moderation of GMST change between 2020 and 2099 avoids 1.8 (0.6) trillion of damages (in 2019 USD) in present value terms, an amount equivalent to roughly 2 percent of 2019 global

output. When accounting for statistical and climatological uncertainty, our results suggest it is unlikely that the CAF is positive, limiting concerns that energy-based adaptation will exacerbate future warming.

Second, the specific result on India's fuel reductions suggests that a great amount of energy is already being used today to heat Indian homes - or for other heating-related energy uses. If possible, it would be very useful to add in the SI section a rough calculation/statistic of present-day fossil fuel consumption for heating services in the country (eg [sic] exploiting IEA or other institutional data), to put the projected reductions into perspective.

The reviewer is correct that this projected decline in energy demand in India under climate change is likely driven in large part by a decline in demand for heating and other energy-intensive activities that increase in frequency and/or intensity on cold days (e.g., indoor cooking and water heating). In India, this demand is met in many homes by the inefficient burning of biomass fuels, which is incredibly emissions-intensive (Navinya et al., 2023). This high emissions intensity renders declines in cold-day energy demand under climate change particularly impactful for the Climate Adaptation Feedback.

We agree with the reviewer that it would be helpful to put these projected declines in perspective of current energy demand in India. Unfortunately, to our knowledge, a comprehensive estimate of energy used for heating services is not available. However, data from the International Energy Agency (IEA) indicate that of the approximately 40 EJ of primary energy used in India, at least 7.5 EJ of this is biomass, an energy source used almost entirely for indoor heating and cooking (IEA 2021, Energy Information Administration (EIA) 2024). In comparison, the forecast reduction in consumption of other fuels (i.e., not electricity) by India in 2099 is about 7 EJ under our baseline SSP2-RCP8.5 scenario, composing over 90 percent of current use of biomass today, or around one third of primary energy consumption in the formal sectors of the economy (EIA 2024). Given that biomass is just one form of indoor heating and cooking, which also relies on coal, oil, and natural gas in India, these data suggest a reduction of 7 EJ is large but plausible. To help the reader put our estimated declines in perspective, we have added the following to the Supplementary Information to contextualize the magnitude of this reduction (SI lines 100-104, changes in red):

“The dashed line in Figure S5 shows that other fuels consumption is projected to fall by about 7 EJs by 2099. **This reduction is sizable, but also plausible, when compared to current Indian fossil fuel use outside the electricity sector. Of the approximately 40 EJ of primary energy use in India today, at least 7.5 EJ is biomass, an energy source used almost exclusively for indoor heating and cooking, activities that are sensitive to colder temperatures (7). The forecast annual reductions in other fuels consumption in 2099 due to adaptation are thus comparable to a decline in current biomass consumption levels by over 90 percent. Of course, other fuel sources – such as coal, oil, and natural gas – are also used for indoor heating and cooking in India, and will additionally contribute to declining other fuels consumption under warmer future climates.**”

References:

Energy Information Administration (2024). EIA. International Energy Database. <https://www.eia.gov/international/data/world>. Accessed May 6th, 2024.

IEA (2021), India Energy Outlook 2021, IEA, Paris, <https://www.iea.org/reports/india-energy-outlook-2021>, License: CC BY 4.0

Navinya, C., Kapoor, T. S., Anurag, G., Lokhande, P., Sharma, R., SV, L. P., ... & Phuleria, H. C. (2023). Heating and lighting: understanding overlooked energy-consumption activities in the Indian residential sector. *Environmental Research Communications*, 5(4), 045004.

REVIEWER 2

We thank Reviewer 2 for their additional comments. In response, we reproduce each of your comments italicized in gray and provide our replies in black. Excerpts from the paper are indented with the line numbers noted. Figures and tables that are part of this document are prefixed with the letter “R” in order to distinguish them from figures in the paper.

While I acknowledge the efforts made by the authors, I remain unconvinced by their response and by their results. In this paper, they simulate with an econometric approach how changes in temperature due to climate change may impact energy demand. They then use country-specific carbon intensities to translate these energy demand variations into carbon emission variations. My first remark is about the novelty of this approach, and what it adds to the literature. The authors explain that their work is novel because, compared to existing papers:

1. it assesses energy demand variations caused by changes in temperature with a much finer spatial resolution (25,000 subnational localities in 146 countries) and a much finer temperature resolution (energy demand response to the full distribution of daily temperature realizations, instead of commonly used heating-degree days and cooling-degree days)

2. it goes beyond estimating energy demand variations, and computes variations in GHG emissions

Regarding point 1, assessing an energy demand response with an extremely fine resolution does not necessary [sic] lead to a better assessment, because of the risk of over-fitting. This is especially concerning because the authors find conclusions opposite to the rest of the literature, and justify this difference by the fact that they precisely compute more response functions. The authors should try to find out why using their data leads to such results. They should try to do their analysis with temperature thresholds, see if the results are different, and explain why assessing an energy-demand response to the full distribution of daily temperature realizations could leads [sic] to better predictions than using the commonly used temperature thresholds. They should also assess the spatial heterogeneity of the response functions, and see for instance if some specific locations are driving the results. Simply saying that, because they have more data points and do many more regressions, they are more correct that [sic] other papers, is not correct.

We appreciate the reviewer highlighting that our prior response and manuscript edits provided insufficient detail and justification for the key methodological differences between our paper and prior work. It was therefore not sufficiently clear why our key findings differ from what is, to our knowledge, the only other existing estimate of the global warming effects of energy-based adaptation (Colelli et al. 2022) and why some aspects of our results differ from other related literature (De Cian and Sue Wing, 2019, Van Ruijven et al., 2019). In this response, we first clarify that there are four key differences between our methods and those from prior work, and provide additional details that were summarized only briefly in the last round. We then address the reviewer’s three specific concerns with the methodology we employ: overfitting, the use of

the full distribution of daily temperatures, and the degree of spatial heterogeneity in demand responses.

Methodological differences between our work and prior analyses

The econometric estimates of energy demand responses to climate change that we rely on come from Rode et al. (*Nature*, 2021), the most recent published estimates of global energy demand responses to climate change. Rode et al. find small aggregate *declines* in final energy consumption under future climate change, while prior global-scale estimates published by De Cian and Sue Wing 2019 (shorthand as DCSW for brevity) and Van Ruijven et al. 2019 (VR) estimate aggregate *increases* in final energy consumption under climate change. When using the Rode et al. results to conduct our calculation of the Climate Adaptation Feedback, we estimate a *negative* CAF. In contrast, when Colelli et al. (2022) use DCSW empirical estimates to conduct their calculation of a similar warming feedback, their results are consistent with a *positive* CAF.

There are 4 key methodological differences across these papers that may cause our central findings to diverge from Colelli et al. (2022). Two of these differences are discussed below in response to the reviewer’s specific concerns – the estimation of energy demand impacts in response to the full distribution of daily temperatures in the Rode et al. estimates (versus capturing only temperature extremes in prior work), and high spatial heterogeneity in the Rode et al. estimates (compared to just two global regions used in prior work). The other two differences are detailed directly here. We note that these four key methodological differences do not simply amount to having more data and/or running more regressions – they include fundamental differences in sample construction, econometric estimation, and projection (i.e., translation of econometric estimates into future estimates of climate change impacts) that have first order implications for core results. These four key differences are:

- 1. Differences in the characterization of temperature exposure (this is discussed below)**
- 2. Differences in spatial heterogeneity of energy demand responses (this is discussed below)**
- 3. Differences in sample construction and econometric estimation**

Both the construction of the sample and the econometric estimation approach in Rode et al. differ from DCSW and VR, with implications for key findings.

On sample construction, Rode et al. are focused on estimating aggregate energy demand responses, instead of characterizing fuel- and sector-specific responses, as in DCSW and VR. Thus, Rode et al. pool *all non-transport energy use* from across all sectors into just two categories: electricity and “other fuels”. This leads to an unbalanced panel dataset of 146 countries over 1970-2010. In contrast, DCSW and VR construct many unbalanced panel datasets with highly differentiated sample sizes for each fuel-by-sector (ranging from 6 to 49 countries) over the period 1971-2014. Rode et al. then estimate a *single* regression that uses interaction

terms to uncover heterogeneity in demand responses to temperature by fuel (electricity and “other fuels”), long-run average income, and long-run average climate. In contrast, DCSW and VR estimate *many* (24) independent regressions for each fuel-by-sector combination and for two geographic regions: temperate and tropical. As detailed in our last round of revisions, calculating the Climate Adaptation Feedback (CAF) requires estimates of *aggregate* energy consumption under climate change; and while fuel-by-sector relationships are critical for answering other policy-relevant questions, such disaggregation is not necessary for calculating the CAF.

These sample construction differences have implications for coverage that almost certainly drive material differences in resulting estimates. On the one hand, the estimating sample in DCSW contains 137 Exajoules (EJs) worth of non-transport final energy consumption in 2010 (DCSW, Table 1), while the Rode et al. sample contains 161 EJs of non-transport final energy consumption during the 2001-10 period, making the aggregate coverage of final energy demand more comprehensive in Rode et al. For example, DCSW and VR include electricity, gas and petroleum, but omit uses of coal outside of the electricity sector. Because over 20% of coal use is estimated to be generated for thermal purposes outside of the power sector (IEA, 2022), any temperature-sensitive energy use in non-electricity coal use is omitted in the DCSW and VR empirical estimates. On the other hand, the sample in Rode et al. does not include transport energy use, while DCSW and VR do – we clearly state in our manuscript (see Discussion section) that the omission of transportation is a limitation of our estimate of the CAF. However, transportation energy use is projected to increase just 11% under climate change in DCSW, and changes are detected only in tropical regions, making it unlikely that this omission in Rode et al. (and therefore our estimates) is first-order in driving the divergence in findings.

These sample construction differences also have implications for econometric estimation. As discussed in detail below, pooling of data across sectors and fuels in Rode et al. allows the authors to tractably characterize much richer geographic heterogeneity in energy demand responses to temperature (using a sample of size $N = 7,563$) without overfitting the data or losing statistical power (discussed below). In contrast, sector-by-fuel-by-region specific regressions in DCSW and VR are estimated on sample sizes ranging from $N = 74$ to $N = 1,560$ (only 3 of 12 sector-fuel-region combinations with sufficient data for estimation have over 1,000 observations), limiting the scope for additional heterogeneity in energy demand estimation. Other econometric differences include controlling for precipitation (Rode et al.) versus humidity (DCSW and VR) and the use of an interaction model (Rode et al.) versus an error correction model (DCSW and VR) to characterize the extensive margin of adaptation to long-run climate change. It is unclear exactly how these differences may translate into differences in final estimates of energy demand under climate change, but given other key benefits of the Rode et al. approach discussed both above and below – improved geographic heterogeneity, pooling across sectors and fuels for increased coverage and statistical power, capturing effects of the full temperature distribution – we view these estimates as the best available for calculating the CAF. Moreover, there is no reason to believe that these controls or econometric specifications would systematically lead to the lower estimates of energy demand under climate change found in Rode et al. versus DCSW and VR.

4. Differences in projection of future energy demand responses to climate change

Rode et al., DCSW, VR, and Colelli et al. (2022) all use the econometric estimates discussed above to project future energy demand under climate change, leveraging climate model projections and simulations of future incomes and population. There are multiple differences in how this projection is conducted in Rode et al. versus these earlier works that contribute to differences in key findings.

First, in contrast with Rode et al.'s estimates, the future projections in DCSW/VR/Colelli do not include the response of fuels in sectors and/or regions for which estimated elasticities are statistically insignificant (from our reading of DCSW Table 3 and the text associated with footnote 12). This sets 25 of the 60 regional fuel-by-sector-by-temperature bin responses to future global warming to zero in DCSW's projections (DCSW, Table 3) in addition to the 12 that are not included due to data limitations. In contrast, Rode et al. (2021) uses the full set of estimated coefficients and variance-covariance matrix of their pooled regression – which includes heterogeneity in demand responses by fuel, long-run average income, and long-run average climate – to construct confidence intervals around future projected impacts that account both for climatological *and* statistical uncertainty (including covarying uncertainty across fuels, temperatures, and incomes). Omitting insignificant responses implicitly assumes that failure to reject the null (i.e., high p -values against a zero null) is equivalent to accepting the null hypothesis of a zero effect (Imbens, 2021). If conducted in-sample, this approach would lead the average in-sample prediction of the econometric model to diverge from the average of the dependent variable in the model. For out-of-sample predictions, as in the case of climate change projections, this approach fails to compute an accurate mean point estimate or confidence intervals of projected energy demand under climate change.

The omission of statistically insignificant coefficients when computing climate change impact projections has first-order implications for key findings. Of the 25 temperature-day response coefficients dropped by DCSW, 13 of them are cold-day effects and 16 of them are non-electricity effects (DCSW, Table 3). As cold-day responses tend to be dominated by non-electricity sources of fuel (e.g., Auffhammer, 2022; Deschenes and Greenstone, 2011), such omissions imply the authors systematically underestimate energy savings realized due to fewer cold days under climate change, as many of these effects are assumed to be zero instead of statistically imprecise estimated point estimates whose uncertainty could be propagated through to final impact projections. Such energy savings have been shown to be substantial in prior work, even excluding the Rode et al. study (e.g., Auffhammer, 2022; Deschenes and Greenstone, 2011; Wenz et al., 2017).

Finally, the translation of energy demand projections under climate change into changes in *emissions* (and, ultimately, to warming) is also different in Rode et al. versus Colelli et al. First, as mentioned in the main text, we compile data on *country-specific* emissions factors that allows us to characterize how different fuels differentially impact warming based on the efficiency or inefficiency of each country's energy sector. In contrast, Colelli et al. use the WITCH integrated assessment model (Emmerling et al. 2016), which produces at most 17 regional estimates of fuel-

specific emissions intensities. Given the wide variation in emissions factors in our data (Figure 2 in the main manuscript), this distinction can have first order implications for resulting estimates of emissions and thus warming due to energy-based adaptation.

Second, aside from differences in econometric inputs, Colelli et al. (2022) represents a distinct methodology for computing emissions due to future adaptation – this paper uses a computable general equilibrium (CGE) model (also called an Integrated Assessment Model or IAM in this setting), while we employ a statistically-based approach. Because of the many differences in CGE and statistics-based approaches (e.g., see Piontek et al., 2021 for a relevant review), identifying what drives final differences in results is difficult. Even if the structural parameters used to calibrate Colelli et al. (2022) were consistent with those estimated in Rode et al. (2021), small differences in structural parameters (e.g., temperature-demand elasticities) can be amplified considerably by choices within the CGE model that compound over long horizons. An example of this specific to the CGE used in Colelli et al. (2022) is how the modeled decision maker produces energy as an input for a final good. In the WITCH model used by Colelli et al., the decision maker combines electricity and other fuels to create a final composite energy good. The degree to which the two fuel types can replace each other is governed by an elasticity of substitution. As this elasticity is less than unity (Emmerling et al. 2019 – section 3.2.4), lower productivity in either sector (i.e., “damage” to the sectoral productivity for a given fuel) mechanically reduces demand for all other fuels as well when holding costs constant. Parameters governing substitutability are tantamount in determining short and medium-run responses of energy consumption to efficiency changes in CGEs (Olavson, Hassler, and Krusell 2021; Casey 2023). There is no analogue to this result from a planner’s optimization problem in our approach, where empirical response functions from Rode et al. (2021) are used directly to forecast future energy consumption. We view the approach from the CGE as highly complementary to the statistically-based approach we take here – the key advantage of our methodology is that it derives directly from historical demand responses and is, by construction, not sensitive to decisions over outside parameters imposed on CGE models that govern production and technology. Of course, the tradeoff is that CGEs can be used to construct counterfactual pathways in which future behavior and technology differ substantially from what has been historically observed.

While we do not have space within the main text to detail all of these distinctions from the prior literature, we have amended the text to reflect more accurately and precisely how our estimate of the CAF, based on the empirical energy demand projections of Rode et al. (2021), is fundamentally different from, and in general complementary to, prior work. Lines 41-52 of the introduction now read (changes in red):

“While our work is not the first to note the potential feedback between energy used for adaptation and the climate (16; 20; 21), it formalizes the notion of the CAF and provides a tractable framework for its calculation. Prior work has used a regional Integrated Assessment Model (IAM) to estimate the effects of energy-based adaptation on global warming (22). Through the lens of an IAM, the feedback in ref. (22) is modeled by allowing changing frequencies of regional temperature

extremes to affect the productivity level of energy inputs in economic production. We instead adopt a data-driven approach that, relative to prior work, increases the spatial resolution of energy demand responses used to forecast changes in emissions, includes the response to the full distribution of daily temperature realizations rather than restricting inputs to realizations of local extremes, and accounts for country-level heterogeneity in emissions intensities, as well as their dynamics, for both electricity and other fuels.”

Concerns with overfitting

The empirical specification in Rode et al. (2021) uses a panel data fixed effects model with interactions between short-run variation in temperature and long-run variation in income and climatology to characterize spatially and temporally heterogeneous energy demand responses. Because such a model requires estimating many interaction terms, the reviewer is rightly concerned that the resulting heterogeneity may reflect overfitting to the estimation sample, instead of true heterogeneity in the population of interest. Before directly examining this possibility below, we note that Rode et al.’s larger sample, which is made possible due to pooling energy demand across fuels and sectors, as discussed above, allows for more statistical power than prior literature, which has estimated fuel-by-sector-by-region specific demand responses. Thus, while the authors of prior work (DCSW; VR; Colelli et al., 2022) discuss power limitations as limiting the amount of heterogeneity that can feasibly be estimated, such constraints are less binding with the Rode et al. sample (of course with the tradeoff that Rode et al. estimate responses that aggregate demand across fuels and sectors).

We examine the potential for overfitting in Rode et al. by systematically evaluating how the model performs when predicting energy consumption on out-of-sample data. When overfitting occurs, a model fits well on the estimation sample, but fails to accurately characterize the outcome variable when used to predict on a held-out test set (Everitt and Skrondal, 2010). Specifically, we run many iterations of a k -fold cross-validation exercise that evaluates the out-of-sample fit of the empirical model in Rode et al. (2021). We compare this out-of-sample fit to that recovered from more restrictive models nested within the Rode et al. specification (i.e., where interaction terms that characterize heterogeneity in energy demand responses to temperature are progressively removed). In each iteration, we divide the sample into 10 equally-sized subsets, estimate the model on 9/10 of the full sample, and calculate the model’s out-of-sample predictions errors for the remaining 1/10 of the data. We do this for each possible composition of the 9/10 subsample, yielding 10 values for out-of-sample Root Mean Squared Error (RMSE) in each iteration of the cross validation procured. For each k -fold iteration, we record the average out-of-sample prediction error (measured as RMSE) over the 10 partitions. We then repeat this exercise 100 times for each model, so as to average out-of-sample error over variation in how to partition the data into 10 subsets.

If the Rode et al. empirical model were overfitting the data sample, we would expect to see substantially higher RMSE values in this exercise for the baseline Rode et al. interaction model,

as compared to resulting RMSE values from alternative models that omit parameters characterizing heterogeneity in the energy demand-temperature response.

Table R.1 presents the results and shows that this is not the case. Specifically, Table R.1 shows the average Root Mean Squared Error (RMSE) for each model on the held-out datasets across the 100 cross-validation iterations. Covariate columns indicate whether income was included as a control and whether income and/or climate were interacted with daily realizations of temperature (as they are in Rode et al.). For both electricity and other fuels, the mean out-of-sample RMSE from the fully interacted Rode et al. (2021) specification is very similar to models with lower complexity. Differences in RMSE across specifications are very low (maximum 2% improvement from the Rode et al. baseline model), even when compared to highly restricted models that remove all interactions between temperature and other covariates (last two rows). We obtain similar results when we use a different number of folds and the ranges across bootstrap runs (shown as the 95% CI) are small.

In conclusion, we do not find any evidence that the interaction terms, which enable a rich characterization of heterogeneity in energy demand responses to temperature, result in overfitting.

Model	Fuel	Covariates			RMSE	
		Income	Temp-Income Interaction	Temp-Climate Interaction	Mean	95% CI
Baseline Rode Model	Electricity	Yes	Yes	Yes	0.218	[0.216, 0.218]
No Income Interaction	Electricity	Yes	No	Yes	0.217	[0.216, 0.217]
No Climate Interaction	Electricity	Yes	Yes	No	0.217	[0.216, 0.217]
No Interactions (I)	Electricity	Yes	No	No	0.217	[0.216, 0.217]
No Interactions (II)	Electricity	No	No	No	0.225	[0.224, 0.225]
Baseline Rode Model	OtherFuels	Yes	Yes	Yes	0.230	[0.229, 0.230]
No Income Interaction	OtherFuels	Yes	No	Yes	0.228	[0.227, 0.228]
No Climate Interaction	OtherFuels	Yes	Yes	No	0.229	[0.228, 0.229]
No Interactions (I)	OtherFuels	Yes	No	No	0.228	[0.227, 0.228]
No Interactions (II)	OtherFuels	No	No	No	0.227	[0.226, 0.227]

Table R.1: Results from a k -fold cross validation exercise examining the out-of-sample fit of the specification in Rode et al. (2021) (top row), as compared to models with lower complexity (bottom four rows). Model specifications are ordered from highest (top) to lowest (bottom) in terms of their complexity in characterizing heterogeneity in energy-temperature demand responses across countries. Mean Root Mean Squared Error (RMSE) is the average out-of-sample RMSE across each of the 10 folds and across 100 bootstrap iterations in which each bootstrap run randomly splits the data into “train” and “test” samples. The 95% CI column reflect the 5-95 confidence interval constructed using the sample variance of the RMSE across the 100 bootstrap iterations for each fuel-model pair.

Concerns with estimating energy demand responses to the full distribution of temperature realizations

The reviewer is additionally concerned that the use of temperature realizations across the entire distribution, rather than counts of numbers of days above or below a certain temperature threshold, may be responsible for differences in key findings across Rode et al. and prior work. We agree that this is highly likely; here we document the critical importance of accounting for the full distribution of temperatures, instead of only quantifying the effects of extreme days on energy demand.

In DSCW and VR, daily temperature realizations only impact energy demand if they reach above 27.5°C or fall below 12.5°C. Thus, changes in future local temperature distributions affect energy demand only insofar as they affect the cumulative annual number of days above 27.5°C or below 12.5°C in that location. In contrast, Rode et al. (2021) empirically estimate and then project into the future how demand responds to the full distribution of daily temperature realizations.

To demonstrate the importance of this distinction, Figure R.1 shows that a substantial share of global daily temperature realizations at the grid-cell level fall into the category of temperatures omitted by DSCW and VR, and that these “moderate” temperatures are dramatically impacted by future climate change. Specifically, the figure shows: a) how much of the current global distribution of population-weighted average daily temperatures at the grid-cell level falls within the [12.5°C, 27.5°C] interval; and b) how many of these days are influenced by future climate change. The figure compares the 2006 distribution with that projected for 2100 under RCP8.5 using the ACCESS1-3 model (Copernicus, 2021). In 2006, daily grid-cell level temperature readings were inside the 12.5°C and 27.5°C degree interval on 66.0% of days on a population-weighted basis. In 2100, this interval is forecast to contain only 41.5% of these grid cell-day observations. Thus, estimates in DCSW and VR omit a substantial shift in exposure to moderate temperatures under climate change, omitting energy demand impacts of temperatures that account for over 65 percent of people-days globally, and almost 50 percent of people-days in places that comprise a large share of energy demand such as the United States and the People’s Republic of China.

Figure R.1: Population-weighted distribution of grid-cell level daily surface temperatures across 160 countries. Blue bars show historical realizations in 2006 while red bars show projections under RCP8.5 from the ACCESS1-3 model in the year 2100 (Copernicus 2021). The 160 countries are drawn from the shapefiles for each ISO-3 code taken from the GADM database (<https://gadm.org>).

Such temperature realizations are a particularly critical omission because these moderate temperatures have meaningful impacts on final energy demand. While such effects are recovered directly in Rode et al. (e.g., see Figure 1), a large body of prior work also indicates that moderate temperatures substantially influence energy demand. For example, Figure R.2 is replicated from Wenz et al. (2017) and shows electricity demand responses to daily temperatures across many European countries. These highly convex responses indicate that there is substantial demand response to temperature realizations within the moderate temperature range omitted by DSCW and VR. Similar findings have been uncovered in other studies (e.g., Deschenes and Greenstone, 2011; Auffhammer et al., 2017). Moreover, the indicator approach used by DSCW and VR imposes the assumption that any temperature above or below the extreme temperature cutoffs of 12.5°C and 27.5°C has the same marginal effect on energy demand; the convexity of responses shown in Figure R.2 and recovered in many prior studies indicates that this approach underestimates true energy demand responses, which continue to rise at temperatures beyond the 12.5°C and 27.5°C thresholds.

Figure R.2 (Figure 1 in Wenz et al., 2017): Estimated electricity demand responses to daily temperatures across the European Union by Wenz et al. (2017). Dose-response functions were estimated independently for each country.

Based on this analysis and review of the literature, we view the inclusion of the full distribution of daily temperatures as critical to constructing an accurate assessment of the energy demand implications of climate change, and thus to an accurate calculation of the CAF.

Concerns regarding reliable estimation of spatial heterogeneity in energy demand responses to temperature variation

The reviewer is correct that the specification in Rode et al. (2021) allows for demand responses to vary across locations and time due to differences in income levels and local climate. This results in a substantial degree of heterogeneity in the shape and magnitude of the temperature response functions both across space and over time, as discussed in detail above. This is illustrated in Figure 1c. of the original Rode et al. (2021) paper replicated here in Figure R.3 below.

Figure R.3: Replication of Rode et al. (2021) Figure 1.c, which displays how energy consumption response functions to temperature realizations depend on average income and baseline local climate.

The reviewer asks that we “*assess the spatial heterogeneity of the response functions, and see for instance if some specific locations are driving the results*”. The spatial heterogeneity of the estimated response functions is plotted, discussed, and detailed in Rode et al. This heterogeneity marks a key contribution of the original Rode et al. paper, as the authors find that strong differences in energy demand sensitivity across rich and poor nations contribute to finding a net decline in energy consumption under climate change (as electricity increases on hot days in poor countries are constrained by low incomes). We note again that there is no evidence of overfitting in this empirical model and that pooling energy demand across fuels and sectors allows Rode et al. to use a much larger sample size when estimating this heterogeneity than was possible in prior work.

Heterogeneity in the projected impacts of climate change on consumption of electricity and other fuels is shown in Figure R.4 below, a replication of Figure 2.a from Rode et al. (2021). This figure shows that while there are substantial differences across space, no individual country or region emerges as an extreme outlier driving global results in one direction or another.

Figure R.4: Replication of Rode et al. (2021) Figure 2.a, which displays the projected impact of climate change on per capita consumption of electricity and other fuels in 2099 under RCP8.5 emissions.

While Figure R.4 suggests no individual region is driving overall global estimates of energy consumption under climate change, we conduct an analysis to specifically evaluate whether individual locations are driving our final CAF result. To do so, we recalculate the cumulative global change in emissions due to adaptive energy use repeatedly, each time removing the emissions effects of adaptive energy use from one country before re-computing global emissions changes. Figure R.5 shows the result, plotting the distribution of changes in cumulative emissions in 2099 under our baseline SSP2-RCP8.5 scenario over simulations in which each of the 142 countries in our sample are removed. The dashed line is our baseline estimate, which includes all 142 countries. This figure shows that all leave-one-out estimates of cumulative emissions in 2099 are well below zero and differ from the full-country estimate by at most 20%. Thus, no single country is driving our estimate of the CAF, and in particular no single country is responsible for generating a negative CAF.

Figure R.5 (Figure S7 in updated SI): Distribution of global emissions changes due to adaptive energy use in 2099 over 142 simulations, each of which removes the adaptive emissions from one country. The full-country estimate is indicated by the dashed line. Results shown are from the baseline SSP2-RCP8.5 scenario.

To demonstrate the robustness of the CAF to this leave-one-out exercise, we have added Figure R.5 to the Supplementary Information as Figure S7 and we reference it in the main text as follows (lines 149-158 edited text in red):

“Under the SSP2-RCP8.5 scenario, our estimated CAF implies that adaptive energy consumption is predicted to lower the change in GMST in 2099 from 4.27°C to 4.15°C, relative to the pre-industrial climate. Using the Data-driven Spatial Climate Impact Model (DSCIM) built by the Climate Impact Lab, we estimate that the decrease in warming due to the CAF lowers the present value of cumulative damages from climate change between 2020 and 2099 by 1.8 (0.6) trillion (2019 USD) (see Methods Section 5.1). Accounting for both climatological and statistical uncertainty in 2099 yields a 90% confidence interval for the CAF of -0.35 to 0.073°C, and the CAF is robust to iteratively leaving out the adaptation of individual countries when forecasting cumulative emissions reductions in 2099 (Supplementary Information Section 5).”

This new Supplementary Information Section 5 reads (lines 134-147; edited text in red):

“Robustness of the Climate Adaptation Feedback to the omission of individual countries

While the analysis in ref. (8) suggests no individual region is driving overall global estimates of energy consumption under climate change, we conduct an analysis to specifically evaluate whether individual locations are driving our final CAF result. To do so, we recalculate the cumulative global change in emissions due to adaptive energy use repeatedly, each time removing the emissions effects of adaptive energy use from one country before re-computing global emissions changes. Figure M1 shows the result, plotting the distribution of changes in cumulative emissions in 2099 under our baseline SSP2-RCP8.5 scenario over simulations in which each of the 142 countries in our sample are removed. The dashed line is our baseline estimate (an emissions reduction of 195 GTCO₂), which includes all 142 countries featured in our main analysis. This figure shows that all leave-one-out estimates of cumulative emissions in 2099 are well below zero and differ from the full-country estimate by at most 20%.”

Regarding point 2, the authors use an extremely crude, and, to my sense, misleading, approach to translate energy demand variations into GHG emissions variations. Compared to the first version of the manuscript, the authors do not suppose any more that countries emission factors will remain constant over the XXIst century. However, they now suppose that countries [sic] emission factors will evolve following the current average global trend in emission intensity. Such a scenario does not make any sense to me: the global trend in emission intensities hides a wide variety of changes at each country level.

In response to the reviewer’s prior comments, we have added a sensitivity analysis in which these emissions factors evolve based on a global average trend in emissions intensities (this is the lime green line in the main manuscript Figure 3). Here we reply to the reviewer’s concerns regarding this sensitivity analysis.

The reviewer is correct in that there is substantial variation in the rate of change in emissions intensities across countries for both electricity and other fuels. To elucidate this, here we re-estimate Equation (1) from the Supplementary Information,

$$\Delta \ln F_{i,t}^h = \gamma_i^h + \varepsilon_{i,h,t} \quad (1')$$

at the *country* level for each country in our data between 2000 and 2018. In Equation (1'), $\Delta \ln F_{i,t}^h$ is the change in the (log) emissions factor for fuel h in country i from the previous year, γ_i^h are country- and fuel-specific decay coefficients, and $\varepsilon_{i,h,t}$ are country-specific error terms. Figure R.6 below shows kernel density functions for the point estimates of decay coefficients $\hat{\gamma}_i^h$ for both fuels across the 140 countries in our sample with sufficient data on which to estimate these trends.. As suggested by the reviewer, this figure reveals substantial dispersion in country-level trends in the emissions intensity of energy consumption. Notably, for both energy sources quite a few coefficients are positive; of the country-level estimates of $\hat{\gamma}_i^h$ for other fuels (electricity), 75 (47) are greater than zero, indicating a positive trend in emission factors over time for several countries.

Figure R.6: Kernel density of country-level emissions intensity decay coefficients for electricity and other fuels between 2000 and 2018. Rates of change in emissions intensities are calculated using data from the International Energy Agency and estimated via Equation (1').

Despite this large heterogeneity, we show below that accounting for country-specific emissions decay rates does not meaningfully impact the CAF or the key findings in our paper. Specifically, we recalculate the CAF implementing country-level trends in emissions factors as an additional

sensitivity exercise. Doing this requires a slight transformation of the estimates in Figure R.6 above due to the large share of positive coefficients. To avoid the scenario in which positive coefficients leading to exponential growth in emissions intensities (a scenario we view as implausible), we censor values of $\hat{\gamma}_i^h$ to be nonpositive. We then recalculate a future pathway for each country-by-fuel specific emissions intensity by recalculating Equation (2) in the Supplementary Information,

$$F_{i,t}^h = \exp \left\{ \hat{\gamma}_i^h (t - 2020) \right\} \bar{F}_i^h \quad (2')$$

and replacing the global trends for each fuel with country-specific ones. In equation (2'), like in the global case, \bar{F}_i^h denotes the original 2010-18 emissions factors calculated based on country-level data between 2010-18 and used for the baseline CAF.

Figure R.7 below shows how allowing for heterogeneous country-by-fuel trends in emissions intensity affects the CAF under our baseline SSP2-RCP8.5 scenario. Our point estimate of the CAF in 2099 falls to 0.08°C from our original estimate of 0.12°C (the dashed gray line). The point estimate remains negative for all SSP-RCP combinations we consider. Thus, while the data do indicate heterogeneity in countries' temporal trends in emissions intensities, accounting for such heterogeneity based on historical time trends does not change the key findings of our analysis.

Figure R.7: Global CAF under an SSP2-RCP8.5 scenario allowing for country-by-fuel specific temporal trends in emissions factors. The temporal trends in emissions factors used here are projected based on historically estimated trends

in emissions factors for each country and fuel over the 2000-2018 period using data from the IEA. Any positive trends were censored to zero. Dashed gray line indicates our baseline estimate of the CAF with emissions factors fixed at 2010-2018 averages.

Moreover, even if the authors were using country-specific trends, it would be strange to rely on such trends. Most countries have indeed already begun to implement deep changes in their electricity production technologies, for instance, so current trends are not particularly representative of what is expected over the next decades.

The reviewer is correct in that if historical data (current trends) are not representative of future trends in emissions factors, our extrapolation of these trends will generate an inaccurate estimate of the CAF. However, the reviewer's concerns generate a clear direction of such bias – if deep changes in electricity production technologies do in fact unfold more rapidly than current trends indicate, the CAF would be *even more negative* than that reported in our paper. As adaptive electricity use raises the CAF, while adaptive use of other fuels lowers it (see Figure 3b in the main text), rapid decarbonization of electricity sectors would ultimately *lower* the CAF relative to our benchmark estimate. Thus, our current results serve as a conservative upper bound on the CAF if future decarbonization proceeds faster than historical trends indicate. We emphasize this aspect of our CAF estimation in the text (lines 310-321 in the manuscript) in part because *a priori* there is a larger concern over the CAF's upper bound – that is, the potential for a positive CAF is concerning because it implies that adaptation may accelerate future warming:

“The use of a fixed SSP-RCP baseline prohibits us from examining the CAF in tandem with decarbonization scenarios more sophisticated than extrapolating historical trends. Any decarbonization assumptions we might employ to alter current CO₂ intensities of energy consumption into the future would be inconsistent with modeling assumptions built into these exogenous scenarios; a fully coupled approach in which behavioral adaptations are built directly into climate and socioeconomic modeling would be required to comprehensively assess the implications of decarbonization for the CAF. However, as long as CO₂ intensities associated with other fuels do not decline dramatically relative to those for electricity, our estimate should serve as an upper bound on the magnitude of the CAF. For example, if the electricity sector continues to decarbonize faster than other fuels (45), there will be fewer additional emissions from increased electricity consumption to offset the decreased emissions from other fuels, implying a more negative CAF than the value we have uncovered here.”

However, to explore whether our qualitative result of a negative CAF is robust to relaxing assumptions on the pathways for carbon intensity, we conduct a Monte Carlo (MC) experiment that explores a wide range of possible declines in emissions factors over time. To do this, we recalculate the CAF after randomly assigning each country a fuel-specific rate of emissions intensity decay drawn from a uniform distribution with support between the largest decay rates (in magnitude) that we estimate in the historical data and zero. Specifically, for each Monte Carlo iteration, we draw a new fuel-specific trend coefficient for each country, ranging from the smallest emissions factor trend observed for each fuel in the historical data (-0.12 and -0.31 for

other fuels and electricity, respectively) to zero, and recalculate the resulting CAF for our baseline SSP2-RCP8.5 scenario. We bound these decay rates to be nonpositive so as to avoid the issues with exponential growth in emissions described above (we see this constraint as in line with the reviewer's concern about rapid technological change that would lower emissions intensities).

Figure R.8 below shows the distribution of the resulting 1,000 CAF simulations. Each draw is a value of the CAF under our baseline SSP2-RCP8.5 scenario in 2099, but recalculated using different simulated paths for emissions intensities at the country-by-fuel level. The MC estimates for the CAF are smaller in magnitude than those using country-specific or global trends, in part because this random assignment of decay factors omits critical covariances observed in the data, such as the fact that large countries with high total emissions (e.g., India and China) tend to have low decay factors relative to the rest of the globe. Nevertheless, all 1,000 CAF estimates generated by the MC exercise are negative. We believe this should assuage concerns that our key qualitative result of a negative CAF is driven by our assumptions surrounding the dynamics of emissions intensity from each fuel over time.

Figure R.8: Histogram of CAFs in year 2099 under SSP2-RCP8.5 recalculated for 1,000 random draws of emissions intensity decay factors for each fuel at the country level.

The authors state that there are "no existing country- specific projections of future emissions factors up to 2100". This assumption is partly false, as this is specifically what Integrated Assessment models are providing (see for instance <https://unfccc.int/topics/mitigation/workstreams/response-measures/modelling-tools-to-assess-the-impact-of-the-implementation-of-response-measures/integrated-assessment-models-iams-and-energy-environment-economy-e3-models> or <https://www.carbonbrief.org/qa-how-integrated-assessment-models-are-used-to-study-climate-change/>). This is for instance what is done in ref.

22 of the paper, in which "fuel and technology choices are optimized intertemporally, under a set of constraints, including carbon and other energy prices".

We thank the reviewer for highlighting that the outputs from some Integrated Assessment Models (IAMs) include implied or explicit pathways for emissions intensities across different forms of both primary and final energy use. We apologize for failing to clarify why such projections are either inappropriate for our CAF exercise or are close analogues to our sensitivity analysis where emissions factors decline over time. Here, we describe how relevant IAMs generate emissions intensity projections in one of two ways; describe why each of these two methods generates outputs that are not suitable for our CAF calculation; and show how we have adjusted the paper's text to make this clear to readers.

Relevant IAMs have been developed both in the climate macroeconomics literature (e.g., William Nordhaus's Dynamic Integrated Model of Climate and the Economy (DICE) framework) as well as some by interdisciplinary teams (e.g., the WITCH model by Emmerling et al. (2016), which the reviewer mentions above and is used in Colelli et al., 2022). These models fall into one of two categories with respect to their treatment of emissions intensity of energy use:

1. **Some IAMs assume an exogenous exponential decay in emissions intensity, very similar to the one we adopt in our manuscript.** Exogenous exponential decay at a globally homogeneous rate is a common assumption in the macroeconomics literature. For example, Barrage and Nordhaus (2024) assume global emissions per unit output declines over time at an exogenous globally-shared rate. Krusell and Smith (2022) assume an exogenous rate of decline in emissions per unit of energy used for a roughly 19,000 region IAM. Golosov et al. (2014) assume that the productivity of energy input in carbon-equivalents rises over time exogenously, resulting in a constant global rate of exponential decline for carbon inputs required to produce a unit of energy as output. Cruz and Rossi-Hansberg (2024) assume that the productivity of energy input in carbon-equivalents rises over time endogenously, but only as a function of local productivity levels which are determined by firms that do not account for emissions in this decision in the absence of taxation. While assumptions of fixed or exogenous rates of changes in emissions factors may be simple, such assumptions are ubiquitous in the economics literature.
2. **Some IAMs combine an exogenous exponential decay in emissions intensity with endogenous emissions abatement decisions made by global or regional social planners choosing optimal climate mitigation pathways.** This combination of exogenous and endogenous emissions intensity is common in both the macroeconomics literature and in IAMs created by interdisciplinary teams that more intricately characterize the energy sector. In DICE and in WITCH, exogenous emissions intensity assumptions are combined with a global social planner that can control both output each period (and in turn gross emissions of GHGs from production) as well as a level of emissions abatement. Each is a choice variable for a social planner that affects the observed net emissions per unit of energy used after the model has been run. Additionally, exogenous rates of technological change that affect energy efficiency are

also imposed. For example, in the WITCH model, exogenous increases in the productivity of energy inputs are calibrated using exponential trends that are targeted to roughly match historical trends globally:

“10.1.2 Energy Intensity

The dynamic calibration of factor productivity of energy services (“tfpn”) is run based on the SSP2. The following income elasticity rule is used for the different regions: industrialized countries (OECD members) are characterized by an elasticity of 0.40 in 2005 whereas non-OECD members have an elasticity of 0.55 based on the higher share of energy expenditures. To take into account economic progress and convergence, the elasticity is assumed to fall exponentially to finally reach a value of 0.2 in the year 2150... These values result in reasonable primary energy demand projections for the respective story lines and baseline GDP and population projections.” (Emmerling et al. 2019).

Neither of these two approaches for generating IAM-based projections of emissions intensities produce results that are appropriate for our data-driven calculation. First, models that assume exogenous exponential declines in the emissions intensity of energy produce results qualitatively and quantitatively similar to our implementation of either globally-shared or country-specific exponential decay in emissions factors described above. For example, Barrage and Nordhaus (2024) find emissions per unit output declined at a rate of 1.5 percent per year historically and Krusell and Smith (2022) assume the emissions from energy use decline by 50 percent by 2125 implying a decay rate of 0.5 percent per annum from 1990 onward. Our factors in the Monte Carlo exercise cover a much broader range of potential decay rates for both forms of fuel, and are centered around similar but slightly more optimistic values for rates of decline. Given that our finding of a negative CAF is robust across a broad range of potential values for these rates of decline, the small differences between the existing factors in the literature listed above and our baseline global ones are unlikely to materially affect our results. Moreover, we calibrate our exogenous rates of decay using best-available data on historical changes in emissions intensities at the fuel-level, while many IAMs make calibration decisions that are based solely on aggregate historical outcomes or future projections (i.e., DICE and WITCH, respectively). Given that a key contribution of our paper is to compute an empirically-based CAF to complement prior work using CGE models (i.e., Colelli et al., 2022), we prefer to implement this approach by building directly from observational data.

Second, the output from models that combine assumptions of exogenous emissions intensity declines with a social planner choosing abatement (and thus further influencing emissions intensities) are conceptually incompatible with our CAF definition and calculation. The CAF represents the additional warming or cooling caused by adaptive energy use *conditional* on a set of plausible future emissions trajectories (we show results for multiple SSP-RCP combinations) that reflect the current market economy and possible future climate mitigation policies pursued by individual countries or coalitions (Riahi et al., 2017). Instead, these IAMs generate emissions factors that emerge as a result of either global or regional social planners choosing levels of

abatement to balance damages from climate change against costs of mitigation. The resulting “optimal” trajectories of emissions, energy use, and emissions intensities are by definition incompatible with a CAF calculation intended to reflect the current market economy, in which no global social planner is optimally managing global carbon emissions.

To ensure that our emissions factor assumptions are put properly in the context of prior literature, and to further underscore that our empirically-based CAF is complementary to the prior IAM-based estimate of effects of adaptation on warming that reflects an optimizing global social planner (Colelli et al., 2022), we have adjusted two sections of text. First, the text in the Results Section now reads (lines 170-179; edited text in red):

“We also recalculate the CAF relaxing assumptions over fixed emissions factors. **The long dashed lime green line in Figure 3a shows the CAF under the assumption that historical global trends in emissions intensity for both fuels between 2000 and 2018 continue for all countries through 2099, an assumption taken in prior related work on global climate damages (36; 37).** Allowing each sector to continue to decarbonize at historical rates lowers the CAF to -0.13°C under SSP2-RCP8.5. As emissions from other fuels have remained relatively unchanged on a unit basis when compared to the substantial reduction in the emissions intensity of electricity generation over the past two decades, extrapolation of this trend lowers emissions from the electricity sector and amplifies the mechanisms leading to a negative CAF (**Supplementary Information Section 3**).”

Second, the text in the Discussion Section now reads (lines 287-299; edited text in red):

“Our analysis has several limitations. First, we define the CAF relative to a fixed SSP for socioeconomic conditions and fixed RCP for baseline global emissions. The advantage of this approach is that SSPs and RCPs are widely used in climate projections and do not already account for emissions arising from energy-based adaptation (46). As discussed above and in Methods Section 5.4, this abstracts from a full characterization of the dynamic interplay between changing adaptive energy demand and climate change. One existing study used computational general equilibrium methods to endogenize the dynamic relationship between climate change and energy demand (22). The authors find, in contrast to our results, evidence of a positive climate feedback from adaptation. **This prior study is complementary to our analysis in that it computes the CAF under future conditions in which a global social planner optimally balances benefits and costs of mitigating climate change, while our data-driven CAF represents the climate feedback that emerges under plausible, but exogenous, future emissions trajectories.**”

More globally, the discrepancy between the extremely detailed regressions of point 1 and the overly simplistic factors used for point 2 is also of concern. Even if point 1 uses very detailed data, the overall level of confidence of any scientific result is always determined by the weakest point of the analysis. The fact that the energy demand response function relies on very fine data

cannot be used as a pretext to hide the fact that simplistic hypotheses on emission factors are one of the main drivers of the results.

As detailed above and evidenced by the implementation of various alternative scenarios for emissions factors, including a Monte Carlo simulation characterizing 1,000 different possible emissions factor decay rate combinations across countries and fuels, we do not believe our results in terms of the sign and magnitude of the CAF are sensitive to our choice of how to model the future path of emissions intensities. Moreover, our assumptions regarding emissions factors are transparent, data-driven, and align with multiple prior economic studies (as discussed in detail above). As discussed above, we have now made clear how our assumptions contrast with the complementary CGE and IAM literature that endogenizes emissions pathways over time based on decisions made by a global social planner.

Another general comment is the fact that the authors often tend to exaggerate their contribution. The term "Climate Adaptation Feedback", especially, is rather misleading. As stated by the authors, they do not capture future shifts in energy demand responses used to adapt to climate change driven by factors outside the two-factor model of Ref. 8 (changes in income and temperature) and they do not take into account any potential changes in building insulation levels, in heating or cooling technologies, or, more generally, any specific climate change adaptation strategies (urban greening etc.). For the sake of clarity I would recommend using a more specific term, such as "heating and cooling demand if countries keep present technologies and emission factors".

We thank the reviewer for raising the concern that the paper was not sufficiently clear with respect to what the CAF does and does not account for. In the paper, we introduce and define the CAF as a general concept, before proceeding to compute it with what we view is the best available data for doing so. Of course, that specific calculation has key limitations and we apologize that those were not sufficiently clear. In response, we have updated the text to ensure limitations of our calculation are evident and that our language does not exaggerate our contribution. However, we disagree with the reviewer's suggested term, as our CAF calculation does not measure only heating and cooling demand, nor does it assume present technologies or adaptation strategies are held fixed. Here, we explain precisely what is and is not captured by our CAF implementation before detailing the textual changes we have made in response to the reviewer.

The reviewer is correct that the model of Rode et al. (2021) forecasts future energy consumption responses to temperature based on the long-run moving average of income, heating degree days (HDDs), and cooling degree days (CDDs). The influence that each of these three factors has in shaping energy demand responses is empirically estimated using historical data, as detailed throughout our previous replies, implying that future demand response adaptations are projected using relationships estimated from recent history. However, this *does not* imply that changes in features such as building insulation, heating or cooling technologies, or targeted adaptation policies are fully ignored, as the reviewer implies. Nor does this modeling approach only capture heating and cooling, as the reviewer suggests – reduced-form regressions in Rode et al. recover elevated aggregate electricity or other fuels use due to short-run temperature fluctuations. This

increased consumption could come from heating and cooling, but also from any other energy-based activity that depends on temperature (e.g., water heating, appliance use, etc.). With aggregate data on final energy consumption, it is not possible to separately identify heating and cooling demand from other energy services, nor is it necessary for accurate calculation of the CAF.

Here we detail how the estimates we use from Rode et al. treat technological change and targeted adaptation policies – these features are not omitted, as the reviewer implies, but of course, they are captured only under certain assumptions.

Technology

Because energy demand responses *do* evolve in the future based on changing incomes and long-run climates, Rode et al. (2021) estimates only fail to account for future technological changes that are orthogonal to income and local climate. To the extent that features like building insulation and air conditioner efficiency are correlated with economic growth, such drivers of energy demand responses are embedded in the Rode et al. estimates of future adaptive energy use. Given a long literature documenting how per-capita income is representative of how close a country is to the technological frontier (Mankiw, Romer, and Weil 1992), this plausibly captures a substantial margin of demand adjustment over time. Similarly, to the extent that hotter or cooler climates have historically influenced building insulation levels, types of indoor heating and cooling technologies, and/or behavioral responses to heat and cold events, such adaptive responses are also included in Rode et al. estimates of future energy demand under warming. Of course, the Rode et al. (2021) model will fail to forecast the effects of technological changes insofar as they either are orthogonal to income growth and changing local climatology or represent marked departures from the historical correlation between changes in income and changes in temperature response functions.

Energy efficiency and public adaptation policy

As the reviewer suggests, future government policies may improve energy efficiency or subsidize public works projects that mitigate the local climatological effects of climate change. As with future technological change, such adaptive actions that may reshape energy demand responses are already accounted for in the Rode et al. estimates, to the extent that they have unfolded under historical climates and incomes. For example, if urban greening has been pursued in hotter and wealthier regions historically, with resulting impacts on electricity use on hot days, it will be accounted for when we project electricity use on future hot days in regions of the globe that have gradually warmed and increased in wealth. On the other hand, adaptive actions or investments that accelerate changes in energy demand responses for reasons orthogonal to income or local climate, or at rates that exceed those historically observed, will not be included in Rode et al. and therefore will not be accounted for in our CAF calculations.

Because our CAF estimate is intended to be empirically-based, we choose not to make assumptions over possible future policies or rates of technological change that are not evident in historic data. Finally, we note that our various sensitivity analyses on emissions factors,

performed in the Supplementary Information and here at more length, show that our result is highly robust to a wide range of efficiency improvements in terms of lower emissions per unit of energy used towards adaptation. However, to more explicitly address that we are not attempting to account for structural changes in energy demand responses caused by technological leaps or government policy, we have adjusted the Discussion Section to read (lines 336-347; edited text in red):

“... Our estimates may also fail to account for long-run changes in the response of energy demand driven by shifts in preferences or technological change, leading us to not fully capture some facets of the extensive margin for adaptation (9;20;21). If the price of electricity falls drastically in the future relative to that of other energy, we may underestimate electricity demand responses which would bias our CAF estimate downward. Conversely, if end-uses for electricity become more efficient in a way previously unexplained by income growth, our CAF will overstate future emissions from electricity. **More generally, our estimates of the CAF will not capture future energy demand responses to climate change driven by factors that are not captured by the two-factor model in ref. (8). This omits the potential for government policy or technological breakthroughs to cause structural shifts in how energy use responds to temperature changes.**”

References:

Auffhammer, M., Baylis, P., & Hausman, C. H. (2017). Climate change is projected to have severe impacts on the frequency and intensity of peak electricity demand across the United States. *Proceedings of the National Academy of Sciences*, 114(8), 1886-1891.

Barrage, Lint, and William Nordhaus. "Policies, projections, and the social cost of carbon: Results from the DICE-2023 model." *Proceedings of the National Academy of Sciences* 121, no. 13 (2024): e2312030121.

Casey, Gregory. "Energy Efficiency and Directed Technical Change: Implications for Climate Change Mitigation." *The Review of Economic Studies*. (Forthcoming).

Colelli, Francesco Pietro, Johannes Emmerling, Giacomo Marangoni, Malcolm N. Mistry, and Enrica De Cian. "Increased energy use for adaptation significantly impacts mitigation pathways." *Nature Communications* 13, no. 1 (2022): 4964.

Colelli, Francesco Pietro, Ian Sue Wing, and Enrica De Cian. "Air-conditioning adoption and electricity demand highlight climate change mitigation–adaptation tradeoffs." *Scientific Reports* 13, no. 1 (2023): 441.

Copernicus Climate Change Service, Climate Data Store, (2021): In situ total column ozone and ozone soundings from 1924 to present from the World Ozone and Ultraviolet Radiation Data Centre. Copernicus Climate Change Service (C3S) Climate Data Store (CDS). DOI: 10.24381/cds.99842490 (Accessed on 28-02-2024).

- Cruz, José-Luis, and Esteban Rossi-Hansberg. "The economic geography of global warming." *Review of Economic Studies* 91, no. 2 (2024): 899-939.
- De Cian, Enrica, and Ian Sue Wing. "Global energy consumption in a warming climate." *Environmental and Resource Economics* 72 (2019): 365-410.
- Emmerling, Johannes, Laurent Drouet, Lara Reis, Michela Bevione, Loic Berger, Valentina Bosetti, Samuel Carrara et al. "The WITCH 2016 model-documentation and implementation of the shared socioeconomic pathways." (2016).
- _____. "WITCH Documentation." (2019). https://doc.witchmodel.org/witch_documentation.pdf. Accessed May 4 2024.
- Everitt, B. S., & Skrondal, A. (2010). *The Cambridge dictionary of statistics*.
- Golosov, Mikhail, John Hassler, Per Krusell, and Aleh Tsyvinski. "Optimal taxes on fossil fuel in general equilibrium." *Econometrica* 82, no. 1 (2014): 41-88.
- Hassler, John, Per Krusell, and Conny Olovsson. "Directed technical change as a response to natural resource scarcity." *Journal of Political Economy* 129, no. 11 (2021): 3039-307
- Hsiang, S. (2016). Climate econometrics. *Annual Review of Resource Economics*, 8, 43-75.
- IEA (2020), *Metallurgical coal consumption, 2018-2021*, IEA, Paris <https://www.iea.org/data-and-statistics/charts/metallurgical-coal-consumption-2018-2021>, License: CC BY 4.0. Accessed May 7, 2024.
- IEA (2022), *Coal 2022*, IEA, Paris <https://www.iea.org/reports/coal-2022>, License: CC BY 4.0. Accessed May 7, 2024.
- Imbens, Guido W. "Statistical significance, p-values, and the reporting of uncertainty." *Journal of Economic Perspectives* 35, no. 3 (2021): 157-174.
- Kotz, Maximilian, Anders Levermann, and Leonie Wenz. "The effect of rainfall changes on economic production." *Nature* 601, no. 7892 (2022): 223-227.
- Krusell, Per, and Anthony A. Smith. "Climate Change Around the World." *NBER Working Paper* w30338 (2022).
- Mankiw, N. Gregory, David Romer, and David N. Weil. "A contribution to the empirics of economic growth." *The quarterly journal of economics* 107, no. 2 (1992): 407-437.
- Piontek, F., Drouet, L., Emmerling, J., Kompas, T., Méjean, A., Otto, C., ... & Tavoni, M. (2021). Integrated perspective on translating biophysical to economic impacts of climate change. *Nature Climate Change*, 11(7), 563-572.

Riahi, K., Van Vuuren, D. P., Kriegler, E., Edmonds, J., O'neill, B. C., Fujimori, S., ... & Tavoni, M. (2017). The Shared Socioeconomic Pathways and their energy, land use, and greenhouse gas emissions implications: An overview. *Global environmental change*, 42, 153-168.

Rode, Ashwin, Tamma Carleton, Michael Delgado, Michael Greenstone, Trevor Houser, Solomon Hsiang, Andrew Hultgren et al. "Estimating a social cost of carbon for global energy consumption." *Nature* 598, no. 7880 (2021): 308-314.

Tabari, Hossein. "Climate change impact on flood and extreme precipitation increases with water availability." *Scientific reports* 10, no. 1 (2020): 13768.

Van Ruijven, Bas J., Enrica De Cian, and Ian Sue Wing. "Amplification of future energy demand growth due to climate change." *Nature Communications* 10, no. 1 (2019): 2762.

Wenz, L., Levermann, A., & Auffhammer, M. (2017). North–south polarization of European electricity consumption under future warming. *Proceedings of the National Academy of Sciences*, 114(38), E7910-E7918.

REVIEWER 3

I think the authors have taken adequate steps to address my concerns and therefore from my perspective the paper can proceed to publication.

REVIEWER 1

I thank the authors for their detailed replies and recommend the paper for publication.

We thank Reviewer 1 for their feedback throughout the revision process.

REVIEWER 2

We thank Reviewer 2 for their additional comments. In response, we reproduce each of your comments italicized in gray and provide our replies in black. Excerpts from the paper are indented with the line numbers noted.

The authors did a nice job carrying out extra analyses to support the robustness of their methodological approach, and properly answered my comments regarding this topic.

However, I think that they did not really respond to my comment stating that they tend to exaggerate their contribution to the literature. Some claims made in the paper are indeed still slightly misleading, especially regarding the novelty of the topic and the comparison with existing results. The authors state that their main contribution is to "formalize the notion of the CAF and provide a tractable framework for its calculation". The CAF, as the authors indicate in the text, has already been presented in several papers, and it is difficult to blankly state, as the authors do, that they "formalize" it in any innovative way. It is also difficult to see why the existing assessment of the CAF, that the authors cite, was not computed through a "tractable framework".

Thank you for this feedback, we appreciate the reviewer's specific suggestions on how to more appropriately frame the contribution of our paper. In response, we have removed all mention of "formalizing" the climate adaptation feedback (CAF), and have also removed any language suggesting that our approach is more "tractable" than the prior methodology in Colelli et al. (2022).

The Introduction paragraph, where the sentences quoted by the reviewer used to be, now contains a more precise set of statements regarding how our analysis compares to the existing literature. This includes the set of papers that project adaptive energy demand under climate change, as well as the only paper that, to our knowledge, translates such projections into an estimate of a temperature feedback (i.e., Colelli et al. 2022). The text now reads (lines 42-64, changes in red):

“While our work is not the first to note the potential feedback between energy used for adaptation and the climate (16; 31; 32), we provide a new empirically-based framework for calculating the magnitude of this feedback as well as the uncertainty that surrounds it. Some prior work estimates the responsiveness of historical energy demand to weather and uses such estimates to project future global energy consumption under climate change (6; 16; 31), but these analyses do not assess the impacts of these changes in energy consumption on global temperatures. One paper does estimate a version of the CAF using a regional Integrated Assessment Model (IAM), modeling adaptation as changes in the productivity of different fuels used as inputs in production, and finds a positive feedback in which adaptive energy use raises global mean surface temperature (30). In contrast to this work, we do not impose the structure of an IAM, and in particular do not assume regional planners set optimal policy for energy use, adaptation, and mitigation (33). Instead, we take a data-driven approach, assessing how adaptation to climate change will affect future temperatures given empirically-estimated energy demand responses and exogenous future trajectories of baseline warming, economic growth, and population (see

Methods). Relative to this prior work, our study increases the spatial resolution of energy demand responses used to forecast changes in emissions, includes the response to the full distribution of daily temperature realizations rather than restricting inputs to realizations of local extremes, reports uncertainty from both climatological and statistical sources, and accounts for country-level heterogeneity in emissions intensities and their dynamics for electricity and other fuels. However, our empirically-based method necessarily foregoes modeling general equilibrium channels — namely how adaptation will affect prices, expectations, and investment — that are captured by IAMs, making the two approaches complementary.”

I have the impression that the main contribution of this paper is to assess the CAF with a new approach, and to show that it leads to different results than the 2022 paper by Colelli et al. As the authors explained in their rebuttal letter, existing papers use IAMs to assess the CAF or future energy demand responses to climate change impacts. Such an approach comes at the expense of granularity. The approach used here relies on historical data, and enables [sic]to take into account much finer changes in temperature variations and a much larger spatial heterogeneity. However, it can't take into account any voluntary government policy or technological innovations in the scenarios derived, as IAMs do. There is a long-standing debate on the respective qualities and drawbacks of both approaches, and it is not really possible to tell if one is really more informative than the other. The main policy-relevant message from the paper, to my sense, is therefore not that the CAF is negative, but that, depending on the approach used to assess it, there is an uncertainty regarding its sign and its magnitude.

I would recommend to clearly state this at the beginning of the text and in the abstract. In section 2, some figure should be given about the results obtained by Colelli et al. (2022) when presenting the concept of the CAF, and, in the methods section, the key differences between the approach used here and in this paper should be explained.

We appreciate the reviewer’s careful differentiation of the two methods – integrated assessment modeling and bottom-up statistical approaches – used to calculate the effects of the energy-adaptation feedback on global temperatures. We agree that they are highly complementary, in that each approach has distinct costs and benefits that makes it better suited for specific policy experiments. Both approaches are used throughout the climate change impacts literature (Piontek et al. 2021) and have been used to guide policy such as the U.S. government’s social cost of carbon (EPA 2023). We have followed the reviewer’s guidance to update multiple parts of the text to more precisely characterize how our contribution builds on and complements Colelli et al. (2022), highlighting the benefits and costs of both approaches. These changes include changes to the Title, Abstract, Introduction, Discussion, and Methods sections, each of which we detail below.

Title: In response to this comment and guidance from the Editor, we have edited our title from its broad prior language (“The Climate Adaptation Feedback”) to a more precise summary of our contribution (“Quantifying the global climate feedback from energy-based adaptation”).

Abstract: We appreciate the guidance from the reviewer to better contextualize our contribution in the abstract. The abstract no longer mentions novelty or the CAF directly, nor does it make broad statements about the “wide-ranging implications” of the CAF. It now reads (lines 2-14, changes in red):

“Many behavioral responses to climate change are carbon-intensive, raising concerns that adaptation may cause additional warming. **The sign and magnitude of this feedback depend on how increased emissions from cooling balance against reduced emissions from heating across space and time. We present an empirical approach that forecasts the effect of future adaptive energy use on global average temperature over the 21st century.** We find energy-based adaptation will lower global mean surface temperature in 2099 by 0.12°C (0.07°C) relative to baseline projections under RCP8.5 (RCP4.5) and avoid 1.8 (0.6) trillion USD (\$2019) in damages. Energy-based adaptation lowers business-as-usual emissions for 85% of countries, reducing the mitigation required to meet their unilateral Nationally Determined Contributions under the UNFCCC by 20% on average. **These findings indicate that while business-as-usual adaptive energy use is unlikely to accelerate warming, a reevaluation of existing mitigation commitments is merited.**”

Introduction: We amend the Introduction and framing of the CAF to reflect that we, unlike prior work, do not use an IAM and instead rely on empirical methods. As described in response to the reviewer’s prior comment, the third paragraph of the Introduction has been amended to no longer claim to have formalized the concept of the CAF or to have a more tractable framework; instead, the focus is on using a data-driven approach to assess the CAF, which has not yet been done. In that paragraph (copied above), we additionally make clear that one key result from Colelli et al. (2022) is a positive warming feedback, thus including a discussion of results from this paper when presenting the CAF, as requested by the reviewer. In addition, we have also amended the last two paragraphs of the Introduction to detail how our method differs in spirit from Colelli et al. (lines 130-140, changes in red):

“To obtain the CAF, we calculate the annual cumulative change in global CO₂ emissions due to energy-based adaptation for horizons from 2020 to 2099 (Figure 2c and Equation (6)). We then translate these cumulative emissions into a change in global temperatures (Δ GMST) using an empirically-derived relationship that leverages simulated warming from an ensemble of Global Climate Models for the two emissions pathways we consider (Figure 2d; Methods). **This method contrasts with other approaches (c.f. 30) in that we abstain from integrating the energy demand responses from ref. (8) into an IAM, where adaptation through energy use is an endogenous outcome when agents solve a fully-specified optimization problem. This allows us to be neutral on how future policy may affect adaptive energy use and instead calculate the CAF as a function of future adaptation solely based on empirical estimates of the effects of historical temperature variation on energy consumption.**”

Discussion: As highlighted by the reviewer, a key difference between our analysis and Colelli et al. (2022) is that we do not make use of an Integrated Assessment Model (IAM) to go from projected changes in energy demand to changes in temperatures. As such, our approach remains neutral on future government policy and climate goals conditional on a baseline SSP-RCP scenario, and does not consider

how demand for energy used for adaptation affects optimal policy choices regionally or carbon taxes globally (as in Colelli et al. 2022). Instead, our contribution centers around the fact that we: (1), estimate a CAF for decentralized economic conditions without optimizing social planners; (2), use direct estimates of how climate change affects energy demand based on historical responses that incorporate how such responses will change under future warming and economic growth; and (3), provide uncertainty associated with our estimate (deriving from both climatological and statistical sources). We have updated our Discussion text to elucidate these differences and, in accordance with the reviewer's requests, report the specific results from Colelli et al. that contrast with our own. The Discussion now reads: (lines 310-340, changes in red):

“Our data-driven approach combined with the updated estimates of energy demand responses we use from ref. (8) leads our findings to contradict several existing estimates in the literature, drawn from both reduced-form and computational general equilibrium methods. Prior work projecting energy demand under SSP-RCP scenarios using alternative demand response estimates projects that, in contrast with ref. (8), aggregate energy demand will rise due to adaptation to climate change (16; 17). The other contemporary paper that considers how adaptation affects the trajectory of climate change uses an IAM to do so and, contrary to our study, finds a positive climate feedback from adaptation: ref. (30) estimate that adaptation increases cumulative emissions in 2099 by approximately 347 GtCO₂-equivalents, which in our framework corresponds to a CAF of approximately +0.17°C.

There are numerous differences between prior studies and ours that may contribute to such divergent results (see Methods). First, our projections of the energy demand response to climate change stem from ref. (8)'s approximately 25,000 heterogeneous empirically-derived fuel-specific energy demand responses representing subnational units across the globe. These estimates include demand responses to the full distribution of daily temperature realizations, comprising both moderate and extreme temperatures. In contrast, refs. (17) and (30) use earlier empirical estimates from ref. (16), who estimate two regional fuel-specific demand responses to the annual frequency of average daily low (<12.5°C) and high (>27.5°C) temperature extremes. Second, while we directly estimate how climate change affects final energy demand using historically observed behavior, ref. (22) model adaptation as the response of global energy demand to temperature-driven changes in the productivity level of energy inputs in economic production. The energy demand induced by adaptation is then computed as the difference in aggregate energy use between the model where climate change affects productivity levels and an alternative specification where fuel-specific productivity levels are temperature-invariant. Third, we account for country-level heterogeneity in emissions intensities, as opposed to heterogeneity across only 17 global macro-regions (30). In sum, while ref. (30) computes the CAF associated with energy productivity shocks from climate change under conditions in which regional planners balance the benefits and costs of responding to climate change, our data-driven approach estimates the CAF for a future where agents adapt to climate change in line with historical behavior. While many data inputs also differ, these distinct methodologies render the approaches highly complementary.”

Methods: We appreciate the reviewer’s suggestion to provide a discussion of the methodology in Colelli et al. (2022) and how it differs from our own in the Methods section of our manuscript. We have now done so in a new Methods subsection titled “Comparison with the Integrated Assessment Model approach” (lines 580-633; changes in red):

“Comparison with the Integrated Assessment Model Approach

Unlike prior literature, we do not use an Integrated Assessment Model (IAM) to estimate the effects of adaptive energy demand on global temperatures. This abstracts from fully specifying a demand system wherein agents balance the benefits of using energy for adaptation against the costs of foregoing income for other forms of consumption. Our data-driven approach avoids making structural assumptions governing the channels through which adaptation to climate change occurs, which an IAM framework usually imposes. This abstention from using an IAM of course poses a tradeoff in that our empirically-based CAF omits general equilibrium channels that may affect energy demand, as highlighted in the Discussion section.

The benchmark study to date examining the potential for feedback between energy used for adaptation and global temperatures (30) uses the WITCH IAM (33) framework. WITCH is a multi-region IAM based on an augmented Ramsey model of optimal savings designed to assess the welfare and temperature effects of climate policy in a dynamic setting (33). Agents in the model consist of regional planners who select sequences of consumption, savings, and energy use in their local economy so as to maximize residents’ welfare (33). The model allows for both a non-cooperative formulation, where up to 17 regional planners maximize the welfare of their populations in a global extensive game, as well as a cooperative formulation under which a single social planner maximizes welfare across all regions globally. Ref. (30) use the non-cooperative formulation of the model as their laboratory for examining the effects of adaptation on energy demand and temperatures. While the non-cooperative formulation is the closest version of the model to a competitive equilibrium, unlike in a decentralized economy, regional planners in the non-cooperative formulation still internalize some climate externalities. These planners choose mitigation and/or abatement efforts to maximize regional welfare; the Nash equilibrium in the non-cooperative formulation of WITCH is a second-best optimal solution, as opposed to a model of outcomes under laissez-faire (33).

To use the WITCH model to estimate the effects of adaptation on energy use, ref. (30) modify the model to allow climate change to affect the productivity of different fuels used as intermediate inputs for producing the modeled final consumption good. Adaptation inside of this alternate specification is taken to be how energy use changes in response to the productivity shocks driven by future climate change (30). The authors then calculate aggregate demand in the alternative (baseline) model specifications that include (exclude) the adaptive energy demand induced by the productivity shocks for three scenarios: no climate policy, a 2.5C limit on warming, and a <2°C limit on warming. For each scenario,

ref. (30) solve both specifications of the model in the non-cooperative mode and raise carbon taxes until the resulting Nash equilibrium yields a level of aggregate emissions that coincides with the policy goal in the scenario. Ref. (30) quantify the additional energy demand (and in turn, emissions) induced by adaptation as the difference between aggregate energy demand in the alternative specification with adaptation and a baseline version of the WITCH model where adaptation is absent and energy-specific productivity levels are temperature-invariant.

In contrast, our data-driven approach is agnostic regarding the underlying structure of the economy, relying instead on empirical estimates of the historical response of energy demand to temperature fluctuations. Our methodology trades the structure of an IAM for increased spatial granularity and a greater ability to account for both statistical uncertainty in future demand responses as well as model uncertainty in how future temperature dynamics will respond to emissions. To do so, we use existing forecasts from ref. (8) that predict how future climate change will affect energy demand based on realized historical adaptation to intra-annual temperature changes. This avoids measuring adaptation as the change in energy used by regional social planners or assuming that the shifts in demand are driven by productivity shocks and/or changes in relative prices. At the same time, the lack of structure precludes us from accounting for future government policy or voluntary actions that may alter how energy demand responds to climate change beyond historical patterns. In that sense, we sacrifice the ability to estimate what the CAF would be in a world where global policy governing emissions or carbon prices change (c.f. (30)).”

As a large part of the difference is coming from the choice of future energy changes projections, the authors should also present the main differences between the projection by Rode et al. (2021) and the other existing projections, both in terms of methodology and in terms of global results.

We appreciate the reviewer’s suggestion for discussing the differences between the future energy projections from Rode et al. (2021) and those from prior literature more thoroughly. We now make clear that the projections for energy demand we use differ from other findings in the literature, and that this difference in inputs is critical in shaping our results with regard to the resulting changes in temperature. We also explicitly compare the estimation procedures and results from De Cian and Sue Wing (2019), Van Ruijven et al. (2019), and Colelli et al. (2022) with our own in the updated Discussion section (lines 310-340, changes in red):

“Our data-driven approach combined with the updated estimates of energy demand responses we use from ref. (8) leads our findings to contradict several existing estimates in the literature, drawn from both reduced-form and computational general equilibrium methods. Prior work projecting energy demand under SSP-RCP scenarios using alternative demand response estimates projects that, in contrast with ref. (8), aggregate energy demand will rise due to adaptation to climate change (16; 17). The other contemporary paper that considers how adaptation affects the trajectory of climate change uses an IAM to do so and, contrary to our study, finds a large positive climate feedback from adaptation: ref. (30) estimate that adaptation increases cumulative emissions in 2099 by approximately 347

GtCO₂-equivalents, which in our framework corresponds to a CAF of approximately +0.17°C.

There are numerous differences between prior studies and ours that may contribute to such divergent results (see Methods). First, our projections of the energy demand response to climate change stem from ref. (8)'s approximately 25,000 heterogeneous empirically-derived fuel-specific energy demand responses representing subnational units across the globe. These estimates include demand responses to the full distribution of daily temperature realizations, comprising both moderate and extreme temperatures. In contrast, refs. (17) and (30) use earlier empirical estimates from ref. (16), who estimate two regional fuel-specific demand responses to the annual frequency of average daily low (<12.5°C) and high (>27.5°C) temperature extremes. Second, while we directly estimate how climate change affects final energy demand using historically observed behavior, ref. (30) model adaptation as the response of global energy demand to temperature-driven changes in the productivity level of energy inputs in economic production. The energy demand induced by adaptation is then computed as the difference in aggregate energy use between the model where climate change affects productivity levels and an alternative specification where fuel-specific productivity levels are temperature-invariant. Third, we account for country-level heterogeneity in emissions intensities, as opposed to heterogeneity across only 17 global macro-regions (30). In sum, while ref. (30) computes the CAF associated with energy productivity shocks from climate change under conditions in which regional planners balance the benefits and costs of responding to climate change, our data-driven approach estimates the CAF for a future where agents adapt to climate change in line with historical behavior. While many data inputs also differ, these distinct methodologies render the approaches highly complementary.”

Minor comments:

- line 296/297 "This prior study is complementary to our analysis in that it computes the CAF under future conditions in which a global social planner optimally balances benefits and costs of mitigating climate change"  if I am not mistaken, the WITCH model used by Colelli et al. does not balance benefits and costs of mitigating climate change (i.e. it does not compute an optimal climate change mitigation level), but balances the benefits and costs of various mitigation policies under the constraint that it should reach a prescribed CO₂ emission trajectory corresponding to some SSP/RCP scenarios

We appreciate the reviewer highlighting this subtle difference and apologize for the lack of clarity and precision in this statement. The reviewer is correct that in the two policy scenarios that keep warming below a specific level in Colelli et al. 2022 (Table 5, rows two and three), climate change targets are imposed on regional social planners through a global carbon tax. However, this does not imply that mitigation decisions are imposed exogenously; even in this constrained version of the optimization problem in WITCH, regional planners still make decisions over emissions so as to maximize local welfare subject to the carbon taxes they face (Emmerling et al. 2016). This entails explicit tradeoffs in terms of the policy levers (e.g., mitigation versus abatement) that the planners use such that their social benefit from emissions matches the costs they pay in taxes that are imposed in the model run to meet given

temperature goals. For example, when a planner uses either the abatement effort module (Emmerling et al. 2019 Section 7.1.2) or the carbon capture and storage (CCS) module (Emmerling et al. 2019 Section 4.7) in WITCH to lower local emissions, they make a tradeoff between the costs of abatement today (foregone consumption) and its benefits in terms of emissions reductions that lower the carbon taxes they pay now (or in the future).

To ensure there is no confusion regarding these points, we have added substantially more text describing the Colelli et al. approach in the Introduction, Discussion, and Methods sections, as detailed in our response to the prior comment above. Additionally, we have removed the quoted sentence in this revision and replaced it with more precise phrasing regarding the differences in methodology between Colelli et al. (2022) and our own paper. This updated text now reads (lines 335-340):

“In sum, while ref. (30) compute the CAF associated with energy productivity shocks from climate change under conditions in which regional planners balance the benefits and costs of responding to climate change, our data-driven approach estimates the CAF for a future where agents adapt to climate change in line with historical behavior. While many data inputs also differ, these distinct methodologies render the two approaches highly complementary.”

- line 310/311 "The use of a fixed SSP-RCP baseline prohibits us from examining the CAF in tandem with decarbonization scenarios more sophisticated than extrapolating historical trends."  I do not understand this sentence. In different SSP-RCP scenarios, decarbonization is supposed to occur at different speeds. Why would using exogenous decarbonization scenarios make less sense than [sic] using a fixed historical trend [sic]?

We appreciate the reviewer pointing out that this statement was unclear and misleading. The reviewer is absolutely correct that each SSP-RCP combination embeds a pathway for global emissions and income levels that, when combined, induces a pathway for decarbonization specific to that scenario (Riahi et al. 2017). However, these implied pathways do not include country-level outputs indicating whether the source of the dynamics will be driven by a lower energy-intensity of output (energy per GDP) or emissions intensity of energy (emissions per unit of energy consumed). Thus, it is not feasible to extract from these scenarios emissions intensities over time that could be used to estimate the CAF. Moreover, there are fairly serious internal inconsistencies with the implied rates of growth of emissions intensity of GDP that the SSP-RCP pairs imply when taken at face value (Welch 2024).

In addition to feasibility, there is a conceptual reason to choose fixed or data-driven emissions factors when computing the CAF conditional on SSP-RCP scenarios. Our approach represents a data-driven, partial-equilibrium, estimate of the CAF constructed assuming a future global market economy that reflects the decentralized equilibrium of today, as opposed to the solution to a planner’s problem (or multiple regional planners’ problems, as in Colelli et al. 2022). In such a decentralized setting, each country’s energy demand response and its emissions implications are computed taking the SSP-RCP as exogenously given. Realizing the specific decarbonization assumptions embedded in each SSP-RCP would require some degree of centralized policy enforcement; instead of taking a stand on such

coordinated policy efforts, we estimate the CAF in partial equilibrium using historical data from today's decentralized economy to inform energy demand and emissions responses.

To clarify why our baseline CAF estimate relies on fixed or statistically-based dynamic emissions factors, we have expanded our consideration of this issue in the Discussion section. We now speak to the inherent inconsistency in our combining positive analysis informed only by historical behavior with projections of climate change and economic growth that are driven in part by normative modeling decisions that underpin SSP-RCP combinations. We clarify why this inconsistency arises – namely the challenges that preclude us from extracting country- and fuel-specific decarbonization pathways from the SSP-RCP scenarios – and describe why this leads us to take the approach we do. The Discussion section now speaks to the limitations of our approach with more precision, and reads (Lines 341-368, changes in red):

“Our analysis has several limitations. First, we define the CAF relative to a fixed SSP for socioeconomic conditions and fixed RCP for baseline global emissions. The advantage of this approach is that SSPs and RCPs are widely used in climate projections and do not already account for emissions arising from energy-based adaptation (45). **The disadvantage is that we cannot readily map the narrative-driven assumptions surrounding global emissions and energy use implied by each SSP-RCP scenario into our framework. This leads to an inconsistency between our use of fixed or exponentially declining emissions factors across all SSP-RCP combinations and the variation in the implied carbon intensity of output induced by the aggregate time series specific to each scenario. However, the baseline coupled SSP-RCP scenarios do not contain sufficient information for us to extract the fuel- and country-specific emissions factors that we would need in order to map the implied variation in decarbonization pathways onto our projections of energy demand. We instead assume that emissions factors stay at measured levels or fall according to current empirically-estimated exponential trends, as these measurements directly reflect the current state of the world and allow us to minimize the normative assumptions we make regarding decarbonization.**

While completely remedying this inconsistency would require a fully coupled approach in which behavioral adaptations are built directly into the modeling forming the SSP-RCP scenarios, our key qualitative findings are unlikely to be affected by such a change for two reasons. First, we conduct a sensitivity analysis that captures one dimension of such a coupling; our dynamic version of the CAF shown in Figure 3 recalculates the energy demand projections from ref. (8) to account for the endogenous response of temperature to prior years' energy demand induced by the CAF itself. Results are virtually unchanged. Second, as long as CO₂ intensities associated with non-electricity energy use do not decline dramatically relative to those for electricity, our estimate should serve as an upper bound on the magnitude of the CAF when general equilibrium effects are small. For example, if the electricity sector continues to decarbonize faster than other fuels (46), there will be fewer additional emissions from increased electricity consumption to offset the decreased emissions from other fuels, implying a more negative CAF than the value we have estimated here.”

References:

Colelli, Francesco Pietro, Johannes Emmerling, Giacomo Marangoni, Malcolm N. Mistry, and Enrica De Cian. "Increased energy use for adaptation significantly impacts mitigation pathways." *Nature Communications* 13, no. 1 (2022): 4964.

De Cian, Enrica, and Ian Sue Wing. "Global energy consumption in a warming climate." *Environmental and Resource Economics* 72 (2019): 365-410.

Emmerling, Johannes, Laurent Drouet, Lara Reis, Michela Bevione, Loic Berger, Valentina Bosetti, Samuel Carrara et al. "The WITCH 2016 model-documentation and implementation of the shared socioeconomic pathways." (2016).

Environmental Protection Agency (EPA). "Report on the Social Cost of Greenhouse Gases: Estimates Incorporating Recent Scientific Advances." (2023). *Docket ID No. EPA-HQ-OAR-2021-0317*. https://www.epa.gov/system/files/documents/2023-12/epa_scghg_2023_report_final.pdf. Accessed July 28th, 2024.

_____. "WITCH Documentation." (2019). https://doc.witchmodel.org/witch_documentation.pdf. Accessed May 4 2024.

Piontek, Franziska, Laurent Drouet, Johannes Emmerling, Tom Kompas, Aurélie Méjean, Christian Otto, James Rising, Bjoern Soergel, Nicolas Taconet, and Massimo Tavoni. "Integrated perspective on translating biophysical to economic impacts of climate change." *Nature Climate Change* 11, no. 7 (2021): 563-572.

Riahi, Keywan, Detlef P. Van Vuuren, Elmar Kriegler, Jae Edmonds, Brian C. O'neill, Shinichiro Fujimori, Nico Bauer et al. "The Shared Socioeconomic Pathways and their energy, land use, and greenhouse gas emissions implications: An overview." *Global environmental change* 42 (2017): 153-168.

Rode, Ashwin, Tamma Carleton, Michael Delgado, Michael Greenstone, Trevor Houser, Solomon Hsiang, Andrew Hultgren et al. "Estimating a social cost of carbon for global energy consumption." *Nature* 598, no. 7880 (2021): 308-314.

Van Ruijven, Bas J., Enrica De Cian, and Ian Sue Wing. "Amplification of future energy demand growth due to climate change." *Nature Communications* 10, no. 1 (2019): 2762.

Welch, Ivo. "The IPCC Shared Socioeconomic Pathways (SSPs): Explained, Evaluated, Replaced." No. w32178. *National Bureau of Economic Research*, 2024.

REVIEWER 2

We thank Reviewer 2 for their additional comments. In response, we reproduce each of your comments italicized in gray and provide our replies in black. Excerpts from the paper are indented with the line numbers noted.

Major Comments

=====

The authors did a nice job modifying the text, which now much clearer [sic], especially regarding its contribution to the literature.

Thank you.

These modifications have only been done in the methodology and introduction sections, however, and the conclusion sections of the paper have not been updated (especially the section "Implications for mitigation policy"). They are still written, therefore, as if the results of this paper were definitive, and not a complementary view in an existing debate. That is why I would suggest removing all the blank [sic] statements, such as "Our findings, inclusive of statistical and climatological uncertainty, indicate that even a modest positive CAF is highly unlikely, alleviating such concerns." and replace them by actual discussions of what the new results add to the debates. For instance, it could be written "Our findings indicate that a positive CAF may not be as likely as thought before".

To address this concern, the above sentence now reads (lines 151-153; edited text in red):

“Our approach, inclusive of statistical and climatological uncertainty, suggests that a modest positive CAF is unlikely, alleviating such concerns.”

Replacing “Our findings..indicate” with “Our approach...suggests” provides a caveat that these results are specific to our empirical framework. The change from “even a modest positive CAF is highly unlikely” to “a modest positive CAF is unlikely” further moderates the language.

In addition, we have also combed through the paper making sure every sentence that discusses our results is appropriately moderated. Specifically, we have changed the following passages (line numbers noted and edited text in red):

In lines 153-154: “Finally, a positive CAF has ramifications...” now reads “Finally, a non-zero CAF has ramifications...”

In lines 430-431: “... necessitates a reevaluation of existing mitigation commitments.” now reads “... has implications for countries' existing mitigation commitments.”

In lines 433-434: "... may lead to a false measure of policy stringency." now reads "... may lead to **an inaccurate** measure of policy stringency."

In lines 439-440: "More broadly, a negative CAF highlights the inherent link between climate mitigation and adaptation" now reads "More broadly, a **non-zero** CAF highlights the inherent link between climate mitigation and adaptation."

Moreover, in the "Implications for mitigation policy" section, the authors should discuss to what extent their hypotheses may be more policy-relevant, or not, than what was done in the existing literature.

We have inserted the following sentence at the start of this section which sets up this discussion in the subsequent three paragraphs (lines 251-253; edited text in red):

"Heterogeneity in country-level adaptation-induced CO₂ emissions can inform the fairness and stringency of countries' Nationally Determined Contributions (NDC) from the Paris Agreement of the United Nations Framework Convention on Climate Change (UNFCCC)."

To our knowledge, previous papers in this literature have not explored the implications of country-level adaptation-induced CO₂ emissions for existing Nationally Determined Contributions. This section of our text does this explicitly, showing both concerns regarding fairness (i.e., countries with large adaptation-induced emissions are also large historical emitters) and stringency (i.e., adaptation-induced emissions abatement in some countries are a large portion of what their required abatement to meet existings NDCs would be in the absence of adaptation). Both policy-relevant implications are new to the literature – we therefore do not see it as necessary, especially given the space constraints at the journal, to discuss prior literature in this section of the paper.

Other Comment

=====

"First, we conduct a sensitivity analysis that captures one dimension of such a coupling; our dynamic version of the CAF shown in Figure 3 recalculates the energy demand projections from ref. (8) to account for the endogenous response of temperature to prior years' energy demand induced by the CAF itself. Results are virtually unchanged"

 This computation is interesting, and useful in the paper, however, I do not see why doing this actually captures one dimension of such a coupling? I would suggest removing this sentence to keep only the second argument "as long as CO₂ intensities associated with non-electricity energy use do not decline dramatically relative to those for electricity, our estimate should serve as an upper bound on the magnitude of the CAF when general equilibrium effects are small"

We believe it is important to keep the current text as is. This part of the Discussion considers the consequences of assuming a fixed emissions scenario for baseline global emissions, a standard assumption in many statistically-based projections of future climate impacts. Emissions through energy-

based climate adaptation could alter that baseline through changes in overall energy demand. While explicitly modeling a full forward-looking dynamic interplay between climate change and adaptive energy demand would require an entirely new analytical framework, the logic of this section is to probe what we can say about this channel within our statistical framework. The exercise is a perturbation approach around our static estimates which tests the sensitivity of our findings to relaxing the assumptions of a fixed pathway for temperature in the baseline and counterfactual scenarios. The dynamic version of our CAF mentioned in the quoted text (and presented in detail in the Appendix) exactly captures this feedback: it shows how forecasted adaptation in 2024 affects forecast temperatures in 2025, which in turn alters energy demand in 2025 and so forth. Importantly, this exercise shows that accounting for this dynamic feedback between adaptation in energy demand and temperature does not alter our estimated CAF, showing that our estimate is robust to concerns about endogenous changes in overall energy demand.

Also, if I am not mistaken, your argument also holds only if there is no switch from non-electricity energy use to electricity energy use (e.g. people switching from coal or gas heating systems to electric heating systems, or people switching from gasoline cars to electric cars). As the electrification of energy uses is a common policy goal in many countries, and is a trend which is widely observable, this precision is important, and questions the relevance of the approach.

We believe there is a misunderstanding here. First, any switch between electricity and non-electricity that is driven by changes in income or long-run climate is already captured by the slope coefficients in Rode et al. (*Nature*, 2021)'s estimates, which vary with projected levels of income and long-run climatology. This allows us to capture any electrification policy goals that are explained by income growth and gradual climate change.

Accounting for the effects of electrification processes that unfold independently from income growth or a warming climate is more nuanced. We read the reviewer's concern in this regard as follows:

“Suppose governments rapidly implement electrification policies worldwide which replace existing technologies which use other fuels with those that use electricity. How would your estimates change given that this would make future demand responses steeper in electricity and flatter in other fuels than those you estimate on historical data?”

Assuming we have read this correctly, results from our paper indicate that rapid electrification worldwide would not qualitatively change our result. This is essentially because energy demand increases due to cold temperatures (e.g., for indoor heating and cooking) would still need to be met in an electrified world – even if this future demand were met by electricity instead of other fuels due to broad-based electrification, energy demand declines due to fewer cold days under climate change will remain. Whether aggregate demand for energy services increases under climate change (due to higher demand from more hot days) or decreases (due to lower demand from fewer cold days) is an empirical question. Our results, derived from projections in Rode et al. (*Nature*, 2021), clearly indicate that aggregate demand falls. This can be seen in Figure R.1, where net change in energy demand across both electricity and other fuels is shown to be negative and declining under future climate change.

Figure R.1: Aggregate energy demand from adaptation under climate change. Figure shows Rode et al. (*Nature*, 2021) forecasts of global changes in energy demand due to adaptation, disaggregated by electricity (blue line) and other fuels (red line), over the next 100 years. Net change in energy demand across both fuel sources is shown in the black dotted line. All values shown derive from the RCP8.5 baseline emissions scenario.

Of course, the emissions implications of electrification – not just projected net changes in energy demand – are critical to determining the climate adaptation feedback. Replacing cold-related energy demand currently met with other fuels by electricity is likely to lower emissions feedbacks, as electricity is, on average, less carbon-intensive than other fuels (see the maps in Figure 2). However, the direction of the CAF would remain negative, as aggregate energy demand remains negative (Figure R.1 and Figure 2c in Rode et al. (*Nature*, 2021)). Panel b of Figure 3 in our manuscript (reproduced as Figure R.2 below) additionally illustrates this point: if all fuels globally were equally carbon intensive (rightmost gray bar), we still see adaptation reducing future warming, as aggregate emissions decline alongside aggregate energy demand.

Figure R.2: Replication of panel b of Figure 3 in the main text. The Figure shows the global CAF estimate if emissions factors were held constant across fuels at the global level

To ensure this aspect of our CAF calculation is clear in the text, we have added the following passage into the Discussion section, after the sentence the referee referred to. This text now reads (lines 370-373; edited text in red):

“Finally, because adaptation lowers the global net demand for energy services across all fuel sources, our estimate of the CAF would be negative even if all energy services required to meet future adaptive demand were provided by electricity.”

Minor comments

=====

In the abstract: "These findings indicate that while business-as-usual adaptive energy use is unlikely to accelerate warming, a reevaluation of existing mitigation commitments is merited"

 I am not convinced by the last statement that you introduce in your abstract, and would suggest removing the sentence "a reevaluation of existing mitigation commitments is merited".

We have softened the language in this abstract sentence, replacing “reevaluation” with “implications. It now reads (lines 12-14; edited text in red):

“These findings indicate that while business-as-usual adaptive energy use is unlikely to accelerate warming, it raises important implications for countries' existing mitigation commitments.”

We believe such implications are nonetheless sufficiently noteworthy to be included in the abstract: adaptation-induced CO₂ emissions abatement comprise 20 percent of NDCs for 52 of the 63 countries with NDCs. That is, we think it important to convey the point that our estimates imply many existing NDCs are too lenient.

"This allows us to be neutral on how future policy may affect adaptive energy use and instead calculate the CAF as a function of future adaptation solely based on empirical estimates of [sic]"

There may have been a communication error, as this comment does not appear to be completely transmitted to us. As such, we are unable to address it.

References:

Rode, Ashwin, Tamma Carleton, Michael Delgado, Michael Greenstone, Trevor Houser, Solomon Hsiang, Andrew Hultgren et al. "Estimating a social cost of carbon for global energy consumption." *Nature* 598, no. 7880 (2021): 308-314.